# MT-DAO: MULTI-TIMESCALE DISTRIBUTED ADAPTIVE OPTIMIZERS WITH LOCAL UPDATES

**Alex Iacob**[†,1,2]     **Andrej Jovanović**[*,1]     **Mher Safaryan**[*,3]     **Meghdad Kurmanji**[1]

**Lorenzo Sani**[1,2]     **Samuel Horváth**[4]     **William F. Shen**[1]     **Xinchi Qiu**[1]

**Nicholas D. Lane**[1,2]

## ABSTRACT

Training large models with distributed data parallelism (DDP) requires frequent communication of gradients across workers, which can saturate bandwidth. *Infrequent* communication strategies (e.g., Local SGD) reduce this overhead but, when applied to adaptive optimizers, often suffer a performance gap relative to *fully synchronous* DDP. We trace this gap to a time-scale mismatch: the optimizer's fast-moving momentum, tuned for frequent updates, decays too quickly to smooth gradients over long intervals, leading to noise-dominated optimization. To address this, we propose MT-DAO, a family of optimizers that employs multiple slow- and fast-moving first momenta or the gradient to track update dynamics across different time scales, for which we provide the first convergence guarantees. Empirically, for language-model pre-training, this eliminates the performance gap with DDP, outperforming infrequent-communication baselines in perplexity and reducing iso-token wall-clock time relative to DDP by 6–27% on Ethernet interconnects. At the 720M scale, MT-DAO reaches a target perplexity in 24% fewer steps and 35% less time than the single-momentum DDP baseline. MT-DAO enables effective cross-datacenter training and training over wide geographic areas.

## 1 INTRODUCTION

The scalability of training infrastructure is impeded by the communication required for Distributed Data Parallelism (DDP). Infrequent parameter-averaging strategies like Local SGD (Stich, 2019) reduce this overhead, yet extensions to adaptive optimizers (Cheng & Glasgow, 2025; Charles et al., 2025) show a performance gap relative to DDP (Sani et al., 2025; Charles et al., 2025). Charles et al. (2025) finds that infrequent averaging, even with Nesterov momentum at round boundaries (Reddi et al., 2021), underperforms DDP for models up to 2.4B parameters and worker counts exceeding 2.

We hypothesize this gap stems from a *timescale mismatch*. Optimizers use fast-moving momenta (low $\beta_1 \approx 0.9$) that smooth high-frequency noise under DDP but *decay too rapidly* between infrequent synchronizations. This decay prevents a stable shared trajectory, leading to our central question:

> *Can a distributed adaptive optimizer with $\beta$'s suited for infrequent communication*
> *close the performance gap with DDP while providing convergence guarantees?*

We propose MT-DAO, which brings multi-momentum optimizers (Lucas et al., 2019; Pagliardini et al., 2025) to the distributed, infrequent-communication regime. MT-DAO resolves the mismatch by using slow-moving momenta (e.g., $\beta \approx 0.999$) to preserve trajectory information across synchronizations while remaining responsive via a fast momentum. In its simplest quasi-hyperbolic form (Ma & Yarats, 2019), MT-DAO uses the current gradient as the fast momentum, adds no memory or communication overhead, and requires only one additional hyperparameter. Crucially, unlike methods that use a

---

[†]`aai30@cam.ac.uk`; [*]Equal contributions; [1]University of Cambridge; [2]Flower Labs; [3]Institute of Science and Technology Austria; [4]Mohamed bin Zayed University of Artificial Intelligence

momentum-based outer optimizer (Reddi et al., 2021; Douillard et al., 2023) at synchronization boundaries, `MT-DAO` needs **no extra memory buffers** or multiple outer hyperparameters.

Empirically, slow momentum acts as a regularizer, improving update alignment by increasing cosine similarity between worker pseudo-gradients (Reddi et al., 2021). This stability lets `MT-DAO` improve perplexity over low-communication baselines. Furthermore, `MT-DAO` matches or exceeds its `DDP` analogue at larger scales, **closing the perplexity gap** for models up to 720M parameters. We validate these findings across six inner optimizers (`ADOPT`, `Adam`, `Muon`, `AdEMAMix`, `AggMo`, and `QHM`) at scales up to 1.3B, and show that the benefits extend to streaming communication regimes (Appendix H.4), varying numbers of workers and batch sizes (Appendix H.3), and multiple momentum counts (Appendix H.2).

> **Contributions :**
>
> 1. **A Provably Convergent Multi-Timescale Framework.** We introduce `MT-DAO`, the first framework to integrate multi-momentum strategies into distributed settings, with convergence guarantees for heterogeneous momentum timescales and synchronization frequencies.
>
> 2. **Closing the Performance Gap Efficiently.** `MT-DAO` matches synchronous `DDP`, outperforming baselines in perplexity, reducing wall-clock time by 6–27%; at 720M it reaches a target perplexity in 24% fewer steps and 35% less time than a DDP baseline. We demonstrate these gains across six inner optimizers and scales up to 1.3B, including downstream task evaluation.
>
> 3. **Noise Suppression and Information Retention.** `MT-DAO`'s slow momentum preserves mutual information across rounds and reduces inter-worker momentum variance.
>
> 4. **Resilience to Infrequent Communication.** `MT-DAO` lowers the rate of change of parameters and momenta, improving tolerance to low communication frequencies.
>
> 5. **Alignment of Worker Trajectories.** `MT-DAO` increases cosine similarity of local worker update trajectories, which reduces worker drift and aligns the overall model update.

## 2 MULTI-TIMESCALE DISTRIBUTED ADAPTIVE OPTIMIZERS (MT-DAO)

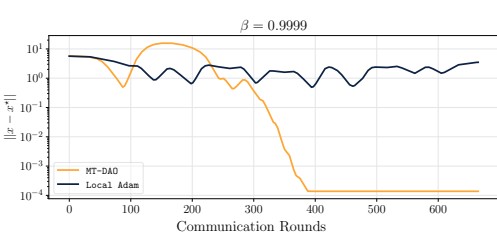
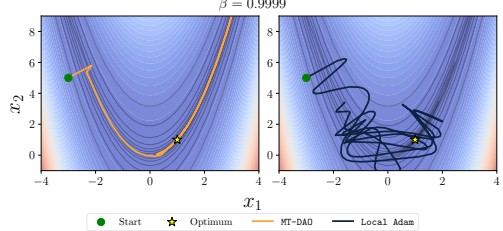

(a) Distance to optimum vs. steps.     (b) Contour plot of trajectories.

Figure 1: To highlight the stability benefit of `MT-DAO`, we illustrate its performance on a toy non-convex problem. Crucially, under a high momentum decay of $\beta = 0.9999$, prior stateful methods like `Local Adam` (Cheng & Glasgow, 2025) become unstable and fail to converge, whereas `MT-DAO` maintains its rapid and stable convergence. We optimize the non-convex Rosenbrock function $f(x_1, x_2) = (1 - x_1)^2 + 100(x_2 - x_1^2)^2$ with $M = 256$ workers and IID Gaussian noise ($\sigma = 2$).

We characterize the conflict between optimizer momentum timescales and communication intervals. Let $M$ workers perform $K$ local updates per round to minimize $f(x) := \frac{1}{M} \sum_{m=1}^{M} f_m(x)$, where $f_m(x) = \mathbb{E}_{\xi \sim \mathcal{D}_m}[F_m(x; \xi)]$. Performance degradation in this regime stems from a mismatch between the optimizer's memory and the communication period.

### 2.1 TIMESCALE MISMATCH

The first momentum in adaptive optimizers is an Exponential Moving Average (EMA): $u_t = \beta u_{t-1} + (1 - \beta)g_t$, with effective memory quantified by half-life $\tau_{0.5}(\beta) = \frac{\ln 0.5}{\ln \beta}$ (Pagliardini et al., 2025).

A typical $\beta_1 = 0.9$ yields $\tau_{0.5} \approx 6.6$ steps, suitable for frequent communication. A conflict arises when $K \gg \tau_{0.5}$, common in communication-efficient training ($K \in [32, 512]$). Unrolling over $K$ local steps gives $u_{t+K} = \beta^K u_t + (1-\beta) \sum_{k=0}^{K-1} \beta^k g_{t+K-k}$. The influence of the global state $u_t$ decays as $\beta^K$ ($\approx 0.03$ for $\beta_1=0.9$, $K=32$), leaving workers reliant on noisy local gradients. For example, if noise is independent across workers, the variance of the final local momentum is $\text{Var}(u_{t+K}) = \frac{1-\beta}{1+\beta}(1-\beta^{2K})\sigma_m^2$ (see Appendix F), with $\sigma_m^2$ being the gradient variance of worker $m$. As $\beta \to 1$ the factor $\frac{1-\beta}{1+\beta}$ suppresses variance. For $\beta \to 0$, variance approaches $\sigma_m^2$, exposing local updates to noise-induced instability. Importantly, while increasing the batch size $B$ reduces $\sigma_m^2$ linearly, $\beta$ controls variance exponentially through the drift term $(1-\beta^{2K})$. This means that in infrequent-communication regimes, precise control of $\beta$ is far more important than in DDP settings, where the exponential time-decay is absent (see Appendix F.1 for the full derivation and Appendix H.3 for empirical validation).

An alternative interpretation of this memory decay is offered by information theory, which quantifies the preserved signal between the initial global momentum $U_t$ and the final local momentum $U_{t+K}$ via their mutual information, $I(U_{t+K}; U_t)$. By modeling the local updates as a linear process $U_{t+K} = \beta^K U_t + L$, where $L$ is the accumulated local gradient noise, a closed-form expression can be derived when assuming Gaussian distributions for the states and noise with covariances $\Sigma_{U_t}$ and $\Sigma_L$ respectively. The mutual information is $I(U_{t+K}; U_t) = \frac{1}{2} \log \det(I + \beta^{2K} \Sigma_{U_t} \Sigma_L^{-1})$ (see Appendix F). As $\beta^K \to 0$, mutual information vanishes, implying statistical independence with respect to the momentum at the start of the interval. As $\beta^K \to 1.0$, the initial signal is preserved.

## 2.2 THE CHALLENGE OF HIGH-$\beta$ OPTIMIZERS

Although both arguments above encourage the use of large $\beta$ values as a solution to this timescale mismatch problem, previous work has shown that high-momentum optimizers are often unfeasible in practice (Lucas et al., 2019). Without modification, they are insufficiently responsive to changes in the loss landscape and are prone to oscillations (see the top of Fig. 2), yielding subpar performance.

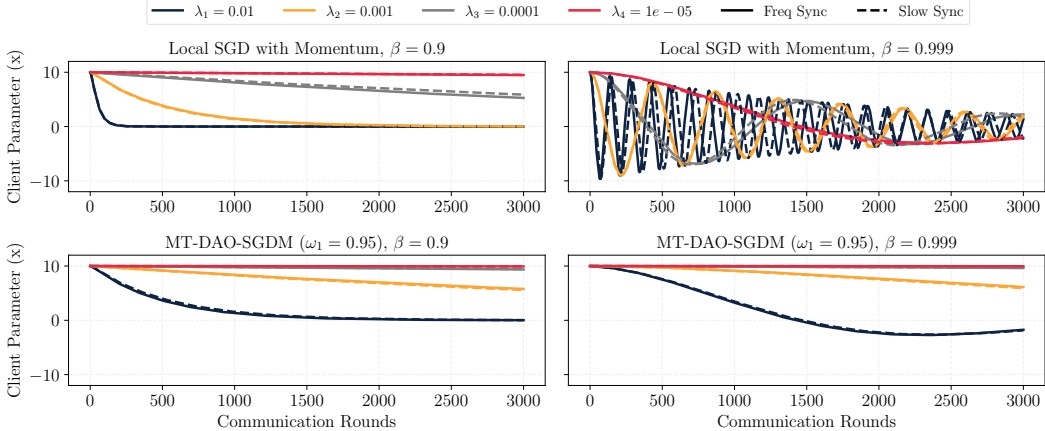

Figure 2: Comparison of Local SGDM with standard momentum **(top)** and MT−DAO−SGDM ($N = 1$ momentum, $\omega_1 = 0.95$) **(bottom)** for the function $f(x; \lambda) = \frac{1}{2}\lambda x^2$ with $x \in \mathbb{R}$ for various parameters controlling the rate of change $\lambda$ and and sync frequencies (frequent: solid, infrequent/slow: dashed). For $\beta = 0.9$, Local SGD with momentum can quickly find the sole global optimum, which does not hold for non-convex functions, however, MT−DAO−SGDM converges despite the slow momentum dampening it. While both optimizers are stable at $\beta = 0.9$, at high momentum ($\beta = 0.999$) Local SGD with standard momentum becomes unstable for high $\lambda$ while MT−DAO−SGDM remains stable.

This instability motivates using multi-momentum methods (Lucas et al., 2019; Ma & Yarats, 2019; Pagliardini et al., 2025). Such methods compose the optimizer update as a linear combination of slow and fast-moving first momenta, or the gradient in the case of Quasi-hyperbolic methods (Ma & Yarats, 2019). This avoids common pitfalls of high-momentum methods by responding to changes in the loss

landscape via the fast momentum/gradient. Recent works (Pagliardini et al., 2025; Semenov et al., 2025) have shown that such optimizers can provide SOTA results, outperforming popular optimizers such as Adam (Loshchilov & Hutter, 2019), Muon (Jordan et al., 2024), and Dion (Ahn et al., 2025).

## 2.3 THE MT-DAO METHOD AND ALGORITHM

---

**Algorithm 1** MT-DAO-Adam, local bias correction omitted to save space.

---

**Require: Model tensors, hyper-parameters**
1:   $x_0 \in \mathbb{R}^d, \{\bar{u}^j_{-1}\}^N_{j=1} \in (\mathbb{R}^d)^N, \bar{v}_{-1} \in \mathbb{R}^d$ — initial params, $N$ first momenta, second momentum
2:   $\{\beta_{1,j}\}^N_{j=1}, \beta_2 \in [0,1)$ — decay rates for each momentum state
3:   $\{\omega_j\}^N_{j=1} \in [0,1]$ — convex combination coefficients for first momenta, $\sum^N_{j=1} w_j \le 1.0$
4:   $\rho \in \mathbb{R}_+, \{\eta_t\}^{T-1}_{t=0}$ — clipping radius, learning-rate schedule
5:   $T, M \in \mathbb{N}_+$ — total optimization steps and number of workers
6:   $K_x, \{K_j\}^N_{j=1}, K_v \in (\mathbb{N}_+)^{N+2}$ — communication periods for parameters and states
7:   OuterOpt $: \mathbb{R}^d \to \mathbb{R}^d$ — update params using an outer optimizer, averaging by default

**Ensure:** $x_T, \{u^j_{T-1}\}^N_{j=1}, v_{T-1}$
8:   **for each worker** $m$: initialize $x^m_0, \{u^{j,m}_{-1}\}, v^m_{-1}$
9:   **for** $t = 0, \dots, T-1$ **do**
10:      **for all** workers $m = 0, \dots, M-1$ **in parallel do**
11:         $\hat{g}^m_t \leftarrow \textbf{clip}(\nabla F(x^m_t; \xi^m_t), \rho)$                                                   *clipped stochastic gradient*
12:         **for** $j = 1$ **to** $N$ **do**                                                                                          *update N first momenta*
13:            $u^{j,m}_t \leftarrow \beta_{1,j}\bar{u}^j_{t-1} + (1-\beta_{1,j})\hat{g}^m_t$
14:            $\bar{u}^j_{t-1} \leftarrow$ **if** $(t \bmod K_j = 0)$ **then** $\mathbb{E}_m[u^{j,m}_{t-1}]$ **else** $u^{j,m}_{t-1}$     *sync $u^j$ every $K_j$ steps*
15:         $v^m_t \leftarrow \beta_2\bar{v}_{t-1} + (1-\beta_2)(g^m_t)^2$
16:         $\bar{v}_{t-1} \leftarrow$ **if** $(t \bmod K_v = 0)$ **then** $\mathbb{E}_m[v^m_{t-1}]$ **else** $v^m_{t-1}$               *sync $v$ every $K_v$ steps*
17:         $\Delta^m_t \leftarrow \frac{1}{\sqrt{v^m_t}+\epsilon}\left[(1-\sum^N_{j=1}\omega_j)\hat{g}^m_t + \sum^N_{j=1}\omega_j u^{j,m}_t\right]$     *form combined update direction*
18:         $x^m_{t+1} \leftarrow \bar{x}_t - \eta_t\Delta^m_t$
19:         $\bar{x}_t \leftarrow$ **if** $(t \bmod K_x = 0)$ **then** OuterOpt$(\mathbb{E}_m[x^m_t])$ **else** $x^m_t$                  *sync $x$ every $K_x$ steps*

---

Algorithm 1 formalizes MT-DAO for Adam (variants: ADOPT in Algorithm 2, SGDM in Algorithm 3). It uses $N$ first-order momenta $\{u^j\}$ and one second-order momentum $v$, combined via weights $\{\omega_j\}$ (purple), with the current gradient receiving weight $1 - \sum^N_{j=1}\omega_j$—implementing the QHM structure. OuterOpt represents any aggregator (e.g., FedAvg (McMahan et al., 2017), Nesterov (Huo et al., 2020), FedOPT (Reddi et al., 2021)); we default to averaging per prior analyses (Cheng & Glasgow, 2025; Iacob et al., 2025). MT-DAO-Adam reduces communication costs by $(\frac{1}{K_x} + \sum^N_{j=1}\frac{1}{K_j} + \frac{1}{K_v})^{-1}$ over DDP.

This generalized framework recovers previous distributed adaptive optimizers (Stich, 2019; Douillard et al., 2023; Cheng & Glasgow, 2025; Iacob et al., 2025). It also introduces **the first-ever formulations for provably convergent distributed variants of multi-momentum optimizers** (Lucas et al., 2019; Ma & Yarats, 2019; Pagliardini et al., 2025). Figure 2 (bottom) shows an example of MT-DAO-SGDM converging for both high and low $\beta_1$ with a quasi-hyperbolic formulation while the Local SGD with momentum averaging method fails for high $\beta_1$. To highlight the stability of MT-DAO-Adam, Figure 1 illustrates its convergence on a common toy non-convex problem (Pagliardini et al., 2025) under high momentum ($\beta = 0.9999$), a setting where prior provably convergent methods like Local Adam (Cheng & Glasgow, 2025) become unstable and do not reach the optimum. Beyond Adam and ADOPT, we empirically validate that MT-DAO is effective with Muon (Appendix H.7), AdEMAMix, and AggMo (Appendix H.2), and that it composes with streaming communication (Appendix H.4). We recommend the robust default $\beta_1=0.999, \omega_1=0.98$ for the quasi-hyperbolic variant, which transfers from 16M to 1.3B without re-tuning (Section 5.5).

## 3 CONVERGENCE GUARANTEES FOR MT-DAO

This section provides a theoretical convergence analysis for the proposed MT-DAO approach using the SGDM optimizer. The analysis, detailed in Appendix D, relies on the following standard assumptions.

**Assumption 1** (Lower bound and smoothness). *The overall loss function $f \colon \mathbb{R}^d \to \mathbb{R}$ is lower bounded by some $f^* \in \mathbb{R}$ and all local loss functions $f_m$ are $L$-smooth:*

$$\|\nabla f_m(x) - \nabla f_m(y)\| \le L\|x - y\|, \quad \text{for any } x, y \in \mathbb{R}^d.$$

**Assumption 2** (Unbiased noise with bounded stochastic variance). *The stochastic gradient $g^m$ of local loss function $f_m$ computed by machine $m$ is unbiased and the noise has bounded variance:*

$$\mathbb{E}[g^m] = \nabla f_m(x), \quad \mathbb{E}[\|g^m - \nabla f_m(x)\|^2] \le \sigma^2, \quad \text{for any } x \in \mathbb{R}^d.$$

**Assumption 3** (Bounded heterogeneity). *For any $x \in \mathbb{R}^d$, the heterogeneity is bounded by*

$$\tfrac{1}{M} \sum_{m=1}^{M} \|\nabla f_m(x)\|^2 \le G^2 + B^2 \|\nabla f(x)\|^2.$$

These are standard assumptions in smooth non-convex optimization (Yu et al., 2019; Karimireddy et al., 2020b; Wang et al., 2021; Yuan et al., 2022), covering homogeneous data as a special case ($G^2 = 0, B^2 = 1$). For analytical tractability, we model periodic synchronization every $K$ steps as a probabilistic event. Model parameters are averaged with probability $p_x = 1/K_x$, the $j$-th momentum is averaged with probability $p_j = 1/K_j$. The gradient is treated as a momentum with $\beta = 0$.

**Theorem 1.** *Let Assumptions 1, 2 and 3 hold. Then, choosing the step size $\eta = \min(\eta_0, \frac{1}{\sqrt{T}})$ where $\eta_0 \stackrel{\text{def}}{=} 1/(4L \max(\beta_\omega, 6\sqrt{\psi \max(1, B^2 - 1)}))$ with constants*

$$\beta_\omega \stackrel{\text{def}}{=} \sum_{j=1}^{N} \frac{\omega_j \beta_j}{1 - \beta_j}, \quad \text{and} \quad \psi \stackrel{\text{def}}{=} \frac{4(1 - p_x)}{p_x^2} \sum_{j=1}^{N} \omega_j \frac{(1 - \beta_j)(1 - p_j)}{1 - (1 - p_j)\beta_j} \tag{1}$$

*the average iterates $x_t = \mathbb{E}_m[x_t^m]$ of* `MT-DAO-SGDM` *converge with the following rate:*

$$\tfrac{1}{T} \sum_{t=0}^{T-1} \mathbb{E}\|\nabla f(x_t)\|^2 \le \frac{4}{\sqrt{T}} \left( f(x_0) - f^* + \frac{L\sigma^2}{2M} \right) + \mathcal{O}\left( \frac{1 + \beta_\omega^2 + \psi}{T} \right). \tag{2}$$

The derived bound in (2) achieves the optimal $\mathcal{O}(1/\sqrt{T})$ asymptotic rate for smooth non-convex stochastic optimization (Arjevani et al., 2023). Distributed factors, such as client drift and data heterogeneity, are contained within the step-size constraint and the higher-order $\mathcal{O}(1/T)$ term, thus not affecting the asymptotic rate. The step size $\eta$ is constrained by $\beta_\omega$ and $\psi$. The dependence $\psi = \mathcal{O}(1/p_x^2)$ shows that model synchronization frequency $p_x$ is critical. The impact of momentum synchronization is nuanced: reducing a momentum's sync frequency $p_j$ increases its contribution to $\psi$, but this is modulated by its decay rate $\beta_j$. This implies "slower" momenta (larger $\beta_j$) are more robust to infrequent synchronization. **This analysis reveals a trade-off: large $\beta_j$ values constrain the step size via $\beta_\omega$ but reduce the communication penalty in $\psi$.** Furthermore, synchronizing only the model always (i.e., $p_x = 1, p_j = 0$) is algorithmically equivalent to synchronizing only the momenta always (i.e., $p_x = 0, p_j = 1$). In the boundary case where only model parameters are synced ($p_x = 1, p_j = 0$), $\psi = 0$ and the rate recovers that of mini-batch SGD (Liu et al., 2020).

## 4 EXPERIMENTAL DESIGN

Building on our analysis, our experimental design answers the following research questions:

**RQ1** Does `MT-DAO` reduce momentum noise and preserve mutual information, as predicted?
**RQ2** Does `MT-DAO` better preserve task performance when decreasing communication frequency?
**RQ3** How does `MT-DAO` perform against `DDP` and prior communication-efficient optimizers?
**RQ4** How does slow momentum affect local optimization trajectories between synchronizations?
**RQ5** How does `MT-DAO` impact downstream task performance vs baselines?

### 4.1 SETUP

**Models and Data.** We use peri-norm (Kim et al., 2025) `GPT`-style transformer models of 16M, 125M, 360M, and 720M parameters (Table 2). The 16M model is used for hyperparameter sweeps and qualitative investigations, while the 125M and 720M models are used for scaling experiments and baseline comparisons. We additionally train a 1.3B parameter model on 80B tokens using `Adam`

to demonstrate optimizer-agnostic scalability (Section 5.5). All models are trained with a sequence length of 2048 on the `SmolLM2` mixture (Allal et al., 2025). We evaluate all models using validation perplexity on a held-out $10\%$ portion of the training mixture, and the 1.3B model additionally on downstream zero-shot and few-shot benchmarks. For further details, please see Appendix B.

**Optimizers and Tuning Methodology.** We use the `ADOPT` optimizer, a variant of `Adam` whose convergence rate is independent of the second-momentum decay rate $\beta_2$ and preserves performance (Taniguchi et al., 2024); we fix $\beta_2 = 0.9999$ to isolate the first momentum dynamics (governed by $\beta_1$ defaulting to 0.9 and $\omega$). For `Adam` we use $(\beta_1, \beta_2) = (0.9, 0.999)$ by default as recommended by Semenov et al. (2025). We use the `CompleteP` parameterization for one-shot transfer of the learning rate (LR) from small to large models (Dey et al., 2025). For each combination of convex coefficients ($\omega$'s) and momentum decays ($\beta$'s), we tune the learning rate on the 16M model and transfer the optimal hyperparameters to larger models directly without ever re-tuning ($\omega$'s) and ($\beta$'s). To establish strong `DDP` baselines, we tune $\omega, \beta_1$ parameters in the `DDP` setting and reuse them for `MT-DAO`. For complete details see Appendix B.2. We always use quasi-hyperbolic `MT-DAO` ($N = 1$) which does not require additional memory and reduces comms costs by $(\frac{1}{K_x} + \frac{1}{K_1} + \frac{1}{K_v})^{-1}$ over `DDP`.

**Baselines.** We compare `MT-DAO` against: the base optimizer (`ADOPT`/`Adam`) with `DDP` and `DDP` analogues to `MT-DAO` such as Quasi-hyperbolic Momentum (`QHM`) (Ma & Yarats, 2019). For communication-efficient baselines we use the provably convergent and stateful `Local Adam` (Cheng & Glasgow, 2025) approach. We also compare against using Nesterov momentum as the outer optimizer (Charles et al., 2025). We evaluate ML performance for communication-efficient methods under the same, fixed synchronization frequency. Unless otherwise stated we use $K = K_x = K_1 = K_v = 32$ steps, based on prior work finding a practical balance of performance efficiency (Charles et al., 2025). We split the dataset in an `IID` fashion across 4 workers using 1 H100 per worker.

**Other Metrics** We analyze flattened models/momenta $s_t \in \mathbb{R}^d$ using several metrics. The **relative change** over $K$ steps is measured as $\|s_{t+K} - s_t\|_2/\|s_t\|_2$. To quantify the dispersion among $M$ worker vectors, we compute the **cross-worker variance**, defined as $\frac{1}{M} \sum_{m=1}^{M} \|s_m - \bar{s}\|_2^2$. The statistical dependency between two random vectors at different timesteps is captured by their **mutual information**, $I(U_{t+K}; U_t)$. Finally, we measure alignment between vectors using **cosine similarity**.

# 5 EVALUATION

This section empirically validates `MT-DAO`, showing its slow momentum preserves information and aligns workers (Sections 5.1 and 5.4), which improves stability under infrequent communication (Section 5.2) and allows it to close the performance gap with `DDP` at scale (Section 5.3). We further demonstrate that these gains translate to downstream task performance at the 1.3B scale using `Adam` (Section 5.5). In total, our evaluation spans six inner optimizers, model scales from 16M to 1.3B, and 235 additional experiments detailed in the appendix.

## 5.1 `MT-DAO` REDUCES MOMENTUM NOISE AND PRESERVES MUTUAL INFORMATION (**RQ1**)

We now empirically validate the motivation of `MT-DAO`. Our results in Fig. 3 demonstrate that the slow-momentum of `MT-DAO` both preserves information about the global optimization direction across communication rounds and suppresses the variance induced by local updates.

> **Slow Momentum Is Preserved:** The slow momentum in `MT-DAO` preserves its direction across communication rounds. This directional memory also reduces the influence of local gradient noise, leading to lower momentum variance across workers and **a more stable optimization path**.

## 5.2 `MT-DAO` IS RESILIENT TO INFREQUENT COMMUNICATION (**RQ2**)

We now investigate if `MT-DAO` provides greater resilience against infrequent synchronization, as predicted by our analysis in Section 3 showing that reducing the communication frequency of momenta with higher $\beta$ has a diminished impact on the step size.

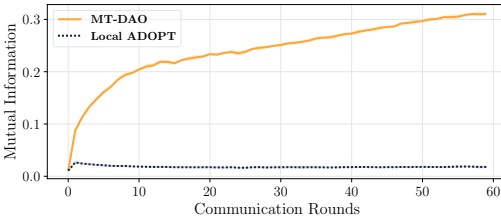 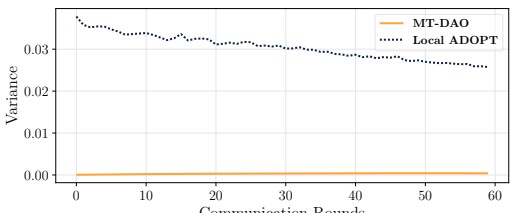

Figure 3: A comparison of MT-DAO ($\beta_1 = 0.999$) versus a Local ADOPT baseline ($\beta_1 = 0.95$) with a communication frequency of $K = 32$. For each communication round, we plot metrics computed between the momentum at the start ($t$) and end ($t + K$) of the round. MT-DAO's slow momentum **preserves mutual information**, $I(U_t; U_{t+K})$, across rounds while the baseline's momentum decays losing the global optimization direction (left). Furthermore, MT-DAO **reduces inter-worker momentum variance**, $\text{Var}(u_{t+K})$, indicating greater stability against local noise (right).

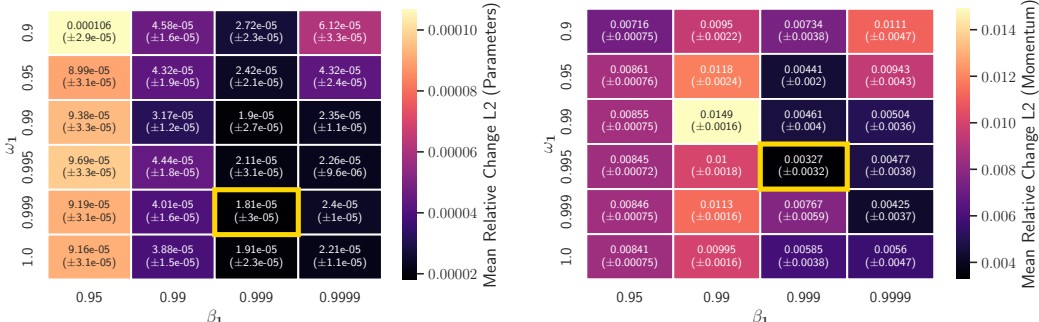

Figure 4: Mean relative L2 change and standard deviation across communication rounds of (left) model parameters and (right) the first momentum state, as a function of momentum decay ($\beta_1$) and weight ($\omega_1$). In both cases, MT-DAO shows a significantly **reduced** relative rate of change with high ($\beta_1, \omega_1$) (minimum in gold), which **reduces worker drift and thus makes parameter averaging more effective**. Each point on the grid corresponds to a configuration evaluated with its own independently tuned learning rate. Local ADOPT corresponds to ($\beta_1 = 0.95, \omega_1 = 1.0$).

Table 1: Demonstration of how parameter synchronization period ($K_x$) affects final perplexity for two MT-DAO configurations with momentum periods $K_1 = K_v = 16$ for our 16M models. Values show the percentage increase in validation perplexity over the $K_x = 16$ baseline. Higher $\beta$ leads to **less performance degradation as $K_x$ increases**.

| $K_x$ | 16 | 32 | 64 | 128 | 256 | 512 | 1024 |
|---|---|---|---|---|---|---|---|
| $\beta_1 = 0.99$ | (37.72) | +1.7% | +3.0% | +3.9% | +5.1% | +5.6% | +6.2% |
| $\beta_1 = 0.995$ | (37.65) | +1.0% | +1.6% | +3.2% | +2.8% | +3.4% | +3.7% |

Table 1 shows that a MT-DAO configuration with higher $\beta_1$ suffers less degradation as $K_x$ increases. This resilience is explained by the reduced rate of change in the model parameters, quantified in Figure 4. High ($\beta_1, \omega_1$) values reduce the local model change $\mathbb{E}_m[\|x^m_{t+K_x} - x_t\|_2]$, bounding inter-worker variance and mitigating drift. Workers thus compute gradients on models closer to the global mean, improving convergence robustness (Li et al., 2020).

> **Slow Momentum as Anchor:** Long-term momentum (high $\beta_1$ and $\omega$) reduces the rate of change of parameters and optimizer states. This stability ensures worker models diverge less prior to synchronization, **which reduces the performance impact of infrequent synchronization**.

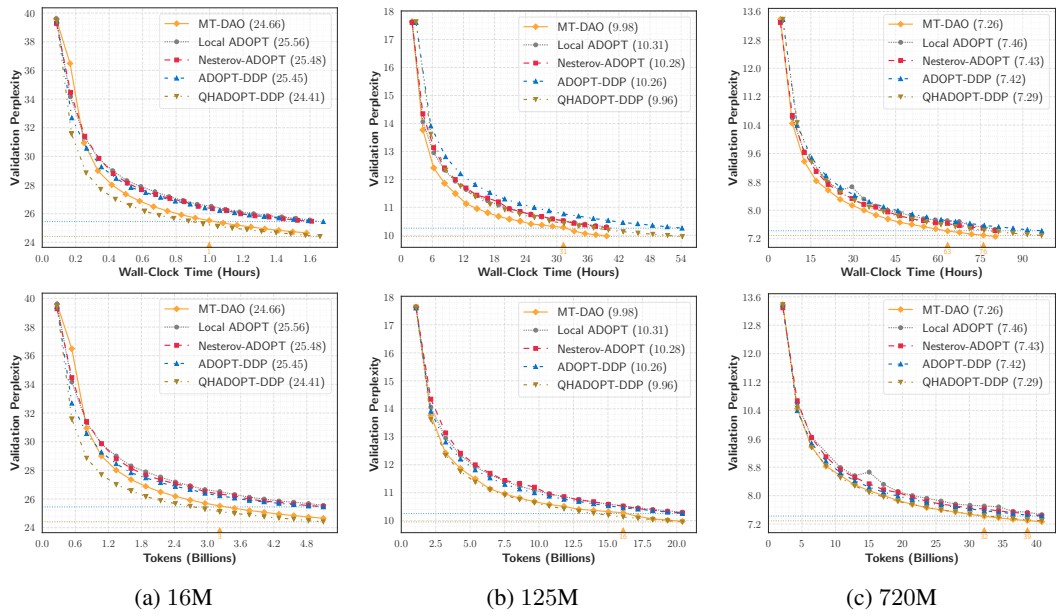

(a) 16M          (b) 125M          (c) 720M

Figure 5: Validation perplexity versus wall-clock time and training tokens for `MT-DAO` ($\beta_1 = 0.999, \omega_1 = 0.98$) and baselines on models of increasing size. Horizontal lines denote the two `DDP` baselines (`ADOPT-DDP` and `QHADOPT-DDP`). For each non-`DDP` method, a colored marker on the x-axis marks the point at which its curve attains a lower/equal perplexity to a `DDP` variant.

## 5.3 `MT-DAO` OUTPERFORMS PRIOR LOW-COMMS OPTIMIZERS AND MATCHES DDP (**RQ3**)

We evaluate `MT-DAO` on 16M, 125M, and 720M parameter language models against other baselines. We report validation perplexity as a function of both training tokens and wall-clock time. Timings are measured on 4 cloud H100s connected via 50–100 Gbit/s Ethernet, including constant implementation overheads, and accounting for communication–computation overlap in `DDP`. These measurements are specific to this hardware; Appendix E provides a bandwidth model that compares communication-efficient methods to `DDP` across a wider range of interconnects. When reporting time-to-target perplexity, we give improvements in both wall-clock time and training tokens.

Across all scales, `MT-DAO` consistently improves over `ADOPT-DDP` and `Local ADOPT` in both tokens and time, closing the gap to synchronous training and reducing end-to-end wall clock by **6–27**%. At 720M, relative to single-momentum `DDP`, `MT-DAO` reaches the same perplexity in **24**% fewer tokens and **35**% less time. Relative to `QHADOPT-DDP`, `MT-DAO` trails at 16M, matches at 125M, and at 720M reaches the `QHADOPT-DDP` target perplexity about 8% faster in wall-clock and $\approx 5\%$ fewer tokens. The additional improvements in time are due to `MT-DAO` communicating **10**× less than `DDP`. The outer `Nesterov` baseline performs better than `Local ADOPT` in our setting yet remains below `MT-DAO` and `DDP`; matching the findings of Charles et al. (2025, Table 4). Mechanistically, `Nesterov` coalesces per-round gradients via an *outer* momentum, whereas `MT-DAO` implements a finer-grained *inner* multi-timescale modification. We note that `MT-DAO` ($N = 1$) does not require the additional momentum buffer of Nesterov and has only one additional hyperparameter to tune instead of two.

> **Improved Performance and Efficiency at Scale:** `MT-DAO` improves performance w.r.t all baselines across model scales, **closing the performance gap to `DDP`.** These gains are not specific to `ADOPT`: we confirm similar trends with `Adam` up to 1.3B (Section 5.5), `Muon` (Appendix H.7), `AdEMAMix`, and `AggMo` (Appendix H.2). We further show robustness across batch sizes and worker counts (Appendix H.3), in the streaming regime (Appendix H.4), and under varying numbers of momenta (Appendix H.2).

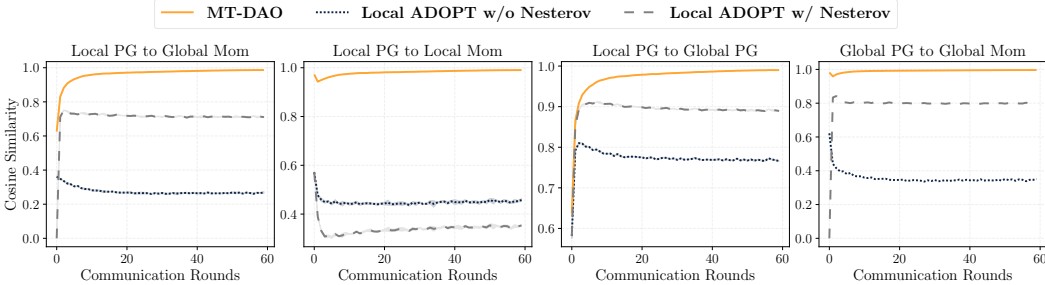

Figure 6: A comparison of update vector alignments for MT-DAO ($\beta_1 = 0.999, \omega_1 = 0.98$) versus Local ADOPT ($\beta_1 = 0.95$), and Local ADOPT ($\beta_1 = 0.95$) with Nesterov. Cosine similarity is measured between: (1) the local pseudo-gradient and global momentum, (2) the local pseudo-gradient and the local momentum, (3) the local and global pseudo-gradients, (4) the local and global momentum. Pseudo-gradient and momentum have been abbreviated as *PG* and *Mom.*

## 5.4 MT-DAO ALIGNS WORKER UPDATE TRAJECTORIES(**RQ4**)

Having established the performance benefits of MT-DAO, we now investigate the underlying mechanism. We hypothesize that the slow momentum reduces worker drift by keeping the optimization trajectories of individual workers aligned with the global optimization direction. To validate this, we measure the cosine similarity between key optimization vectors. We define the per-round local update as the "pseudo-gradient" ($\Delta^m = x_{t+K}^m - x_t^m$), and the global pseudo-gradient as the average of local ones (Reddi et al., 2021). To provide a comprehensive comparison, we define the "global momentum" for each method: for MT-DAO and Local ADOPT, it is the average of worker momenta at the end of a round, while for the Nesterov variant, it is the state of the outer Nesterov momentum.

The results in Fig. 6 show that MT-DAO achieves near-perfect alignment (cosine similarity $> 0.95$) across all four metrics. This indicates that: (1) each worker's update is consistent with its own momentum history (Local PG to Local Mom), (2) workers are in strong agreement with each other (Local PG to Global PG), and (3, 4) both local and global updates are aligned with the long-term global trajectory (Local/Global PG to Global Mom). This demonstrates that the slow momentum acts as a regularizer, ensuring all workers maintain a stable and shared optimization path.

In contrast, the Nesterov outer optimizer presents mixed results. As an EMA of global pseudo-gradients, it is better aligned to the global pseudo-gradient than the Local ADOPT momentum and it improves the alignment between the local and global pseudo-gradients compared to standard Local ADOPT. However, it never reaches the degree of alignment of MT-DAO in any metric.

> **Slow Momentum as Regularizer:** MT-DAO's slow momentum acts as a regularizer for each worker, ensuring that **local updates remain aligned with their history and the global trajectory.**

## 5.5 MT-DAO IMPROVES DOWNSTREAM TASK PERFORMANCE (**RQ5**)

To validate that perplexity gains translate to practical utility, we evaluate downstream zero-shot and few-shot accuracy at the 1.3B parameter scale using Adam (Fig. 7). MT-DAO demonstrates consistent superiority over AdamW-DDP in aggregate performance, excelling on the most challenging reasoning tasks (ARC-CHALLENGE, HELLASWAG). Compared to QHAdamW-DDP, MT-DAO remains competitive early in training and progressively closes the gap as training continues, mirroring the perplexity trends. Using 4 machines with 8 H100 GPUs each connected via 100 Gbit inter-node links, MT-DAO achieves target accuracies *earlier in time* than its DDP counterparts (Fig. 12).

## 6 RELATED WORK

Standard Distributed Data Parallelism's (DDP) per-step synchronization creates a communication bottleneck (Sergeev & Balso, 2018). This is mitigated by two orthogonal strategies: payload compression and infrequent synchronization. Compression shrinks transmissions via quantization (Alistarh et al.,

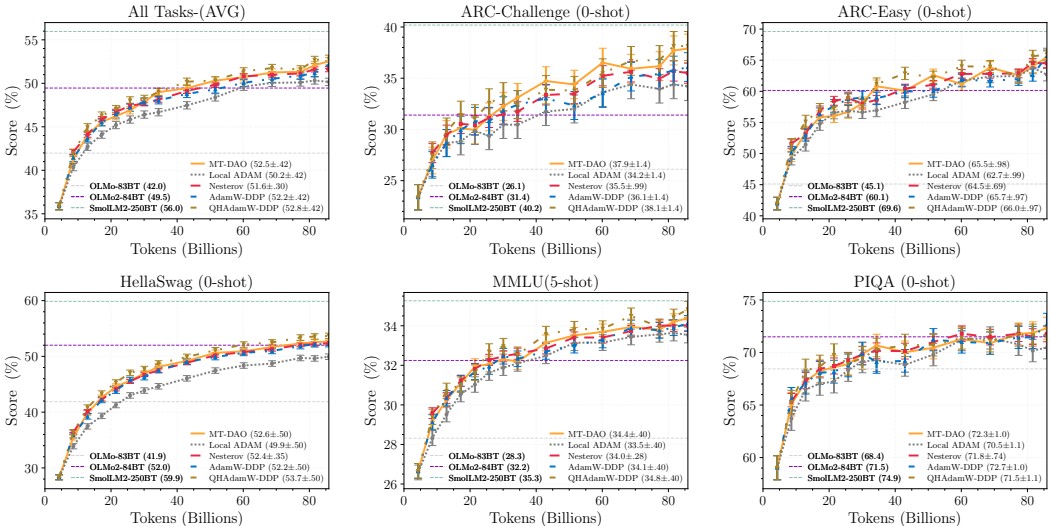

Figure 7: **Downstream task accuracy versus training tokens** at 1.3B scale on ARC-CHALLENGE, ARC-EASY, HELLASWAG, MMLU, and PIQA. `MT-DAO` **exceeds** `AdamW-DDP` on the aggregate and on reasoning-heavy tasks; **closes the gap** to `QHAdamW-DDP` with training. Time-normalized results: Figure 12.

2017), sparsification (Lin et al., 2018b), mixes thereof (Wang et al., 2023b), low-rank updates (Robert et al., 2025), or communicating select momentum components (Peng et al., 2024). Our work advances infrequent synchronization (Stich, 2019; McMahan et al., 2017) which allows local updates between communications and is complementary to compression.

Adapting stateful optimizers like `Adam` to infrequent synchronization is not straightforward. `Local Adam` (Cheng & Glasgow, 2025) provided the first convergence proofs at the cost of synchronizing all optimizer states. Douillard et al. (2023); Charles et al. (2025) showed that a Nesterov-based outer optimizer improves performance. Recently Iacob et al. (2025) improved the communication efficiency of `Local Adam` by decoupling parameter and momentum sync frequencies. However, these methods use single-timescale optimizers with small $\beta_1$ values ill-suited to low communication frequencies due to momentum decay. While naively increasing momentum often harms task performance, recent optimizers that track gradients across multiple timescales have shown significant benefits. `QHM` (Ma & Yarats, 2019) decouples momentum decay from gradient weight, while `AggMo` (Lucas et al., 2019) averages multiple velocity vectors for stability. Building on this, `AdEMAMix` (Pagliardini et al., 2025) mixes fast and slow momenta to accelerate convergence, demonstrating that slow momentum acts as memory, reducing forgetting in LLMs. More recently, `Muon` (Jordan et al., 2024) uses Newton-Schulz preconditioning for update orthogonalization. We show that `MT-DAO` is compatible with all of these optimizers, providing empirical validation with `Adam`, `ADOPT`, `Muon`, `AdEMAMix`, and `AggMo` (Appendix H). For further related work see Appendix G.

## 7 CONCLUSION

A persistent challenge in distributed training has been the performance gap between fully-synchronous and communication-efficient optimizers. We identify the rapid decay of momentum as a key cause: standard momenta are temporally mismatched with the long intervals of infrequent communication. `MT-DAO` resolves this with a multi-timescale optimizer whose slow momentum persists across communication rounds. Our theory shows that higher $\beta$ reduces sensitivity to synchronization frequency, and our experiments demonstrate that this approach closes the performance gap with `DDP` by maintaining a stable, shared optimization trajectory. These findings establish momentum timescale management as a critical factor for performant distributed training, providing a practical path for communication-constrained, cross-datacenter, and wide-area settings.

ACKNOWLEDGMENTS

All costs for the computational resources used for this work were funded by Flower Labs, and the research conducted by a team of researchers from Flower Labs, The University of Cambridge, The Institute of Science and Technology of Austria, The University of Warwick, and Mohamed bin Zayed University of Artificial Intelligence. Support for university-based researchers came from a variety of sources, but in particular, the following funding organizations are acknowledged: the European Research Council (REDIAL), the Royal Academy of Engineering (DANTE), the Ministry of Education of Romania through the Credit and Scholarship Agency, and the European Union's Horizon 2020 research and innovation programme under the Marie Skłodowska-Curie grant agreement No 101034413.

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

# Appendix

# A Table of Contents

# A   LIMITATIONS

**Limitations.** While we have substantially expanded our empirical evaluation to models up to 1.3B parameters, multiple optimizers (`ADOPT`, `Adam`, `Muon`), and several ablation dimensions ($N$, batch size, workers, clipping, streaming), our validation remains limited to language-model pre-training. Extending `MT-DAO` to other modalities (e.g., vision, multimodal) and to non-IID data distributions beyond the homogeneous setting is left for future work. Additionally, while our convergence analysis covers `MT-DAO-SGDM`, extending theoretical guarantees to adaptive variants and to the streaming setting remains an open direction. Finally, our experiments focus on the memory- and communication-efficient $N{=}1$ variant, which we justify empirically (Appendix H.2); whether higher $N$ provides benefits in regimes with very long communication intervals ($K \gg 512$) warrants further investigation.

# B   EXPERIMENTAL DETAILS

Here we provide additional experimental details complementing those in Section 4.1, including: a) model architecture details and the model parameterization (Appendix B.1), b) our hyperparameter sweep procedure to select optimizer-specific settings (Appendix B.2), and c) the results of our tuning sweeps for `MT-DAO`.

## B.1   ARCHITECTURE DETAILS AND PARAMETRIZATION

Table 2: Model architecture and training hyperparameters. Architectural parameters include the number of transformer blocks (#Blocks), attention heads (#Heads), embedding dimension ($d_{\text{model}}$), vocabulary size ($|\mathcal{V}|$), and feedforward expansion ratio (Exp. Ratio). Key training parameters are the global batch size ($|\mathcal{B}_{\text{G}}|$) and the total number of training steps ($T$). All models use RoPE positional embeddings (Su et al., 2024), the `SiLU` activation function, norm-based gradient clipping with a bound of $\rho$, and are initialized with a typical (Semenov et al., 2025; Dey et al., 2025) $\sigma = 0.02$. For `Adam` we use the $\rho$ values recommended by Semenov et al. (2025). Sequence length is standard for models at these scales.

| Model Size | Blocks | $d_{\text{model}}$ | $|\mathcal{V}|$ | #Heads | Exp.∼Ratio | ROPE $\theta$ | ACT | Init $\sigma$ | $\rho_{\text{Adopt}}$ | $\rho_{\text{Adam}}$ | Seq Len | $|\mathcal{B}_{\text{G}}|$ | T |
|---|---|---|---|---|---|---|---|---|---|---|---|---|---|
| 16M | 4 | 256 | 50K | 4 | 4 | 10000 | SiLU | 0.02 | 1.0 | 1.0 | 2048 | 64 | 4608, 12288, 40960 |
| 125M | 12 | 768 | 50K | 12 | 4 | 10000 | SiLU | 0.02 | 1.0 | 0.5 | 2048 | 256 | 4608, 12288, 40960 |
| 360M | 24 | 1024 | 50K | 16 | 4 | 10000 | SiLU | 0.02 | 1.0 | 0.25 | 2048 | 256 | 12288, 40960 |
| 720M | 12 | 2048 | 50K | 16 | 4 | 10000 | SiLU | 0.02 | 1.0 | 0.1 | 2048 | 512 | 4608, 12288, 40960 |
| 1.3B | 24 | 2048 | 50K | 16 | 4 | 10000 | SiLU | 0.02 | 1.0 | 0.1 | 2048 | 1024 | 12288, 40960 |

Table 2 summarizes the architectural details of our models, which follow established practices for large language models at their respective scales. To improve training stability and final performance, we adopt two key modifications. First, following the recommendations of Kim et al. (2025), we use a Peri-LayerNorm transformer structure instead of pre-norm. Second, we use the `CompleteP` (Dey et al., 2025) parametrization with $\alpha = 1.0$, which enables the effective transfer of optimizer hyperparameters from a small model to its larger-scale counterparts in a one-shot manner. This property allows us to perform comprehensive hyperparameter sweeps on our smallest model size and reserve computationally expensive scaling experiments for direct comparisons against baselines.

We set batch sizes and training durations following recent best practices (Zhang et al., 2025). For the smallest model size, the initial batch size is determined using the noise-scale estimator for the critical batch size (McCandlish et al., 2018) and then doubled until the efficiency deviates from a linear trend by 20%. For our 125M and 720M models we follow the batch size recommendations from Semenov et al. (2025). Training durations are set as multiples of the compute-optimal token budget (Hoffmann et al., 2022): for the 16M model, we tune using $\approx 2\times$ this budget and run baseline comparisons at $\approx 16\times$; for the 125M model, we use $\approx 8\times$; and for the 720M model, we use $\approx 2.83\times$. We chose the 720M model size as a good balance between scale and computational efficiency following Semenov et al. (2025), with the 360M being chosen as an efficient middle-ground between 720M and 125M. For the 1.3B we use a standard batch size of 2M tokens.

All models are trained using the warmup-stable-decay (`WSD`) learning rate schedule (Hägele et al., 2024), with warmup and decay periods selected based on established recommendations (Zhang et al., 2025; Hägele et al., 2024; Allal et al., 2025; Semenov et al., 2025).

For all longer training runs and baseline comparisons, we use the industry-standard warmup of $T_{\text{WARM}} = 2048$ steps. We use a cooldown period equal to the warmup period in all cases, using 1-sqrt cooldown (Hägele et al., 2024).

We use slightly different tuning configurations for vs `Adam`. For the 16M model tuning runs for `ADOPT`, which last 4608 steps, the warmup period is set to $T_{\text{WARM}} = 512$ steps. For `Adam` we use 12288 steps as the tuning period with a warmup period set to $T_{\text{WARM}} = 2048$.

### B.2 Optimizer Hyperparameter Sweeping Procedure

Our tuning procedure is designed to ensure that both our method and the baselines are evaluated under their optimal DDP configurations, providing a fair comparison. Given that previous work has shown that the learning rate (LR) tends to transfer effectively between DDP and distributed settings (Iacob et al., 2025), we first tune all parameters to achieve the best possible performance under DDP and then transfer these settings to `MT-DAO`. Unlike methods such as `AdEMAMix` that use schedulers for optimizer parameters, we employ a simple switch from a base optimizer (e.g., `ADOPT`) to its multi-timescale variant at the end of the warmup period. This necessitates a two-phase LR tuning process to ensure identical starting conditions for both optimizers:

1. **Phase 1: Base Optimizer Tuning.** We first tune the learning rate for the base optimizer over the entire training run to achieve the lowest final perplexity. This ensures the baseline itself is as strong as possible.

2. **Phase 2: `MT-DAO`/Quasi-hyperbolic Tuning.** Using the model state from the end of the base optimizer's warmup, we then tune the learning rate for the post-switch phase of `MT-DAO` and of its `DDP` analogue. With a `WSD` scheduler, this corresponds to tuning the LR for the constant "stable" portion of training and for the cooldown.

While more complex scheduling manipulations might yield further gains for `MT-DAO`, this two-phase approach provides the cleanest methodology for comparison. For every combination of momentum decay rates ($\beta$'s) and convex coefficients ($\omega$'s) used by `MT-DAO`, we independently perform this tuning procedure. For `ADOPT` the LR sweeps in both phases search over values between $2^{-10}$ and $2^{-6}$ using powers of two, with the search grid refined by manually adding half-power steps (e.g., $2^{-8.5}, 2^{-7.5}, 2^{-6.5}$) around the optimal value.

### B.3 Optimizer Tuning Results

First, our tuning of `ADOPT` for `DDP` revealed an optimal lr $\eta^* = 2^{-8}$ while our tuning for `Adam` revealed an optimal learning rate of $\eta^* = 10^{-3}$. We now present the results of our tuning for the post-warmup lr for `MT-DAO-ADOPT` with $N = 1$ first momenta in Fig. 8.

A clear trend emerges from our results: methods with higher momentum decay rates ($\beta$s) or higher weights ($\omega$s) ascribed to the slow-moving momenta can tolerate significantly higher learning rates than standard momentum methods. This finding is in strong agreement with the previous findings of Lucas et al. (2019), who similarly found that `AggMo` can effectively utilize learning rates that are orders of magnitude higher than those suitable for classical momentum.

> **Takeaway:** Multi-timescale optimizers that emphasize slow-moving momenta (via high $\beta$ or $\omega$ values) are not only more stable but can also leverage much higher learning rates, enabling faster convergence than their single-timescale counterparts.

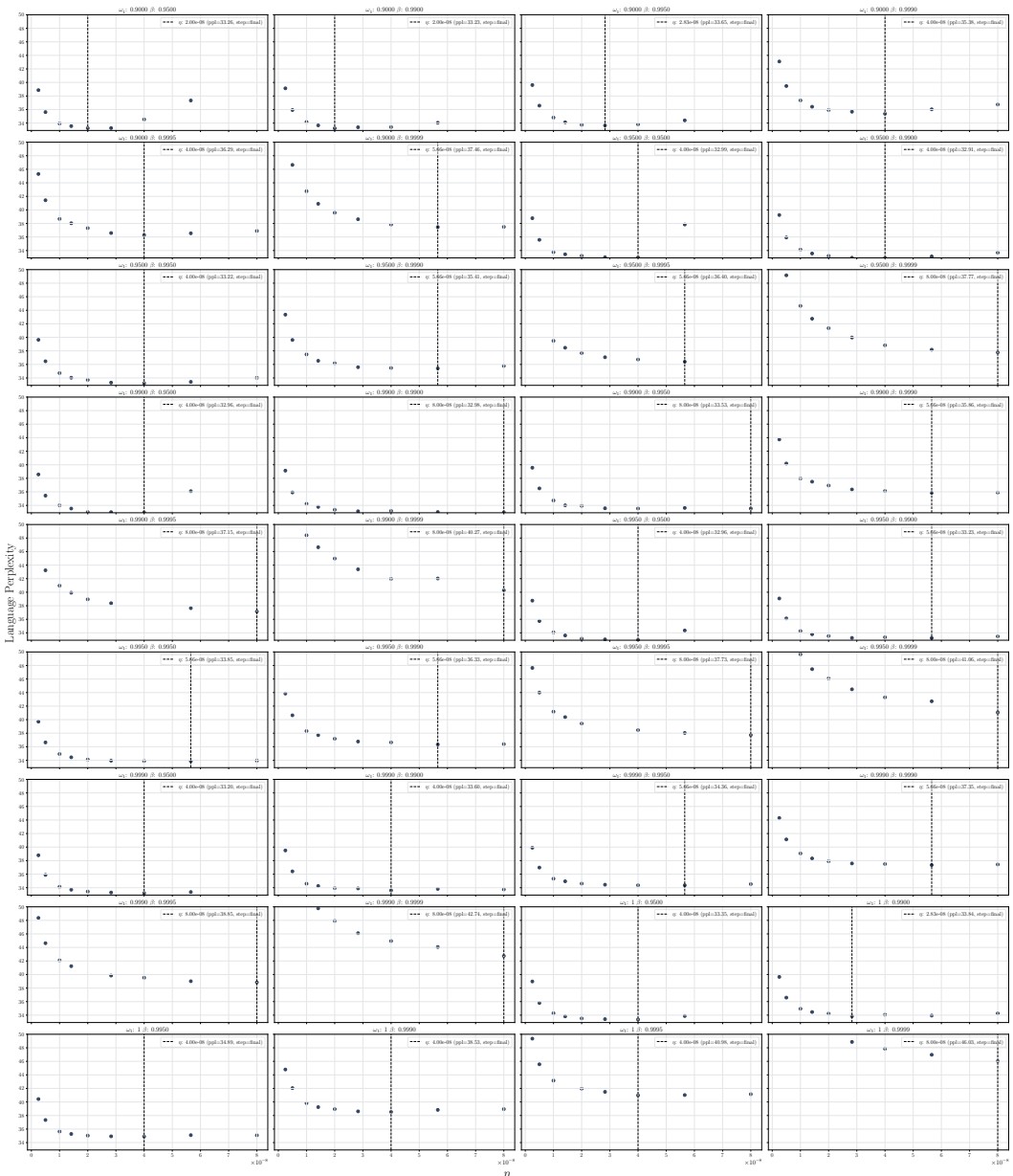

Figure 8: Visualizing the learning rate sweeps for different MT-DAO configurations. Each subplot shows the final perplexity for a given convex coefficient ($\omega$) and momentum decay ($\beta_1$), where $\beta_2 = 0.9999$ was kept constant. The sweep demonstrates that the optimal learning rate and final performance are highly dependent on the choice of these internal hyperparameters, with $\beta_1 \in [0.995, 0.999]$ and $\omega \in [0.9, 0.99]$ performing best for these short tuning experiments. The vertical line in each subplot marks the best-performing lr for that configuration. Switch scale referes to the multiple of the base learning rate that we select, the chosen learning rate can be computed via multiplication with $\eta_{\mathrm{BASE}}$.

## C   DETERMINISTIC OPTIMIZER-SPECIFIC VARIANTS OF MT-DAO

---

**Algorithm 2** `MT-DAO-ADOPT`

---

**Require: Model tensors, hyper-parameters**
1: $x_0 \in \mathbb{R}^d, \{\bar{u}^j_{-1}\}^N_{j=1} \in (\mathbb{R}^d)^N, \bar{v}_{-1} \in \mathbb{R}^d$ — initial params, $N$ first momenta, second momentum
2: $\{\beta_{1,j}\}^N_{j=1}, \beta_2 \in [0, 1)$ — decay rates for each momentum state
3: $\{\omega_j\}^N_{j=1} \in [0, 1]$ — convex combination coefficients for first momenta, $\sum^N_{j=1} \omega_j \leq 1.0$
4: $\{c_t\}^{T-1}_{t=0}, \{\eta_t\}^{T-1}_{t=0}$ — clipping and learning-rate schedules
5: $T, M \in \mathbb{N}_+$ — total optimization steps and number of workers
6: $K_x, \{K_j\}^N_{j=1}, K_v \in (\mathbb{N}_+)^{N+2}$ — communication periods for parameters and states
7: $\mathtt{OuterOpt} : \mathbb{R}^d \rightarrow \mathbb{R}^d$ — update params using an outer optimizer, averaging by default

**Ensure:** $x_T, \{u^j_{T-1}\}^N_{j=1}, v_{T-1}$
8: **for each worker** $m$: initialize $x^m_0, \{u^{j,m}_{-1}\}, v^m_{-1}$
9: **for** $t = 0, \dots, T - 1$ **do**
10:     **for all** workers $m = 0, \dots, M - 1$ **in parallel do**
11:         $g^m_t \leftarrow \nabla F(x^m_t; \xi^m_t)$          *stochastic gradient*
12:         $v^m_t \leftarrow \beta_2 \bar{v}_{t-1} + (1 - \beta_2)(g^m_t)^2$          *update second momentum with raw gradient*
13:         $\bar{v}_t \leftarrow$ **if** $(t \bmod K_v = 0)$ **then** $\mathbb{E}_m[v^m_t]$ **else** $v^m_t$          *sync $v$ every $K_v$ steps*
14:         $\hat{g}^m_t \leftarrow \frac{g^m_t}{\sqrt{v^m_t} + \epsilon}$          *normalize gradient (ADOPT core step)*
15:         $\tilde{g}^m_t \leftarrow \mathbf{clip}(\hat{g}^m_t, c_t)$          *clip the normalized gradient*
16:         **for** $j = 1$ **to** $N$ **do**          *update N first momenta*
17:             $u^{j,m}_t \leftarrow \beta_{1,j} \bar{u}^j_{t-1} + (1 - \beta_{1,j})\tilde{g}^m_t$          *use clipped, normalized gradient*
18:             $\bar{u}^j_t \leftarrow$ **if** $(t \bmod K_j = 0)$ **then** $\mathbb{E}_m[u^{j,m}_t]$ **else** $u^{j,m}_t$          *sync $u^j$ every $K_j$ steps*
19:         $\Delta^m_t \leftarrow (1 - \sum^N_{j=1} \omega_j)\tilde{g}^m_t + \sum^N_{j=1} \omega_j u^{j,m}_t$          *form combined update direction*
20:         $x^m_{t+1} \leftarrow \bar{x}_t - \eta_t \Delta^m_t$          *apply combined update*
21:         $\bar{x}_{t+1} \leftarrow$ **if** $((t + 1) \bmod K_x = 0)$ **then** $\mathtt{OuterOpt}(\mathbb{E}_m[x^m_{t+1}])$ **else** $x^m_{t+1}$          *sync $x$ every $K_x$ steps*

---

**Algorithm 3** `MT-DAO-SGDM`

---

**Require: Model tensors, hyper-parameters**
1: $x_0 \in \mathbb{R}^d, \{\bar{u}^j_{-1}\}^N_{j=1} \in (\mathbb{R}^d)^N$ — initial params, $N$ first momenta
2: $\{\beta_{1,j}\}^N_{j=1} \in [0, 1)$ — decay rates for each momentum state
3: $\{\omega_j\}^N_{j=1} \in [0, 1]$ — convex combination coefficients for first momenta, $\sum^N_{j=1} \omega_j \leq 1.0$
4: $\rho \in \mathbb{R}_+, \{\eta_t\}^{T-1}_{t=0}$ — clipping radius, learning-rate schedule
5: $T, M \in \mathbb{N}_+$ — total optimization steps and number of workers
6: $K_x, \{K_j\}^N_{j=1} \in (\mathbb{N}_+)^{N+1}$ — communication periods for parameters and states
7: $\mathtt{OuterOpt} : \mathbb{R}^d \rightarrow \mathbb{R}^d$ — update params using an outer optimizer, averaging by default

**Ensure:** $x_T, \{u^j_{T-1}\}^N_{j=1}$
8: **for each worker** $m$: initialize $x^m_0, \{u^{j,m}_{-1}\}$
9: **for** $t = 0, \dots, T - 1$ **do**
10:     **for all** workers $m = 0, \dots, M - 1$ **in parallel do**
11:         $\hat{g}^m_t \leftarrow \mathbf{clip}(\nabla F(x^m_t; \xi^m_t), \rho)$          *clipped stochastic gradient*
12:         **for** $j = 1$ **to** $N$ **do**          *update N first momenta*
13:             $u^{j,m}_t \leftarrow \beta_{1,j} \bar{u}^j_{t-1} + (1 - \beta_{1,j})\hat{g}^m_t$
14:             $\bar{u}^j_t \leftarrow$ **if** $(t \bmod K_j = 0)$ **then** $\mathbb{E}_m[u^{j,m}_t]$ **else** $u^{j,m}_t$          *sync $u^j$ every $K_j$ steps*
15:         $\Delta^m_t \leftarrow (1 - \sum^N_{j=1} \omega_j)\hat{g}^m_t + \sum^N_{j=1} \omega_j u^{j,m}_t$          *form combined update direction (unnormalized)*
16:         $x^m_{t+1} \leftarrow \bar{x}_t - \eta_t \Delta^m_t$
17:         $\bar{x}_{t+1} \leftarrow$ **if** $((t + 1) \bmod K_x = 0)$ **then** $\mathtt{OuterOpt}(\mathbb{E}_m[x^m_{t+1}])$ **else** $x^m_{t+1}$          *sync $x$ every $K_x$ steps*

---

## D    CONVERGENCE ANALYSIS OF `MT-DAO-SGDM`

In order to facilitate the technical presentation, we model synchronization frequencies by assigning probabilities to each averaging event. For example, the parameters $x^m_t$ are synchronized with the

---

**Algorithm 4** `MT-DAO-SGDM`, probabilistic variant

---

**Require: Model tensors, hyper-parameters**

1:    $x_0 \in \mathbb{R}^d$, $\{u_{-1}^j\}_{j=1}^N \in (\mathbb{R}^d)^N$ — initial parameters, $N$ first momenta

2:    $\{\beta_j\}_{j=1}^N \in [0, 1)$ — decay rates for each momentum state

3:    $\{\omega_j\}_{j=1}^N \in [0, 1]$ — convex combination of non-negative coefficients for first momenta, $\sum_{j=1}^N w_j = 1$

4:    $\{\eta_t\}_{t=0}^{T-1}$ — learning-rate schedule

5:    $T, M \in \mathbb{N}_+$ — total optimization steps and number of workers

6:    $p_x = \frac{1}{K_x}, \{p_j = \frac{1}{K_j}\}_{j=1}^N \in [0, 1]^{N+1}$ — communication periods/probabilities for parameters and states

**Ensure:** $x_T$, $\{u_{T-1}^j\}_{j=1}^N$

7: **for each worker** $m$: initialize $x_0^m$, $\{u_{-1}^{j,m}\}$

8: **for** $t = 0, \ldots, T - 1$ **do**

9:      **for all** workers $m = 0, \ldots, M - 1$ **in parallel do**

10:         $g_t^m \leftarrow \nabla F_m(x_t^m; \xi_t^m)$                                   *stochastic gradient*

11:         **for** $j = 1$ **to** $N$ **do**                            *update N first momenta*

12:         $u_t^{j,m} \leftarrow \begin{cases} \mathbb{E}_m[\beta_j u_{t-1}^{j,m} + (1 - \beta_j)g_t^m], & \text{with probability } p_j \\ \beta_j u_{t-1}^{j,m} + (1 - \beta_j)g_t^m, & \text{with probability } 1 - p_j \end{cases}$     *sync u*

13:         $\Delta_t^m \leftarrow \sum_{j=1}^N \omega_j u_t^{j,m}$                           *form combined update direction*

14:         $x_{t+1}^m \leftarrow \begin{cases} \mathbb{E}_m[x_t^m - \eta_t \Delta_t^m], & \text{with probability } p_x \\ x_t^m - \eta_t \Delta_t^m, & \text{with probability } 1 - p_x \end{cases}$         *sync x*

---

probability $p_x = \frac{1}{K_x}$, which is statistically equivalent to performing the averaging in every $\frac{1}{p_x} = K_x$ iteration. Similarly, momentum $u_t^{j,m}$ synchronization happens with probability $p_j = \frac{1}{K_j}$, which can differ from $p_x$. Note that QHM structure is included since we can choose $\beta_1 = 0$ and get $u_t^{1,m} = g_t^m$.

Auxiliary notation. Let $\mathbb{E}_m$ and $\mathbb{E}_j$ be the averaging operators with weights $\frac{1}{M}$ across $M$ workers and $\omega_j$ across $N$ momenta.

$$u_t^j \overset{\text{def}}{=} \mathbb{E}_m[u_t^{j,m}] = \beta_j u_{t-1}^j + (1 - \beta_j)g_t, \text{where } g_t = \mathbb{E}_m[g_t^m]$$

$$x_{t+1}^{j,m} \overset{\text{def}}{=} \begin{cases} \mathbb{E}_m[x_t^{j,m} - \eta u_t^{j,m}], & \text{with probability } p_x \\ x_t^{j,m} - \eta u_t^{j,m}, & \text{with probability } 1 - p_x \end{cases}$$

$$x_{t+1}^j \overset{\text{def}}{=} \mathbb{E}_m[x_{t+1}^{j,m}] = x_t^j - \eta u_t^j, \quad x_{t+1}^m = \mathbb{E}_j[x_{t+1}^{j,m}] = \text{(line 14)}.$$

For the sake of notation, we also let $u_t^m = \Delta_t^m = \mathbb{E}_j[u_t^{j,m}]$, $u_t = \mathbb{E}_m[u_t^m]$, $x_t = \mathbb{E}_m[x_t^m]$ in the upcoming derivations.

Step 1 (virtual iterates). Letting $x_{-1}^j = x_0^j = x_0$, define the global virtual iterations as follows

$$z_t^j \overset{\text{def}}{=} \frac{1}{1 - \beta_j} x_t^j - \frac{\beta_j}{1 - \beta_j} x_{t-1}^j, \quad \text{and} \quad z_t \overset{\text{def}}{=} \mathbb{E}_j[z_t^j] \quad \text{for } t \geq 0.$$

The key property of this virtual iterates we are going to exploit in the next steps is that they follow averaged gradients, namely for any $t \geq 0$ we have

$$
\begin{aligned}
z_{t+1} - z_t &= \mathbb{E}_j[z_{t+1}^j - z_t^j] \\
&= \mathbb{E}_j \left[ \left( \frac{1}{1 - \beta_j} x_{t+1}^j - \frac{\beta_j}{1 - \beta_j} x_t^j \right) - \left( \frac{1}{1 - \beta_j} x_t^j - \frac{\beta_j}{1 - \beta_j} x_{t-1}^j \right) \right] \\
&= \mathbb{E}_j \left[ \frac{1}{1 - \beta_j} (x_{t+1}^j - x_t^j) - \frac{\beta_j}{1 - \beta_j} (x_t^j - x_{t-1}^j) \right] \\
&= \mathbb{E}_j \left[ \frac{1}{1 - \beta_j} (-\eta u_t^j) - \frac{\beta_j}{1 - \beta_j} (-\eta u_{t-1}^j) \right] \\
&= \mathbb{E}_j \left[ \frac{-\eta}{1 - \beta_j} (u_t^j - \beta_j u_{t-1}^j) \right] = \mathbb{E}_j[-\eta g_t] = -\eta g_t.
\end{aligned}
$$

Step 2 (smoothness over virtual iterates). Then we apply smoothness of the global loss function $f$ over these global virtual iterates.

$$
\begin{aligned}
f(z_{t+1}) &\leq f(z_t) + \langle \nabla f(z_t), z_{t+1} - z_t \rangle + \frac{L}{2} \|z_{t+1} - z_t\|^2 \\
&= f(z_t) + \underbrace{\langle \nabla f(x_t), z_{t+1} - z_t \rangle}_{I} + \underbrace{\langle \nabla f(z_t) - \nabla f(x_t), z_{t+1} - z_t \rangle}_{II} + \underbrace{\frac{L}{2} \|z_{t+1} - z_t\|^2}_{III}.
\end{aligned}
$$

In the next step, we separately bound each term appearing in the above bound.

Step 3a (one step progress). Bounding term I.

$$
\begin{aligned}
& \mathbb{E} \langle \nabla f(x_t), z_{t+1} - z_t \rangle \\
={}& -\eta \mathbb{E} \left\langle \nabla f(x_t), \frac{1}{M} \sum_{m=1}^{M} g_t^m \right\rangle = -\eta \mathbb{E} \left\langle \nabla f(x_t), \frac{1}{M} \sum_{m=1}^{M} \nabla f_m(x_t^m) \right\rangle \\
={}& -\frac{\eta}{2} \mathbb{E} \|\nabla f(x_t)\|^2 - \frac{\eta}{2} \mathbb{E} \left\| \frac{1}{M} \sum_{m=1}^{M} \nabla f_m(x_t^m) \right\|^2 + \frac{\eta}{2} \mathbb{E} \left\| \nabla f(x_t) - \frac{1}{M} \sum_{m=1}^{M} \nabla f_m(x_t^m) \right\|^2 \\
={}& -\frac{\eta}{2} \mathbb{E} \|\nabla f(x_t)\|^2 - \frac{\eta}{2} \mathbb{E} \left\| \frac{1}{M} \sum_{m=1}^{M} \nabla f_m(x_t^m) \right\|^2 + \frac{\eta}{2} \mathbb{E} \left\| \frac{1}{M} \sum_{m=1}^{M} \nabla f_m(x_t) - \nabla f_m(x_t^m) \right\|^2 \\
\leq{}& -\frac{\eta}{2} \mathbb{E} \|\nabla f(x_t)\|^2 - \frac{\eta}{2} \mathbb{E} \left\| \frac{1}{M} \sum_{m=1}^{M} \nabla f_m(x_t^m) \right\|^2 + \frac{\eta}{2M} \sum_{m=1}^{M} \mathbb{E} \|\nabla f_m(x_t) - \nabla f_m(x_t^m)\|^2 \\
\leq{}& -\frac{\eta}{2} \mathbb{E} \|\nabla f(x_t)\|^2 - \frac{\eta}{2} \mathbb{E} \left\| \frac{1}{M} \sum_{m=1}^{M} \nabla f_m(x_t^m) \right\|^2 + \frac{\eta L^2}{2M} \sum_{m=1}^{M} \underbrace{\mathbb{E} \|x_t - x_t^m\|^2}_{\text{Lemma 3}}.
\end{aligned}
$$

Step 3b (one step progress). Bounding term II.

$$
\begin{aligned}
\mathbb{E} \langle \nabla f(z_t) - \nabla f(x_t), z_{t+1} - z_t \rangle &= -\eta \mathbb{E} \left\langle \nabla f(z_t) - \nabla f(x_t), \frac{1}{M} \sum_{m=1}^{M} \nabla f_m(x_t^m) \right\rangle \\
&\leq \frac{\eta \rho}{2} \mathbb{E} \|\nabla f(z_t) - \nabla f(x_t)\|^2 + \frac{\eta}{2\rho} \mathbb{E} \left\| \frac{1}{M} \sum_{m=1}^{M} \nabla f_m(x_t^m) \right\|^2 \\
&\leq \frac{\eta \rho L^2}{2} \underbrace{\mathbb{E} \|z_t - x_t\|^2}_{\text{Lemma 2}} + \frac{\eta}{2\rho} \mathbb{E} \left\| \frac{1}{M} \sum_{m=1}^{M} \nabla f_m(x_t^m) \right\|^2.
\end{aligned}
$$

Step 3c (one step progress). Bounding term III.

$$
\begin{aligned}
\frac{L}{2}\mathbb{E}\|z_{t+1}-z_t\|^2 &= \frac{\eta^2 L}{2}\mathbb{E}\left\|\frac{1}{M}\sum_{m=1}^{M}g_t^m\right\|^2 \\
&= \frac{\eta^2 L}{2}\mathbb{E}\left\|\frac{1}{M}\sum_{m=1}^{M}g_t^m-\nabla f_m(x_t^m)\right\|^2 + \frac{\eta^2 L}{2}\mathbb{E}\left\|\frac{1}{M}\sum_{m=1}^{M}\nabla f_m(x_t^m)\right\|^2 \\
&= \frac{\eta^2 L}{2M^2}\sum_{m=1}^{M}\mathbb{E}\|g_t^m-\nabla f_m(x_t^m)\|^2 + \frac{\eta^2 L}{2}\mathbb{E}\left\|\frac{1}{M}\sum_{m=1}^{M}\nabla f_m(x_t^m)\right\|^2 \\
&\leq \frac{\eta^2 L}{2M}\sigma^2 + \frac{\eta^2 L}{2}\mathbb{E}\left\|\frac{1}{M}\sum_{m=1}^{M}\nabla f_m(x_t^m)\right\|^2.
\end{aligned}
$$

Step 3abc (one step progress). Combining previous bounds.

$$
\begin{aligned}
\mathbb{E}f(z_{t+1})-\mathbb{E}f(z_t) &\leq \mathbb{E}\underbrace{\langle\nabla f(x_t),z_{t+1}-z_t\rangle}_{I}+\mathbb{E}\underbrace{\langle\nabla f(z_t)-\nabla f(x_t),z_{t+1}-z_t\rangle}_{II}+\mathbb{E}\underbrace{\frac{L}{2}\|z_{t+1}-z_t\|^2}_{III} \\
&\leq -\frac{\eta}{2}\mathbb{E}\|\nabla f(x_t)\|^2 - \frac{\eta}{2}\mathbb{E}\left\|\frac{1}{M}\sum_{m=1}^{M}\nabla f_m(x_t^m)\right\|^2 + \frac{\eta L^2}{2M}\sum_{m=1}^{M}\underbrace{\mathbb{E}\|x_t-x_t^m\|^2}_{\text{Lemma 3}} \\
&\quad + \frac{\eta\rho L^2}{2}\underbrace{\mathbb{E}\|z_t-x_t\|^2}_{\text{Lemma 2}} + \frac{\eta}{2\rho}\mathbb{E}\left\|\frac{1}{M}\sum_{m=1}^{M}\nabla f_m(x_t^m)\right\|^2 \\
&\quad + \frac{\eta^2 L}{2K}\sigma^2 + \frac{\eta^2 L}{2}\mathbb{E}\left\|\frac{1}{M}\sum_{m=1}^{M}\nabla f_m(x_t^m)\right\|^2 \\
&\leq -\frac{\eta}{2}\mathbb{E}\|\nabla f(x_t)\|^2 - \frac{\eta}{2}\left(1-\frac{1}{\rho}-\eta L\right)\mathbb{E}\left\|\frac{1}{M}\sum_{m=1}^{M}\nabla f_m(x_t^m)\right\|^2 \\
&\quad + \frac{\eta\rho L^2}{2}\underbrace{\mathbb{E}\|z_t-x_t\|^2}_{\text{Lemma 2}} + \frac{\eta L^2}{2M}\sum_{m=1}^{M}\underbrace{\mathbb{E}\|x_t-x_t^m\|^2}_{\text{Lemma 3}} + \frac{\eta^2 L}{2M}\sigma^2.
\end{aligned}
$$

Step 4 (final). Now we average over the iterates and apply the bounds derived in Lemmas 2 and 3.

$$
\begin{aligned}
\frac{\mathbb{E}[f(z_T) - f(z_0)]}{T} &= \frac{1}{T} \sum_{t=0}^{T-1} \mathbb{E}[f(z_{t+1}) - f(z_t)] \\
&\leq -\frac{\eta}{2T} \sum_{t=0}^{T-1} \mathbb{E}\|\nabla f(x_t)\|^2 - \frac{\eta}{2}\left(1 - \frac{1}{\rho} - \eta L\right) \frac{1}{T} \sum_{t=0}^{T-1} \mathbb{E}\left\|\frac{1}{M}\sum_{m=1}^{M} \nabla f_m(x_t^m)\right\|^2 \\
&\quad + \frac{\eta\rho L^2}{2} \underbrace{\frac{1}{T}\sum_{t=0}^{T-1} \mathbb{E}\|z_t - x_t\|^2}_{\text{Lemma 1}} + \frac{\eta L^2}{2}\underbrace{\frac{1}{TM}\sum_{t=0}^{T-1}\sum_{m=1}^{M} \mathbb{E}\|x_t - x_t^m\|^2}_{\text{Lemma 2}} + \frac{\eta^2 L}{2M}\sigma^2 \\
&\leq -\frac{\eta}{2T} \sum_{t=0}^{T-1} \mathbb{E}\|\nabla f(x_t)\|^2 - \frac{\eta}{2}\left(1 - \frac{1}{\rho} - \eta L\right) \frac{1}{T} \sum_{t=0}^{T-1} \mathbb{E}\left\|\frac{1}{M}\sum_{m=1}^{M} \nabla f_m(x_t^m)\right\|^2 + \frac{\eta^2 L}{2M}\sigma^2 \\
&\quad + \frac{\eta\rho L^2}{2}\left(\frac{\eta^2 \beta_\omega^2}{M}\sigma^2 + \eta^2 \beta_\omega^2 \frac{1}{T}\sum_{\tau=0}^{T-1} \mathbb{E}\left\|\frac{1}{M}\sum_{m=1}^{M}\nabla f_m(x_\tau^m)\right\|^2\right) \\
&\quad + \frac{\eta L^2}{2}\left(12\eta^2(B^2-1)\psi \cdot \frac{1}{T}\sum_{t=0}^{T-1}\mathbb{E}\|\nabla f(\theta^t)\|^2 + 4\eta^2\psi(\sigma^2 + 3G^2)\right) \\
&\leq -\frac{\eta}{2}\left(1 - 12\eta^2 L^2(B^2-1)\psi\right)\frac{1}{T}\sum_{t=0}^{T-1}\mathbb{E}\|\nabla f(x_t)\|^2 \\
&\quad - \frac{\eta}{2}\left(1 - \frac{1}{\rho} - \eta L - \eta^2\beta_\omega^2\rho L^2\right)\frac{1}{T}\sum_{t=0}^{T-1}\mathbb{E}\left\|\frac{1}{M}\sum_{m=1}^{M}\nabla f_m(x_t^m)\right\|^2 \\
&\quad + \frac{\eta^2 L}{2M}\sigma^2 + \frac{\eta^3 \rho L^2 \beta_\omega^2}{2M}\sigma^2 + 2\eta^3 L^2 \psi(\sigma^2 + 3G^2).
\end{aligned}
$$

Next, we choose $\rho = 2$ and step size $\eta$ such that

$$
\begin{aligned}
12\eta^2 L^2(B^2-1)\psi &\leq \frac{1}{2} &\Longleftrightarrow&\quad \text{to bound the first term} \\
\eta L + 2\eta^2 \beta_\omega^2 L^2 &\leq \frac{1}{2} &\Longleftrightarrow&\quad \text{to bound the second term} \\
12\eta^2 L^2 \psi &\leq \frac{1}{2} &\Longleftrightarrow&\quad \text{from Lemma 3}
\end{aligned}
$$

Notice that

$$
\eta_0 \overset{\text{def}}{=} \left(4L \max\left(\beta_\omega, 6\sqrt{\psi \max(1, B^2-1)}\right)\right)^{-1}
$$

satisfies all three bounds. Then, with any $\eta \leq \eta_0$ we get

$$
\begin{aligned}
\frac{\mathbb{E}[f(z_T) - f(z_0)]}{T} &\leq -\frac{\eta}{4T}\sum_{t=0}^{T-1}\mathbb{E}\|\nabla f(x_t)\|^2 \\
&\quad + \frac{\eta^2 L}{2M}\sigma^2 + \frac{\eta^3 \rho L^2 \beta_\omega^2}{2M}\sigma^2 + 2\eta^3 L^2 \psi(\sigma^2 + 3G^2).
\end{aligned}
$$

Noticing that $z_0 = x_0$ and $f^* \leq f(z_T)$, we have

$$
\frac{1}{T}\sum_{t=0}^{T-1}\mathbb{E}\|\nabla f(x_t)\|^2 \leq \frac{4(f(x_0) - f^*)}{\eta T} + \frac{2\eta L}{M}\sigma^2 + \frac{4\eta^2 L^2 \beta_\omega^2}{M}\sigma^2 + 8\eta^2 L^2 \psi(\sigma^2 + 3G^2).
$$

Furthermore, choosing $\eta = \min(\eta_0, \frac{1}{\sqrt{T}})$, we get the following rate:

$$\frac{1}{T}\sum_{t=0}^{T-1}\mathbb{E}\|\nabla f(x_t)\|^2$$

$$\leq \quad \max\left(1, \frac{1}{\eta_0\sqrt{T}}\right)\frac{4(f(x_0) - f^*)}{\sqrt{T}} + \frac{2L\sigma^2}{M\sqrt{T}} + \frac{4L^2\beta_\omega^2\sigma^2}{MT} + \frac{8L^2\psi(\sigma^2 + 3G^2)}{T}$$

$$\leq \quad \frac{4(f(x_0) - f^*)}{\sqrt{T}} + \frac{2L\sigma^2}{M\sqrt{T}} + \frac{4(f(x_0) - f^*)}{\eta_0 T} + \frac{4L^2\beta_\omega^2\sigma^2}{MT} + \frac{8L^2\psi(\sigma^2 + 3G^2)}{T}$$

$$= \quad \frac{4}{\sqrt{T}}\left(f(x_0) - f^* + \frac{L\sigma^2}{2M}\right) + \mathcal{O}\left(\frac{1 + \beta_\omega^2 + \psi}{T}\right).$$

## D.1 KEY LEMMAS

**Lemma 2.** *For all $T \geq 1$, we have*

$$\sum_{t=0}^{T-1}\|z_t - x_t\|^2 \leq \frac{\eta^2\beta_\omega^2}{M}T\sigma^2 + \eta^2\beta_\omega^2\sum_{t=0}^{T-1}\mathbb{E}\left\|\frac{1}{M}\sum_{m=1}^{M}\nabla f_m(x_t^m)\right\|^2, \tag{3}$$

*where*

$$\beta_\omega \stackrel{\text{def}}{=} \sum_{j=1}^{N}\frac{\omega_j\beta_j}{1 - \beta_j}.$$

*Proof.* Since $u_{-1} = 0$, unrolling the update rule of momentum, for any $t \geq 0$ we get

$$u_t^j = \beta_j u_{t-1}^j + (1 - \beta_j)g_t = (1 - \beta_j)\sum_{\tau=0}^{t}\beta_j^{t-\tau}g^\tau.$$

Using this and the definition of the average iterates, we have

$$z_t^j - x_t^j = \frac{\beta_j}{1 - \beta_j}(x_t^j - x_{t-1}^j) = -\frac{\eta\beta_j}{1 - \beta_j}u_t^j = -\eta\beta_j\sum_{\tau=0}^{t}\beta_j^{t-\tau}g_\tau$$

$$z_t - x_t = \mathbb{E}_j[z_t^j - x_t^j] = \mathbb{E}_j\left[-\eta\beta_j\sum_{\tau=0}^{t}\beta_j^{t-\tau}g_\tau\right] = -\eta\sum_{\tau=0}^{t}\mathbb{E}_j\left[\beta_j^{t-\tau+1}\right]g_\tau$$

$$= -\eta\sum_{\tau=0}^{t}\beta_\omega^{(t-\tau+1)}g_\tau, \quad \text{where } \beta_\omega^{(\tau)} = \mathbb{E}_j\left[\beta_j^\tau\right] = \sum_{j=1}^{N}\omega_j\beta_j^\tau.$$

Let us make another notation for the sum of weights in the above sum and bound it as follows:

$$s_t \quad \stackrel{\text{def}}{=} \quad \sum_{\tau=0}^{t}\beta_\omega^{(t-\tau+1)} = \sum_{\tau=0}^{t}\sum_{j=1}^{N}\omega_j\beta_j^{t-\tau+1}$$

$$= \quad \sum_{j=1}^{N}\omega_j\sum_{\tau=0}^{t}\beta_j^{t-\tau+1} = \sum_{j=1}^{N}\omega_j\frac{\beta_j - \beta_j^{t+2}}{1 - \beta_j} \leq \sum_{j=1}^{N}\frac{\omega_j\beta_j}{1 - \beta_j} \stackrel{\text{def}}{=} \beta_\omega.$$

Using convexity of squared norm, we have

$$\|z_t - x_t\|^2 \quad = \quad \eta^2 s_t^2\left\|\sum_{\tau=0}^{t}\frac{\beta_\omega^{(t-\tau+1)}}{s_t}g_\tau\right\|^2 \leq \eta^2 s_t^2\sum_{\tau=0}^{t}\frac{\beta_\omega^{(t-\tau+1)}}{s_t}\|g_\tau\|^2 \leq \eta^2\beta_\omega\sum_{\tau=0}^{t}\beta_\omega^{(t-\tau+1)}\|g_\tau\|^2$$

Summing over the iterates yields

$$
\begin{aligned}
\sum_{t=0}^{T-1} \mathbb{E}\|z_t - x_t\|^2 &\leq \eta^2 \beta_\omega \sum_{t=0}^{T-1} \sum_{\tau=0}^{t} \beta_\omega^{(t-\tau+1)} \mathbb{E}\|g_\tau\|^2 \\
&= \eta^2 \beta_\omega \sum_{\tau=0}^{T-1} \sum_{t=\tau}^{T-1} \beta_\omega^{(t-\tau+1)} \mathbb{E}\|g_\tau\|^2 = \eta^2 \beta_\omega \sum_{\tau=0}^{T-1} \left( \sum_{t=1}^{T-\tau} \beta_\omega^{(t)} \right) \mathbb{E}\|g_\tau\|^2 \\
&= \eta^2 \beta_\omega \sum_{\tau=0}^{T-1} \left( \sum_{j=1}^{N} \omega_j \frac{\beta_j - \beta_j^{T-\tau+1}}{1-\beta_j} \right) \mathbb{E}\|g_\tau\|^2 \\
&\leq \eta^2 \beta_\omega^2 \sum_{\tau=0}^{T-1} \mathbb{E}\|g_\tau\|^2 \\
&= \eta^2 \beta_\omega^2 \sum_{\tau=0}^{T-1} \mathbb{E}\left\| \frac{1}{M} \sum_{m=1}^{M} g_\tau^m - \nabla f_m(x_\tau^m) \right\|^2 + \eta^2 \beta_\omega^2 \sum_{\tau=0}^{T-1} \mathbb{E}\left\| \frac{1}{M} \sum_{m=1}^{M} \nabla f_m(x_\tau^m) \right\|^2 \\
&= \frac{\eta^2 \beta_\omega^2}{M^2} \sum_{\tau=0}^{T-1} \sum_{m=1}^{M} \mathbb{E}\|g_\tau^m - \nabla f_m(x_\tau^m)\|^2 + \eta^2 \beta_\omega^2 \sum_{\tau=0}^{T-1} \mathbb{E}\left\| \frac{1}{M} \sum_{m=1}^{M} \nabla f_m(x_\tau^m) \right\|^2 \\
&= \frac{\eta^2 \beta_\omega^2}{M} T\sigma^2 + \eta^2 \beta_\omega^2 \sum_{\tau=0}^{T-1} \mathbb{E}\left\| \frac{1}{M} \sum_{m=1}^{M} \nabla f_m(x_\tau^m) \right\|^2.
\end{aligned}
$$

$\square$

**Lemma 3.** *If* $24\eta^2 L^2 \psi \leq 1$, *then*

$$
\frac{1}{MT} \sum_{t=0}^{T-1} \sum_{m=1}^{M} \mathbb{E}\|x_t - x_t^m\|^2 \leq 12\eta^2(B^2-1)\psi \cdot \frac{1}{T} \sum_{t=0}^{T-1} \mathbb{E}\|\nabla f(x_t)\|^2 + 4\eta^2 \psi(\sigma^2 + 3G^2),
$$

*where*

$$
\psi = \frac{4(1-p_x)}{p_x^2} \cdot \sum_{j=1}^{N} \omega_j \frac{(1-\beta_j)(1-p_j)}{1-(1-p_j)\beta_j}
$$

*Proof.* Let us expand the term $\mathbb{E}\|x_{t+1} - x_{t+1}^m\|^2$ using $x_{t+1}^m$'s probabilistic update rule:

$$
\begin{aligned}
\mathbb{E}\|x_{t+1} - x_{t+1}^m\|^2 &= p_x \cdot 0 + (1-p_x) \cdot \mathbb{E}\|x_t - \eta u_t - (x_t^m - \eta u_t^m)\|^2 \\
&= (1-p_x) \cdot \mathbb{E}\|x_t - x_t^m - \eta(u_t - u_t^m)\|^2 \\
&\leq (1-p_x)(1+s)\mathbb{E}\|x_t - x_t^m\|^2 + \eta^2(1-p_x)(1+1/s)\mathbb{E}\|u_t - u_t^m\|^2 \\
&\leq \eta^2(1-p_x)(1+1/s) \sum_{\tau=1}^{t} ((1-p_x)(1+s))^{t-\tau} \mathbb{E}\|u_\tau - u_\tau^m\|^2.
\end{aligned}
$$

where $s > 0$ will be chosen later. Next we expand the term $\mathbb{E}\|u_t^j - u_t^{j,m}\|^2$ using $u_t^{j,m}$'s probabilistic update rule:

$$
\begin{aligned}
\mathbb{E}\|u_t^j - u_t^{j,m}\|^2 &= p_j \cdot 0 + (1-p_j) \cdot \mathbb{E}\left\|\frac{1}{M}\sum_{m=1}^M (\beta_j u_{t-1}^{j,m} + (1-\beta_j)g_{t-1}^m) - (\beta_j u_{t-1}^{j,m} + (1-\beta_j)g_{t-1}^m)\right\|^2 \\
&= (1-p_j)\mathbb{E}\left\|\beta_j(u_{t-1}^j - u_{t-1}^{j,m}) + (1-\beta_j)(g_{t-1} - g_{t-1}^m)\right\|^2 \\
&\leq (1-p_j)\beta_j \mathbb{E}\|(u_{t-1}^j - u_{t-1}^{j,m})\|^2 + (1-p_j)(1-\beta_j)\mathbb{E}\|g_{t-1} - g_{t-1}^m\|^2 \\
&\leq (1-p_j)(1-\beta_j)\sum_{\tau=0}^{t-1}((1-p_j)\beta_j)^{t-\tau-1}\mathbb{E}\|g_\tau - g_\tau^m\|^2
\end{aligned}
$$

$$
\begin{aligned}
\mathbb{E}\|u_t - u_t^m\|^2 &\leq \sum_{j=1}^N \omega_j \mathbb{E}\|u_t^j - u_t^{j,m}\|^2 \\
&\leq \sum_{j=1}^N \omega_j(1-p_j)(1-\beta_j)\sum_{\tau=0}^{t-1}((1-p_j)\beta_j)^{t-\tau-1}\mathbb{E}\|g_\tau - g_\tau^m\|^2 \\
&\leq \sum_{\tau=0}^{t-1}\left(\sum_{j=1}^N \omega_j(1-p_j)(1-\beta_j)((1-p_j)\beta_j)^{t-\tau-1}\right)\mathbb{E}\|g_\tau - g_\tau^m\|^2 \\
&\leq \sum_{\tau=0}^{t-1}\left(\sum_{j=1}^N \omega_j(1-p_j)(1-\beta_j)q_j^{t-\tau-1}\right)\mathbb{E}\|g_\tau - g_\tau^m\|^2.
\end{aligned}
$$

Denote $q_x = (1-p_x)(1+s)$, $q_x' = (1-p_x)(1+1/s)$ and $q_j = (1-p_j)\beta_j$. Combining the previous two bounds, we get

$$
\begin{aligned}
&\frac{1}{M}\sum_{m=1}^M \mathbb{E}\|x_t - x_t^m\|^2 \\
\leq\ & \eta^2 q_x' \sum_{\tau=1}^t q_x^{t-\tau}\frac{1}{M}\sum_{m=1}^M \mathbb{E}\|u_\tau - u_\tau^m\|^2 \qquad\qquad\qquad\qquad\qquad (4) \\
\leq\ & \eta^2 q_x' \sum_{\tau=1}^t q_x^{t-\tau}\frac{1}{M}\sum_{m=1}^M\sum_{\nu=0}^{\tau-1}\left(\sum_{j=1}^N \omega_j(1-p_j)(1-\beta_j)q_j^{\tau-\nu-1}\right)\mathbb{E}\|g_\nu - g_\nu^m\|^2 \\
=\ & \eta^2 q_x' \sum_{j=1}^N \omega_j(1-p_j)(1-\beta_j)\sum_{\tau=1}^t\sum_{\nu=0}^{\tau-1}q_x^{t-\tau-1}q_j^{\tau-\nu}\left[\frac{1}{M}\sum_{m=1}^M \mathbb{E}\|g_\nu - g_\nu^m\|^2\right] \\
=\ & \eta^2 q_x' \sum_{j=1}^N \omega_j(1-p_j)(1-\beta_j)\sum_{\nu=0}^{t-1}\sum_{\tau=\nu+1}^t q_x^{t-\tau}q_j^{\tau-\nu-1}\left[\frac{1}{M}\sum_{m=1}^M \mathbb{E}\|g_\nu - g_\nu^m\|^2\right] \\
=\ & \eta^2 q_x' \sum_{j=1}^N \omega_j(1-p_j)(1-\beta_j)\sum_{\nu=0}^{t-1}\frac{q_x^{t-\nu} - q_j^{t-\nu}}{q_x - q_j}\left[\frac{1}{M}\sum_{m=1}^M \mathbb{E}\|g_\nu - g_\nu^m\|^2\right], \\
=\ & \eta^2 q_x' \sum_{j=1}^N \omega_j(1-\beta_j)(1-p_j)\sum_{\nu=0}^{t-1}\frac{q_x^{t-\nu} - q_j^{t-\nu}}{q_x - q_j}\left[\frac{1}{M}\sum_{m=1}^M \mathbb{E}\|g_\nu - g_\nu^m\|^2\right],.
\end{aligned}
$$

Next, we bound the gradient term above.

$$
\begin{aligned}
\frac{1}{M}\sum_{m=1}^{M}\mathbb{E}\|g_t^m - g_t\|^2 
&= \frac{1}{M}\sum_{m=1}^{M}\mathbb{E}\left\|g_t^m - \frac{1}{M}\sum_{i=1}^{K}g_t^i\right\|^2 \\
&\leq \frac{2}{K}\sum_{m=1}^{M}\mathbb{E}\left\|g_t^m - \nabla f_m(x_t^m) - \frac{1}{M}\sum_{i=1}^{M}(g_t^i - \nabla f_i(x_t^i))\right\|^2 \\
&\quad + \frac{2}{M}\sum_{m=1}^{M}\mathbb{E}\left\|\nabla f_m(x_t^m) - \frac{1}{M}\sum_{i=1}^{M}\nabla f_i(x_t^i)\right\|^2 \\
(\text{Lemma } 4) \quad &\leq \frac{2}{M}\sum_{m=1}^{M}\mathbb{E}\|g_t^m - \nabla f_m(x_t^m)\|^2 - 2\mathbb{E}\left\|\frac{1}{M}\sum_{m=1}^{M}(g_t^m - \nabla f_m(x_t^m))\right\|^2 \\
&\quad + \frac{12L^2}{M}\sum_{m=1}^{M}\mathbb{E}\|x_t - x_t^m\|^2 + 6(B^2-1)\mathbb{E}\|\nabla f(x_t)\|^2 + 6G^2 \\
&\leq 2\sigma^2 + \frac{12L^2}{M}\sum_{m=1}^{M}\mathbb{E}\|x_t - x_t^m\|^2 + 6(B^2-1)\mathbb{E}\|\nabla f(x_t)\|^2 + 6G^2.
\end{aligned}
$$

Averaging over the iterates and plugging this bound to the previous one, we get

$$
\begin{aligned}
&\frac{1}{MT}\sum_{t=0}^{T-1}\sum_{m=1}^{M}\mathbb{E}\|x_t - x_t^m\|^2 \\
\leq\ & \frac{1}{MT}\sum_{t=1}^{T}\sum_{m=1}^{M}\mathbb{E}\|x_t - x_t^m\|^2 \\
\leq\ & \frac{\eta^2 q_x'}{T}\sum_{j=1}^{N}\omega_j(1-\beta_j)(1-p_j)\sum_{t=1}^{T}\sum_{\tau=0}^{t-1}\frac{q_x^{t-\tau} - q_j^{t-\tau}}{q_x - q_j}\left[\frac{1}{M}\sum_{m=1}^{M}\mathbb{E}\|g_\tau - g_\tau^m\|^2\right] \\
=\ & \frac{\eta^2 q_x'}{T}\sum_{j=1}^{N}\omega_j(1-\beta_j)(1-p_j)\sum_{\tau=0}^{T-1}\sum_{t=\tau+1}^{T}\frac{q_x^{t-\tau} - q_j^{t-\tau}}{q_x - q_j}\left[\frac{1}{M}\sum_{m=1}^{M}\mathbb{E}\|g_\tau - g_\tau^m\|^2\right] \\
=\ & \frac{\eta^2 q_x'}{T}\sum_{j=1}^{N}\frac{\omega_j(1-\beta_j)(1-p_j)}{q_x - q_j}\sum_{\tau=0}^{T-1}\left(\frac{q_x(1-q_x^{T-\tau})}{1-q_x} - \frac{q_j(1-q_j^{T-\tau})}{1-q_j}\right)\left[\frac{1}{M}\sum_{m=1}^{M}\mathbb{E}\|g_\tau - g_\tau^m\|^2\right] \\
\leq\ & \frac{\eta^2 q_x'}{T}\sum_{j=1}^{N}\frac{\omega_j(1-\beta_j)(1-p_j)}{q_x - q_j}\sum_{\tau=0}^{T-1}\left(\frac{q_x}{1-q_x} - \frac{q_j}{1-q_j}\right)\left[\frac{1}{M}\sum_{m=1}^{M}\mathbb{E}\|g_\tau - g_\tau^m\|^2\right] \\
=\ & \frac{\eta^2 q_x'}{T}\sum_{j=1}^{N}\frac{\omega_j(1-\beta_j)(1-p_j)}{(1-q_x)(1-q_j)}\sum_{\tau=0}^{T-1}\left[\frac{1}{M}\sum_{m=1}^{M}\mathbb{E}\|g_\tau - g_\tau^m\|^2\right]
\end{aligned}
$$

Now, let us optimize the factor

$$
\frac{q_x'}{1-q_x} = \frac{(1-p_x)(1+{}^1\!/s)}{1-(1-p_x)(1+s)}
$$

by choosing optimal value for $s$ introduced earlier. By the first order optimality condition, we find that the optimal value is $s^* = \frac{1}{\sqrt{1-p_x}} - 1$. Hence, the minimal value of the factor is

$$
\begin{aligned}
\frac{q_x'}{1-q_x} &= \frac{1-p_x}{(1-\sqrt{1-p_x})^2} \\
&= \frac{(1-p_x)(1-\sqrt{1-p_x})^2}{(1-\sqrt{1-p_x})^2(1+\sqrt{1-p_x})^2} = \frac{(1-p_x)(1+\sqrt{1-p_x})^2}{p_x^2} \leq \frac{4(1-p_x)}{p_x^2}.
\end{aligned}
$$

Letting

$$\psi = \frac{4(1-p_x)}{p_x^2} \sum_{j=1}^{N} \omega_j \frac{(1-\beta_j)(1-p_j)}{1-q_j} = \frac{4(1-p_x)}{p_x^2} \sum_{j=1}^{N} \omega_j \frac{(1-\beta_j)(1-p_j)}{1-(1-p_j)\beta_j}$$

and continuing the chain of bounds, we get

$$\frac{1}{MT} \sum_{t=0}^{T-1} \sum_{m=1}^{M} \mathbb{E}\|x_t - x_t^m\|^2$$

$$\leq \eta^2 \psi \cdot \frac{1}{T} \sum_{t=0}^{T-1} \left[ \frac{1}{K} \sum_{m=1}^{M} \mathbb{E}\|g_t - g_t^m\|^2 \right]$$

$$\leq \eta^2 \psi \cdot \frac{1}{T} \sum_{t=0}^{T-1} \left[ \frac{12L^2}{M} \sum_{m=1}^{M} \mathbb{E}\|x_t - x_t^m\|^2 + 6(B^2-1)\mathbb{E}\|\nabla f(x_t)\|^2 + 2\sigma^2 + 6G^2 \right]$$

$$\leq 12\eta^2 L^2 \psi \cdot \frac{1}{TM} \sum_{t=0}^{T-1} \sum_{m=1}^{M} \mathbb{E}\|x_t - x_t^m\|^2$$

$$+ 6\eta^2(B^2-1)\psi \cdot \frac{1}{T} \sum_{t=0}^{T-1} \mathbb{E}\|\nabla f(x_t)\|^2 + 2\eta^2 \psi(\sigma^2 + 3G^2).$$

Assuming $12\eta^2 L^2 \psi \leq 1/2$ and reordering the first term in the bound, we arrive

$$\frac{1}{MT} \sum_{t=0}^{T-1} \sum_{m=1}^{M} \mathbb{E}\|x_t - x_t^m\|^2 \leq 12\eta^2(B^2-1)\psi \cdot \frac{1}{T} \sum_{t=0}^{T-1} \mathbb{E}\|\nabla f(x_t)\|^2 + 4\eta^2 \psi(\sigma^2 + 3G^2).$$

$\square$

**Lemma 4.** *Under smoothness and bounded heterogeneity assumptions 1 and 3, we have*

$$\frac{1}{M} \sum_{m=1}^{M} \left\| \nabla f_m(y^m) - \frac{1}{K} \sum_{i=1}^{K} \nabla f_i(y^i) \right\|^2 \leq \frac{6L^2}{M} \sum_{m=1}^{M} \|y - y^m\|^2 + 3(B^2-1)\|\nabla f(y)\|^2 + 3G^2,$$

*for any $y^1, \ldots, y^m \in \mathbb{R}^d$ and $y = \mathbb{E}_m[y^m]$.*

*Proof.* The proof follows from Lemma 5 of (Iacob et al., 2025) as the result does not depend on the optimizer. $\square$

## E  WALL-CLOCK TIME MODELING

To assess the practical benefits of our proposal, we analyze its impact on total wall-clock time by modeling two distinct synchronization strategies: a simple unified frequency approach and a desynchronized approach based on optimizer state half-lives. We adopt the model from DES-LOC (Iacob et al., 2025) for estimating total training time.

### E.1  WALL-CLOCK TIME MODEL

The total wall-clock time is modeled as the sum of computational and communication time: $t_{\text{total}} = t_{\text{compute}} + t_{\text{comms}}$. The computation time, $t_{\text{compute}}$, is a function of model and dataset size, while the communication time, $t_{\text{comms}}$, depends on the number and size of synchronization events.

For a training process of $T$ total steps, the communication time for an AllReduce operation (Sergeev & Balso, 2018) depends on the payload size, number of workers $M$, bandwidth $B$, and latency $l$. The total time for different methods and strategies is:

**Unified Frequency Methods:** Parameters and all optimizer states are synchronized together every $K$ steps. The total payload is $3d$ (for parameters, first and second momenta). This applies to `Local Adam` and a baseline version of our method, `MT-DAO` (Unified).

$$t_{\text{total,Unified}} = t_{\text{compute}} + \frac{T}{K} \cdot \left[ \frac{2(3d)}{B} \left( 1 - \frac{1}{M} \right) + l \right] \qquad (5)$$

**Half-Life Based Methods:** Parameters ($K_x$), first momentum ($K_u$), and second momentum ($K_v$) are synchronized at different frequencies. This applies to `DES-LOC` and our proposed method, `MT-DAO` (Half-Life).

$$t_{\text{total,Half-Life}} = t_{\text{compute}} + \left( \frac{T}{K_x} + \frac{T}{K_u} + \frac{T}{K_v} \right) \cdot \left[ \frac{2d}{B} \left( 1 - \frac{1}{M} \right) + l \right] \qquad (6)$$

**Limitation:** This model does not account for any potential overlap between computation and communication.

### E.2 EXPERIMENTAL CONFIGURATION

We compare the two synchronization strategies. The **Unified Frequency** strategy serves as a baseline, where all states are synchronized together every $K_x = 32$ steps. This includes `Local Adam`, `Local ADOPT`, and a variant of our method, `MT-DAO` (Unified), which uses a high $\beta_1$ value but is forced to sync at the same frequent rate as its parameters. These methods have equivalent communication costs and will overlap for an iso-token budget, however, the results in Section 5 show that `MT-DAO` achieves the same perplexity as `Local ADOPT` in many fewer optimization steps, outperforming on time-to-perplexity metrics.

The **Half-Life Based** strategy aims to improve efficiency by synchronizing states less frequently if they change slowly. The synchronization frequency is set based on the state's half-life, $\tau_{0.5}(\beta) = \ln(0.5)/\ln(\beta)$. This includes `DES-LOC` and `MT-DAO` (Half-Life). The quasi-hyperbolic (QH) configuration of `MT-DAO` allows it to use an extremely high $\beta_1 = 0.999$, leading to a very long half-life and thus a much lower communication frequency for its first momentum. We use $\beta_2 = 0.999$ for ADAM variants and $\beta_2 = 0.9999$ for ADOPT variants.

Table 3 details the configurations for both strategies.

Table 3: Hyperparameter configurations and synchronization frequencies ($K$) for modeled methods, grouped by synchronization strategy. For the Half-Life strategy, momentum frequencies are set to the closest power of two to their half-life.

| Strategy | Method | $\omega$ **Values** | $\beta_1$ **Values** | $\beta_2$ **Value** | Sync Freq. $K_{u_1}$ | Sync Freq. $K_v$ |
|---|---|---|---|---|---|---|
| *Unified Frequency (All states sync every $K_x = 32$ steps)* | | | | | | |
| Unified | `Local Adam` | N/A | {0.95} | 0.99 | 32 | 32 |
| Unified | `Local ADOPT` | N/A | {0.95} | 0.9999 | 32 | 32 |
| Unified | `MT-DAO-Adam` (Unified) | {0.95} | {0.999} | 0.999 | 32 | 32 |
| Unified | `MT-DAO-ADOPT` (Unified) | {0.95} | {0.999} | 0.9999 | 32 | 32 |
| *Half-Life Based Frequency (States sync at different rates from $K_x = 32$)* | | | | | | |
| Half-Life | `DES-LOC-ADAM` | N/A | {0.95} | 0.99 | 32 | 69 |
| Half-Life | `DES-LOC-ADOPT` | N/A | {0.95} | 0.9999 | 32 | 6931 |
| Half-Life | `MT-DAO-Adam` (Half-Life) | {0.95} | {0.999} | 0.999 | 693 | 693 |
| Half-Life | `MT-DAO-ADOPT` (Half-Life) | {0.95} | {0.999} | 0.9999 | 693 | 6931 |

### E.3 MODELING RESULTS

The following figures present the estimated wall-clock time and communication costs when training a 1B model on 4 H100 machines with a batch size of 2M tokens and sequence length of 2048. The results demonstrate that `MT-DAO` significantly reduces communication cost with both strategies, with the half-life one being generally more effective.

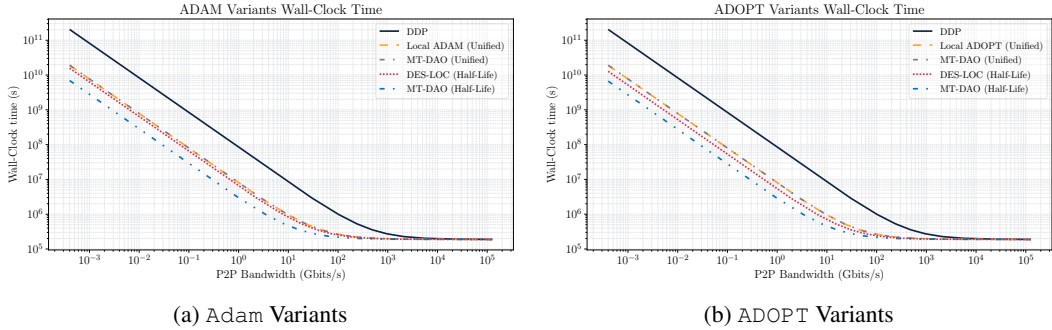

(a) `Adam` Variants

(b) `ADOPT` Variants

Figure 9: Estimated total wall-clock time as a function of interconnect bandwidth. For both (a) `Adam` and (b) `ADOPT`, methods using the Half-Life strategy outperform those using a Unified frequency.

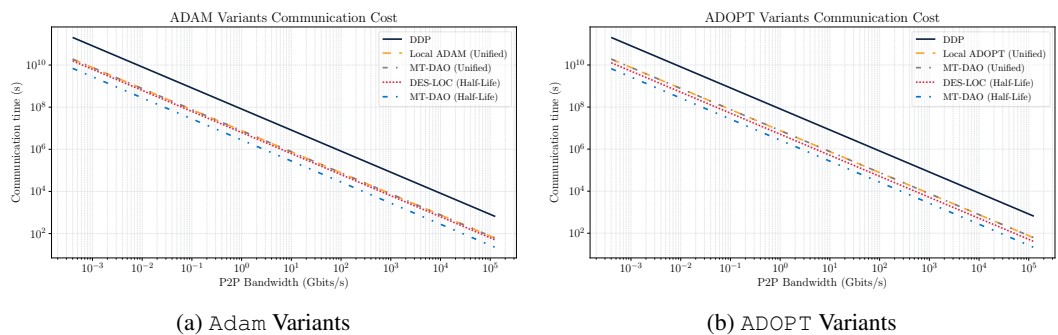

(a) `Adam` Variants

(b) `ADOPT` Variants

Figure 10: Estimated total communication time. The plots clearly distinguish the two strategies. The Unified frequency methods (`Local Adam` and `MT-DAO` (Unified)) have identical high costs. The Half-Life methods are more efficient, with `MT-DAO` (Half-Life) being the most efficient due to its ability to leverage a high-$\beta$ momentum that requires infrequent updates.

> **Takeaway:** `MT-DAO` can significantly reduce communication costs across bandwidths.

## F  DERIVATIONS OF MUTUAL INFORMATION AND VARIANCE

This section provides the detailed derivations for the expressions referenced in the main text.

### F.1  VARIANCE OF LOCAL MOMENTUM

The variance of the final local momentum, $\text{Var}(u_{t+K})$, is derived under the assumption that the stochastic gradients $g_t$ are independent and identically distributed random variables with variance $\sigma_m^2$.

The unrolled momentum update over $K$ local steps is given by:

$$u_{t+K} = \beta^K u_t + (1 - \beta) \sum_{k=0}^{K-1} \beta^k g_{t+K-k}$$

The variance is calculated with respect to the randomness in the local gradients $\{g_{t+1}, \ldots, g_{t+K}\}$. The initial momentum $u_t$ is treated as a constant, as it is a synchronized state before local updates begin.

Applying the variance operator:

$$\text{Var}(u_{t+K}) = \text{Var}\left( \beta^K u_t + (1 - \beta) \sum_{k=0}^{K-1} \beta^k g_{t+K-k} \right)$$

Since $u_t$ is constant, $\text{Var}(\beta^K u_t) = 0$. Using the property $\text{Var}(aX) = a^2 \text{Var}(X)$:

$$\text{Var}(u_{t+K}) = (1-\beta)^2 \text{Var}\left(\sum_{k=0}^{K-1} \beta^k g_{t+K-k}\right)$$

Given the assumption that the gradients $g_{t+i}$ are independent, the variance of their weighted sum is the weighted sum of their variances, where weights are squared:

$$\text{Var}\left(\sum_{k=0}^{K-1} \beta^k g_{t+K-k}\right) = \sum_{k=0}^{K-1} \text{Var}(\beta^k g_{t+K-k}) = \sum_{k=0}^{K-1} (\beta^k)^2 \text{Var}(g_{t+K-k})$$

Assuming each local gradient has variance $\sigma_m^2$:

$$\text{Var}\left(\sum_{k=0}^{K-1} \beta^k g_{t+K-k}\right) = \sum_{k=0}^{K-1} \beta^{2k} \sigma_m^2 = \sigma_m^2 \sum_{k=0}^{K-1} (\beta^2)^k$$

The summation is a finite geometric series, $\sum_{i=0}^{n-1} r^i = \frac{1-r^n}{1-r}$. With $r = \beta^2$ and $n = K$:

$$\sum_{k=0}^{K-1} (\beta^2)^k = \frac{1 - (\beta^2)^K}{1 - \beta^2} = \frac{1 - \beta^{2K}}{1 - \beta^2}$$

Substituting this back into the expression for $\text{Var}(u_{t+K})$:

$$\text{Var}(u_{t+K}) = (1-\beta)^2 \sigma_m^2 \frac{1 - \beta^{2K}}{1 - \beta^2}$$

By factoring the denominator $1 - \beta^2 = (1-\beta)(1+\beta)$, we can simplify the expression:

$$\text{Var}(u_{t+K}) = (1-\beta)^2 \sigma_m^2 \frac{1 - \beta^{2K}}{(1-\beta)(1+\beta)} = \frac{1-\beta}{1+\beta}(1 - \beta^{2K})\sigma_m^2$$

This completes the derivation.

### F.2 MUTUAL INFORMATION

The mutual information $I(U_{t+K}; U_t)$ is derived by modeling the momentum states as multivariate Gaussian random vectors. The model for the update process is:

$$U_{t+K} = \beta^K U_t + L$$

The following assumptions are made:

1. The initial momentum $U_t$ is a Gaussian random vector with zero mean and covariance $\Sigma_{U_t}$, i.e., $U_t \sim \mathcal{N}(0, \Sigma_{U_t})$.
2. The accumulated local gradient noise $L$ is a Gaussian random vector with zero mean and covariance $\Sigma_L$, i.e., $L \sim \mathcal{N}(0, \Sigma_L)$.
3. $U_t$ and $L$ are statistically independent.

The mutual information between two random vectors $X$ and $Y$ is defined as $I(X;Y) = h(Y) - h(Y|X)$, where $h(\cdot)$ is the differential entropy. For a $d$-dimensional Gaussian vector $Z \sim \mathcal{N}(\mu, \Sigma)$, the entropy is $h(Z) = \frac{1}{2} \log \det(2\pi e \Sigma)$.

First, we determine the distribution of $U_{t+K}$. As a linear combination of independent Gaussian vectors, it is also Gaussian.

- **Mean**: $\mathbb{E}[U_{t+K}] = \mathbb{E}[\beta^K U_t + L] = \beta^K \mathbb{E}[U_t] + \mathbb{E}[L] = 0$.
- **Covariance**: $\text{Cov}(U_{t+K}) = \text{Cov}(\beta^K U_t + L)$. Due to the independence of $U_t$ and $L$:

$$\Sigma_{U_{t+K}} = \text{Cov}(\beta^K U_t) + \text{Cov}(L) = \beta^{2K} \Sigma_{U_t} + \Sigma_L$$

Thus, $U_{t+K} \sim \mathcal{N}(0, \beta^{2K}\Sigma_{U_t} + \Sigma_L)$.

The entropy of $U_{t+K}$ is:

$$h(U_{t+K}) = \frac{1}{2}\log\det\left(2\pi e(\beta^{2K}\Sigma_{U_t} + \Sigma_L)\right)$$

Next, we determine the conditional entropy $h(U_{t+K}|U_t)$. The distribution of $U_{t+K}$ conditioned on a specific value $U_t = u_t$ is:

$$U_{t+K}|U_t = u_t \sim \mathcal{N}(\beta^K u_t, \Sigma_L)$$

The entropy of this conditional distribution is:

$$h(U_{t+K}|U_t = u_t) = \frac{1}{2}\log\det(2\pi e\Sigma_L)$$

Since this expression does not depend on the specific value $u_t$, the conditional entropy $h(U_{t+K}|U_t)$ is the same.

Now, we compute the mutual information:

$$I(U_{t+K}; U_t) = h(U_{t+K}) - h(U_{t+K}|U_t)$$

$$I(U_{t+K}; U_t) = \frac{1}{2}\log\det\left(2\pi e(\beta^{2K}\Sigma_{U_t} + \Sigma_L)\right) - \frac{1}{2}\log\det(2\pi e\Sigma_L)$$

Using the logarithmic property $\log a - \log b = \log(a/b)$:

$$I(U_{t+K}; U_t) = \frac{1}{2}\log\left(\frac{\det(2\pi e(\beta^{2K}\Sigma_{U_t} + \Sigma_L))}{\det(2\pi e\Sigma_L)}\right)$$

The constant factors $(2\pi e)^d$ cancel out. Using the determinant property $\frac{\det(A)}{\det(B)} = \det(AB^{-1})$:

$$I(U_{t+K}; U_t) = \frac{1}{2}\log\det\left((\beta^{2K}\Sigma_{U_t} + \Sigma_L)\Sigma_L^{-1}\right)$$

Distributing $\Sigma_L^{-1}$ inside the determinant:

$$I(U_{t+K}; U_t) = \frac{1}{2}\log\det\left(\beta^{2K}\Sigma_{U_t}\Sigma_L^{-1} + \Sigma_L\Sigma_L^{-1}\right)$$

$$I(U_{t+K}; U_t) = \frac{1}{2}\log\det\left(I + \beta^{2K}\Sigma_{U_t}\Sigma_L^{-1}\right)$$

This completes the derivation.

## G  EXTENDED RELATED WORK

**Strategies for Communication-Efficient Distributed Training.**  A substantial body of research aims to curtail communication overhead in distributed training, primarily by either reducing the frequency of synchronizations or compressing the data transmitted per round. The first approach, often termed periodic or local SGD, involves performing multiple local optimization steps between global aggregations. This strategy has been extensively analyzed in both IID and non-IID contexts (see Kairouz et al. (2021) for a survey and Lin et al. (2018a)). In the realm of foundation-model pre-training, methods like **DiLoCo** (Charles et al., 2025) have shown that infrequent synchronization can, with careful tuning, achieve performance comparable to or better than standard data parallelism, with scaling laws characterizing its behavior across model sizes (Charles et al., 2025). This paradigm has also been adapted for federated-style pre-training (Sani et al., 2025) and variants with overlapping or eager updates (Douillard et al., 2025; Kale et al., 2025) or with different inner optimizers (Thérien et al., 2025). The second strategy involves compressing communication payloads. Techniques range from randomized quantization (**QSGD**) (Alistarh et al., 2017) and sparse updates tailored for non-IID data (**STC**, **ZeroFL**) (Sattler et al., 2019; Qiu et al., 2022) to one-bit aggregation (**signSGD-MV**) (Bernstein et al., 2018). In practice, these two strategies are often combined; for instance, **FedPAQ** integrates local training with quantization and partial participation to provide strong theoretical guarantees (Reisizadeh et al., 2020).

**Multi-Timescale Momentum for Temporal Mismatches.** The temporal discrepancy between frequent local updates and infrequent global synchronizations creates a need for optimizers that can integrate information across different timescales. Standard momentum, while beneficial in low-curvature landscapes (Sutskever et al., 2013), imposes a compromise: low decay values are responsive but slow, whereas high decay values are fast but prone to oscillations (Lucas et al., 2019). A single exponential moving average (EMA) cannot effectively weight both recent and distant gradients (Pagliardini et al., 2025). Multi-timescale optimizers address this limitation. **Quasi-Hyperbolic Momentum (QHM)** decouples the current gradient's weight from the momentum decay rate ($\beta$) (Ma & Yarats, 2019), recovering methods like Nesterov and Triple Momentum (Scoy et al., 2018). **Aggregated Momentum (AggMo)** maintains and averages multiple momentum buffers with distinct $\beta$ values, using faster-decaying terms to passively damp oscillations caused by slower, more aggressive terms (Lucas et al., 2019). Similarly, **AdEMAMix** mixes a fast EMA with an ultra-slow one (e.g., $\beta_3 = 0.9999$), demonstrating that long-term gradient memory significantly reduces catastrophic forgetting in language models (Pagliardini et al., 2025). This principle of leveraging multiple timescales is also present in other contexts. Optimizers like `Grokfast` (Lee et al., 2024) and `AdMeta` (Chen et al., 2023b) employ nested EMAs for different purposes, providing orthogonal evidence for the value of long-term momentum. While these methods have shown promise in step-wise synchronous training, their potential to resolve the temporal mismatch in communication-efficient distributed optimization remains largely unexplored.

**Perspectives from Federated Optimization.** The field of Federated Learning (FL), particularly in the cross-device setting, offers a rich history of methods for managing statistical heterogeneity and communication constraints, which are central challenges. The foundational **FedAvg** algorithm (McMahan et al., 2017) has inspired numerous successors (see survey by Kairouz et al. (2021)). To counteract *client drift* caused by non-IID data, **FedProx** introduces a proximal regularizer for stability (Li et al., 2020), **SCAFFOLD** employs control variates to reduce gradient variance (Karimireddy et al., 2020b), and **FedNova** normalizes local updates to correct for objective inconsistency (Wang et al., 2020). Server-side momentum (**FedAvgM**) has also been shown to stabilize aggregation under data skew (Hsu et al., 2019). Adaptive methods have been extended to this setting in **Adaptive Federated Optimization** (FEDOPT), which provides nonconvex guarantees for *FedAdam*, *FedYogi*, and *FedAdagrad* (Kingma & Ba, 2015). Furthermore, **Mime** adapts centralized algorithms to FL by marrying control variates with server statistics (Karimireddy et al., 2020a). Personalization techniques, such as meta-learning-based **Per-FedAvg** (Fallah et al., 2020) and **FedL2P** (Lee et al., 2023) or the regularized **Ditto** (Li et al., 2021), complement these global models by improving per-client utility.

**Orthogonal Approaches in Payload Compression and Optimizer Design.** Orthogonal to reducing synchronization frequency, another line of work focuses on compressing the communication payload itself, often in combination with periodic training. Foundational methods include quantization, as in **QSGD** (Alistarh et al., 2017), and sparsification, as in **Deep Gradient Compression** (Lin et al., 2018b), with convergence analyses providing theoretical grounding (Alistarh et al., 2018). More recent work like **CocktailSGD** combines random and top-$k$ sparsification with quantization for aggressive compression during LLM fine-tuning (Wang et al., 2023a). Beyond compressing gradients, some methods compress the optimizer *states*. For instance, **LDAdam** performs adaptive updates using low-rank approximations of gradient statistics (Robert et al., 2025), while **DeMo** decouples momentum across workers and communicates only selected components (Peng et al., 2024). Other advanced optimizers aim for stability and efficiency through different mechanisms; for example, `Lion` uses a sign function with interpolated momentum (Chen et al., 2023a), and `Sophia` employs a Hessian-based pre-conditioner to temper step sizes in high-curvature directions (Liu et al., 2024). These approaches are generally compatible with and can be composed with infrequent synchronization strategies.

## H   ADDITIONAL RESULTS

This section presents the full suite of additional experiments conducted during the review process. We begin with a toy-problem stability check (Fig. 11), followed by time-normalized downstream evaluation (Fig. 12), momentum-count ablations $N \in \{1, 2, 3\}$ (Appendix H.2), batch-size and worker-count sweeps (Appendix H.3), a streaming communication comparison (Appendix H.4),

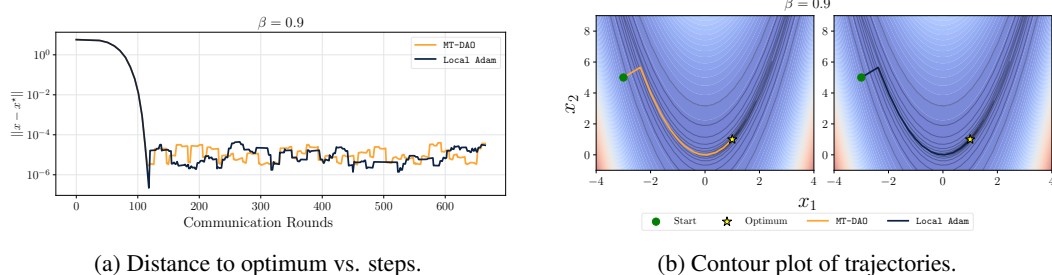

(a) Distance to optimum vs. steps.

(b) Contour plot of trajectories.

Figure 11: `MT-DAO` remains stable in both fast $\beta$ regimes, as pictured here, and in slow $\beta$ regimes as in Figure 1. This is unlike prior stateful methods like `Local Adam` which only offer stable convergence for fast $\beta$ values As before, we optimize the non-convex Rosenbrock function $f(x_1, x_2) = (1 - x_1)^2 + 100(x_2 - x_1^2)^2$ with $M = 256$ workers and IID Gaussian noise ($\sigma = 2$).

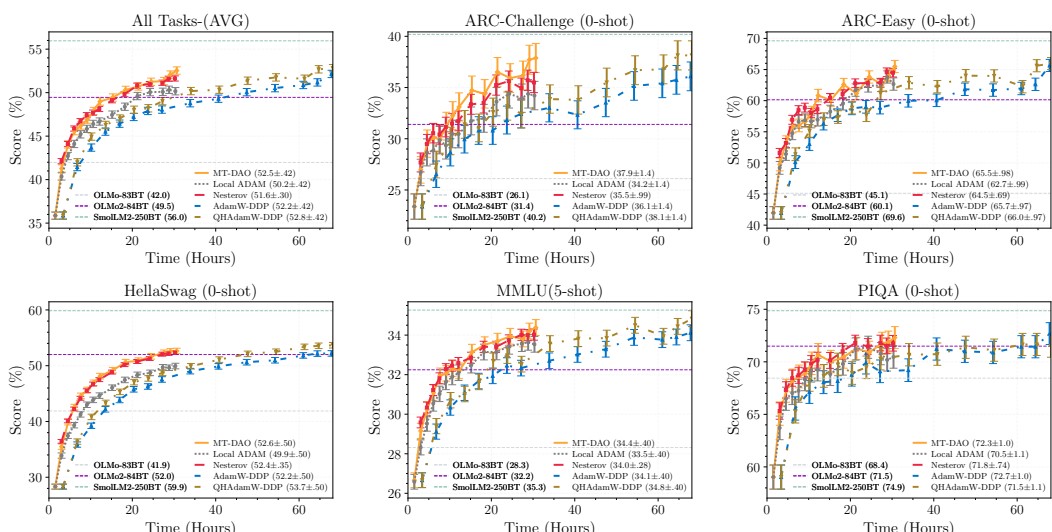

Figure 12: **Downstream task accuracy versus time** on ARC-CHALLENGE, ARC-EASY, HELLASWAG, MMLU, and PIQA. Curves compare `MT-DAO`, `Local ADOPT`, `Nesterov`, `AdamW-DDP`, and `QHAdamW-DDP`; horizontal reference lines (when shown) indicate external model baselines. Error bars denote $\pm\sigma$ over samples. `MT-DAO` **exceeds** `AdamW-DDP` on the aggregate and on reasoning-heavy tasks (ARC-CHALLENGE, HELLASWAG); with smaller but steady gains on MMLU. As training length increases, `MT-DAO` **closes the gap** to `QHAdamW-DDP`, consistent with the perplexity results. `Nesterov` tracks `AdamW-DDP` and **outperforms** `Local ADOPT`, but remains below `MT-DAO` across metrics. tokens-normalized results are provided in Figure 7.

gradient-clipping ablations (Appendix H.5), the synchronous-limit $K{=}1$ analysis, and `Muon` experiments (Appendix H.7). In total, these comprise 235 new experiments spanning five inner optimizers, model scales from 16M to 360M, and up to 8 workers.

To investigate the stability of `MT-DAO` under varied momentum parameterizations, we first examine its performance in a fast $\beta$ regime. Figure 11 presents the results of this comparison, plotting both the convergence rate in terms of distance to the optimum and the optimization trajectories on the function's contour plot.

## H.1 Downstream Evaluation

> **Time-to-Target Advantage:** `MT-DAO` achieves downstream target accuracies faster in wall-clock than `DDP` baselines in a realistic multi-node setup ($4 \times 8$ H100, 100 Gbit), maintaining or improving quality while reducing communication.

## H.2 Ablations on the Number of Momenta $N$

In our main work, we focus on `MT-DAO` with $N = 1$ specifically in its Quasi-Hyperbolic (QH) formulation. We argue this provides a "free lunch": it captures multi-timescale dynamics (via the instantaneous gradient and one slow momentum) without the memory and communication overhead of storing and synchronizing additional buffers.

In this section, we empirically validate this design choice by ablating the number of momenta $N$. Guided by prior work on multi-momentum optimizers (Lucas et al., 2019; Ma & Yarats, 2019; Pagliardini et al., 2025), we define the optimizer update as a convex combination of $N$ states with weights $\sum_{i=1}^{N} \omega_i = 1.0$. We distinguish between two families:

1. **Standard variants:** $N$ momentum buffers are maintained. This corresponds to `AdEMAMix` ($N = 2$) or `AggMo` ($N = 3$).

2. **Quasi-Hyperbolic (QH) variants:** The fastest momentum buffer is replaced by the current preconditioned gradient $g_t$. Mathematically, this is equivalent to setting the fastest $\beta = 0$. This corresponds to ($N = 1$, used in our main text) or `QHAdEMAMix` ($N = 2$).

**Experimental Setup.** To ensure fair comparisons without the confounding factors of complex scheduling introduced in Pagliardini et al. (2025), we use `Adam` as the inner optimizer for all variants in this section. We utilize the WSD learning rate scheduler. Due to the combinatorial explosion of hyperparameters as $N$ increases, we fix the decay rates $\beta$ to exponentially spaced values following the recommendation of Lucas et al. (2019): $\beta_i = 1 - 0.1^i$.

- **Tuning Protocol:** We tune the mixing weights $\omega$ and the learning rate $\eta$ on the 16M model for $12,288$ steps. We then transfer the optimal configurations to 125M and 360M scales trained for $40,960$ steps.

- **Baselines:** We compare `MT-DAO` against `DDP` baselines for every $N$ to measure the performance gap.

### H.2.1 `MT-DAO` ($N = 1$) vs. `MT-DAO` ($N = 2$)

We first compare our default `MT-DAO` ($N = 1$) against a standard `MT-DAO` ($N = 2$) which maintains two explicit momentum buffers.

- **Configurations:** For $N = 1$, we use the gradient and a slow momentum with $\beta_1 = 0.999$. For $N = 2$, we use a fast momentum ($\beta_{1,1} = 0.9$) and a slow momentum ($\beta_{1,2} = 0.999$).

- **Grid Search:** We sweep the weight of the slow momentum $\omega_{\text{slow}} \in \{0.5, 0.6, 0.7, 0.8, 0.9, 0.95\}$ and the learning rate $\eta \in \{1, 2, 4, 8, 16\} \times \eta_{\text{base}}$.

**Results.** Figure 20 and Fig. 21 visualize the tuning results. We find that shifting weight towards the slow momentum is consistently optimal. Figure 13 compares the convergence of the optimal configurations across scales.

We observe a critical distinction between the `DDP` and `MT-DAO` regimes. In `DDP`, adding a fast momentum buffer ($N = 2$) yields a slight perplexity gain over $N = 1$ (0.01 at 16M, 0.02 at 125M and 0.44 at 360M). However, for `MT-DAO`, the $N = 2$ variant performs *worse* than the $N = 1$ variant by approximately 1.5% at 125M scale.

**Interpretation:** In the infrequent communication regime ($K = 32$), a fast momentum ($\beta = 0.9$, $\tau_{0.5} \approx 6$ steps) decays almost entirely between synchronization steps ($\beta^{32} \approx 0.03$). Consequently, the worker's fast momentum buffer becomes decorrelated from the global direction, injecting noise. In contrast, the QH formulation ($N = 1$) uses the instantaneous gradient as the fast component.

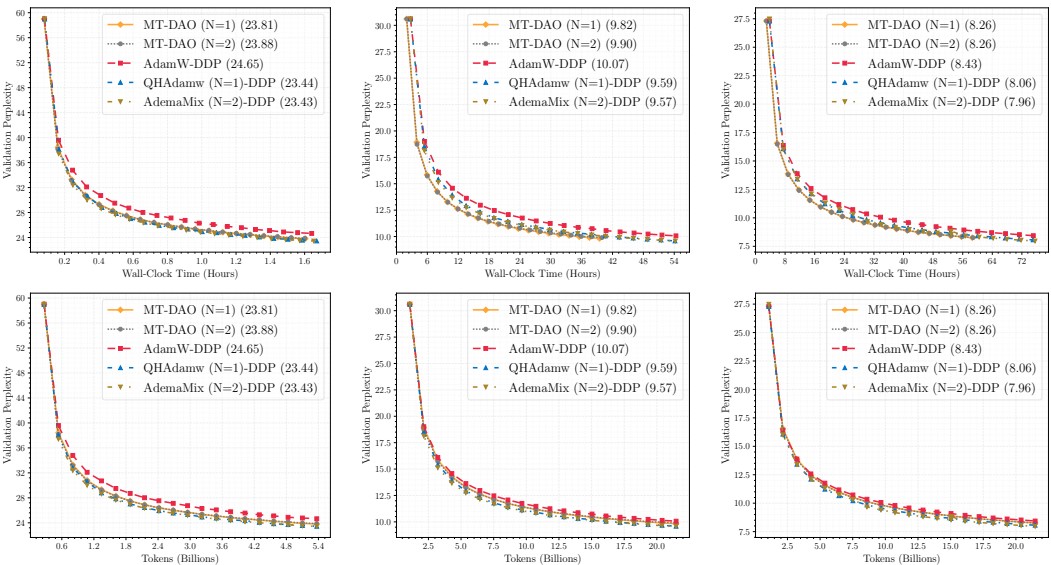

Figure 13: **MT-DAO** ($N = 1$) **vs.** $N = 2$ **variants.** Validation perplexity vs. Wall-clock Time (top) and Tokens (bottom) for 16M, 125M, and 360M models. **Takeaway:** The $N = 1$ Quasi-Hyperbolic variant matches or beats the standard $N = 2$ variant in token-efficiency while being slightly faster in wall-clock time due to lower communication volume.

Thus, `MT-DAO` ($N = 1$) captures the multi-timescale benefit without the instability of a decaying fast momentum.

> **Memory and Communication Efficiency of** $N = 1$**:** `MT-DAO` with $N = 1$ (Quasi-Hyperbolic) matches the token-wise performance of $N = 2$ methods while reducing model-state memory usage and communication volume by 33%.

### H.2.2 QUASI-HYPERBOLIC $N = 2$ VS. STANDARD $N = 3$

To see whether increasing the number of momenta terms beyond $N = 1$ brings about further improvement, we proceed to compare a QH $N = 2$ variant (`QHAdEMAMix`) against a standard $N = 3$ (`AggMo`). We repeat the same experimental design as was done in H.2.1, specifically:

- **Configurations:** For `QHAdEMAMix` ($N = 2$), we use the instantaneous gradient, a medium momentum ($\beta_{1,1} = 0.99$) and a slow momentum ($\beta_{1,2} = 0.999$) term. `AggMo` ($N = 3$) instead uses a fast momentum, ($\beta_{1,1} = 0.9$) a medium momentum ($\beta_{1,2} = 0.99$), and finally a slow ($\beta_{1,3} = 0.999$) momentum term.

- **Grid Search:** For `QHAdEMAMix` and `AggMo`, we sweep over the configurations:

$$\{(0.05, 0.05, 0.9), (0.05, 0.25, 0.8), (0.05, 0.25, 0.7), (0.05, 0.35, 0.6),$$
$$(0.05, 0.45, 0.5), (0.1, 0.1, 0.8), (0.1, 0.2, 0.7), (0.1, 0.3, 0.6), (0.1, 0.4, 0.5)\}$$

  In the case of `QHAdEMAMix`, the first $\omega$ in the tuple refers to $\omega_g$, which is applied to the gradient, whilst for `AggMo` this is $\omega_1$ and this is applied to the fast momentum term. All other elements of the procedure and all other hyper-parameters remain the same, including the optimal base learning rate $\eta = 0.01$.

We note that in the main text we used a slightly higher $\omega_1 = 0.98$ as recommended by Ma & Yarats (2019) beyond what we found to be optimal in short runs in order to guarantee stability at very long training horizons. However, the goal for this section is to provide a fair comparison across $N$'s under similar tuning budgets, so we always use the optimum found by our 12288-step runs.

**Results.** Observing Fig. 22 and Fig. 23, we find that similar trends occur as in H.2.1, where both formulations prefer hyperparameters that shift the weight of the $\omega$ terms to the slow momentum.

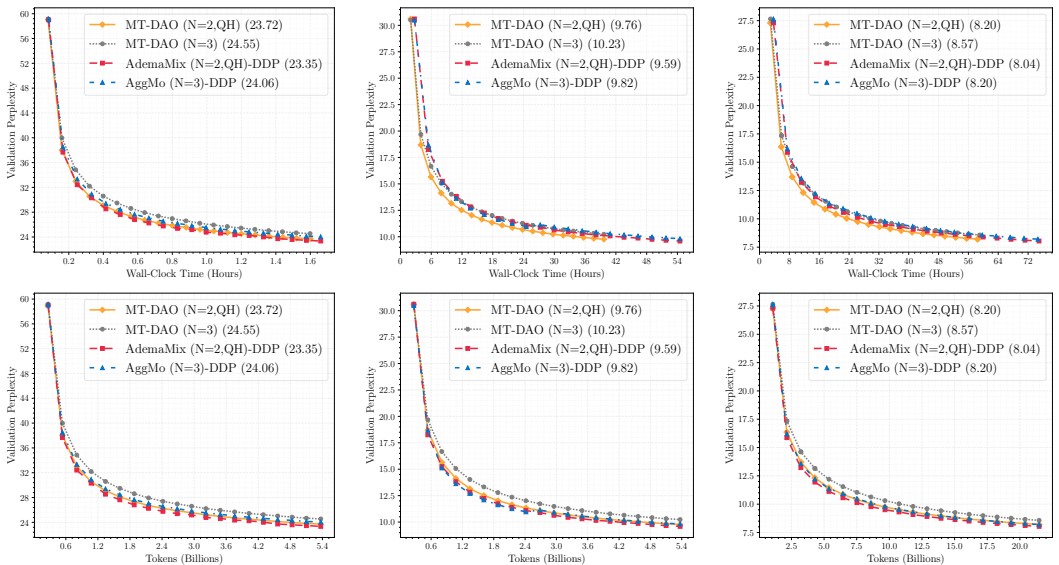

Figure 14: **MT-DAO QH** ($N = 2$) **vs. Standard** $N = 3$ ($K = 32$). Validation perplexity across scales. The QH $N = 2$ variant (orange) consistently overlaps or outperforms the $N = 3$ variant while requiring one fewer memory buffer.

When evaluating the well-tuned models in Figure 14, we see that across model scales, the MT-DAO ($N = 2$,QH) formulations perform better than their $N = 3$ counterparts (0.83 at 16M, 0.47 at 125M and 0.11 at 360M), with the gap between them decreasing as the model size increases. When comparing these results with those observed in Fig. 23, we arrive at the following conclusion: the addition of more momentum buffers has diminishing returns for our method as $N > 1$. As such, this motivates our decision to prioritze MT-DAO with $N = 1$ in our main experiments: it provides a good tradeoff in terms of performance, whilst minimizing the memory overhead of maintaining additional momentum buffers.

> **Diminishing Returns of High N:** Increasing $N$ beyond 1 yields marginal token-wise gains that are outweighed by the increased memory, communication, and tuning complexity. The Quasi-Hyperbolic formulation is consistently the superior choice for infrequent synchronization.

### H.3 ABLATIONS ON THE NUMBER OF WORKERS AND BATCH SIZE

In this section, we investigate the interplay between the global batch size ($B_G$) and the number of workers ($M$) for both MT-DAO and Local Adam. We utilize our small model configuration since we can easily use batch sizes orders of magnitude larger than necessary, simulating very large-batch regimes. For Local Adam, we employ the standard AdamW hyperparameters $\beta_1 = 0.9, \beta_2 = 0.999$. For MT-DAO, we utilize our robust configuration with $\beta_1 = 0.999, \beta_2 = 0.999$ and a convex combination coefficient $\omega_1 = 0.98$. We vary the global batch size $B_G \in \{64, 128, 256, 512\}$ and the worker count $M \in \{1, 2, 4, 8\}$, noting that the effective per-worker batch size $B$ is given by $B_G/M$.

Figure 15 demonstrates that MT-DAO is significantly more robust to increases in the worker count $M$. As $M$ increases for a fixed global batch size, the per-worker batch size $B_G/M$ decreases, injecting higher noise into the local updates. As shown in Figure 15(c), the performance delta generally increases as we move down the y-axis (increasing workers), indicating that Local Adam degrades faster than MT-DAO in high-noise regimes. This robustness stems from the multi-timescale design: MT-DAO relies heavily on the slow, shared momentum (governed by $\omega_1 = 0.98$) rather than the noisy local gradients.

This empirical behavior aligns with our variance analysis derived in Appendix F.1. While increasing the batch size $B_G$ reduces variance linearly, the momentum decay rate $\beta$ controls variance exponentially relative to the synchronization period $K$. The variance of the local momentum state $u_{t+K}$ can

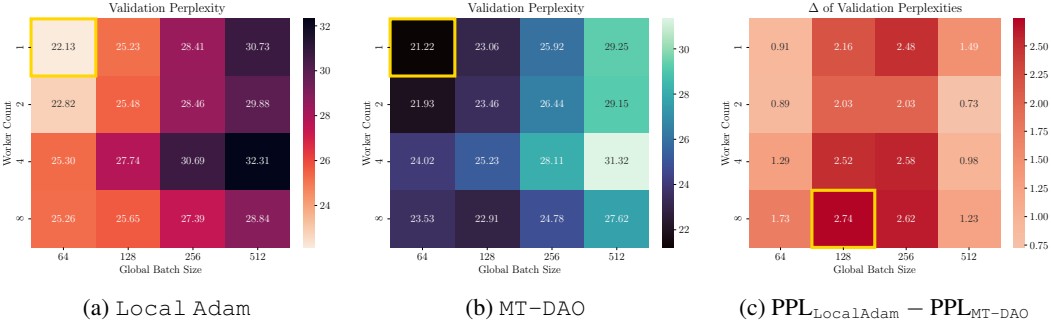

(a) `Local Adam`     (b) `MT-DAO`     (c) $PPL_{LocalAdam} - PPL_{MT-DAO}$

Figure 15: **Robustness to Batch Size and Worker Count:** Validation perplexity heatmaps for (a) `Local Adam` and (b) `MT-DAO` across varying worker counts and global batch sizes. (c) shows the performance gap, where positive values indicate `MT-DAO` outperforms `Local Adam` (lower perplexity is better). `MT-DAO` demonstrates superior stability as the number of workers increases (and per-worker batch size decreases) compared to the baseline.

be approximated as follows for a global batch size $B_G$ and number of workers $M$:

$$\text{Var}(u_{t+K}) \approx \frac{1-\beta}{1+\beta} \underbrace{\left(1-\beta^{2K}\right)}_{\text{Drift Term}} \cdot \underbrace{\frac{\sigma^2}{B_G/M}}_{\text{Batch Noise}} \tag{7}$$

Standard optimizers with low $\beta$ (e.g., 0.9) fail to suppress the drift term $(1-\beta^{2K})$ when $K$ is large. `MT-DAO` utilizes a high $\beta$ (0.999), suppressing this term significantly. Consequently, `MT-DAO` is less sensitive to the linear increase in batch noise $\frac{\sigma^2}{B_G/M}$ caused by increasing $M$, allowing it to scale efficiently even with small per-worker batches.

**Performance at Large Batch Sizes.** Finally, we observe that `MT-DAO` maintains a performance advantage even at large global batch sizes ($B = 512$), more appropriate for a model roughly $45\times$ larger. According to the empirical model of large batch training (McCandlish et al., 2018) and more recent works on the topic (Zhang et al., 2025), there exists a critical batch size past which further increases in batch size do not lead to linear improvements in performance-for-compute due to a decrease in sample efficiency. McCandlish et al. (2018) argue that this happens once the batch size sufficiently denoises the gradient. Given the much better performance of $B_G \in \{64, 128\}$ compared to $B_G = 512$, our model is in this regime. However, despite the global batch size being sufficient to reduce gradient noise, `Local Adam` still struggles due to the timescale mismatch inherent in infrequent communication. `MT-DAO` stabilizes training by preserving the global optimization direction across rounds, proving that the benefits of multi-timescale tracking extend beyond merely compensating for small-batch noise. We also observe that for larger numbers of workers, the fall-off in performance as the global batch size increases is not as sharp as for a single worker (equivalent to `DDP`), for example, in the 8 worker case, the optimal batch size of `MT-DAO` is 128 rather than 64. This corroborates the findings of Charles et al. (2025), which indicate that communication-efficient training methods can benefit from larger batch sizes compared to `DDP`.

> **Robustness to Parallelism:** `MT-DAO` consistently outperforms `Local Adam` as the number of workers increases, effectively mitigating the variance introduced by smaller per-worker batch sizes. It provides stability across diverse batching regimes, from noise-dominated small batches to large-batch settings typical of massive-scale training. We also observe a potential increase in the optimal global batch size for `MT-DAO` at higher numbers of workers.

## H.4 `MT-DAO` COMPARISON AGAINST STREAMINGDILOCO

In this section, we demonstrate that the benefits of `MT-DAO` are orthogonal to and synergistic with the streaming techniques introduced by Douillard et al. (2025). We introduce Streaming `MT-DAO`, a variant that synchronizes fragments of the model and optimizer states continuously in the background, as formalized in Algorithm 5 for `Adam`. We adopt the fragment-based approach where the model

$\theta$ is partitioned into disjoint fragments $\mathcal{F}_1, \ldots, \mathcal{F}_F$. We extend `StreamingDiLoCo`, which only synchronizes parameters (and an outer optimizer state), so that our framework supports $N$ inner momentum states. We decouple the synchronization frequencies of the model parameters ($K_x$), the $N$ first momenta ($\{K_j\}$), and the second momentum ($K_v$). We use averaging as the outer optimizer for all model states by default. The inner update step (L.24) retains the multi-timescale structure of `MT-DAO`. For the `Streaming Local Adam` baseline, we implement the same algorithm but with $N = 1$, standard single-timescale momentum ($\omega_1 = 1.0$).

---

**Algorithm 5** `Streaming MT-DAO` (Fragmented, Overlapped, Multi-Scale)

---

**Require: Model tensors, hyper-parameters**
1:    $x_0 \in \mathbb{R}^d, \{u_{-1}^j\}_{j=1}^N \in (\mathbb{R}^d)^N, v_{-1} \in \mathbb{R}^d$ — initial params, $N$ first momenta, second momentum
2:    $\mathcal{Z}$ — Global state (for all $Z \in \{x, u, v\}$)
3:    $\mathcal{F}_1, \ldots, \mathcal{F}_F$ — partition of tensors into $F$ fragments with offsets $t_f$
4:    $K_x, \{K_j\}_{j=1}^N, K_v \in (\mathbb{N}_+)^{N+2}$ — independent sync periods for tensors
5:    $\alpha, \tau \in [0,1], \mathbb{N}$ — blending factor, inner overlap delay
6:    `OuterOpt`$_Z$ — outer optimizer specific to tensor type $Z \in \{x, u, v\}$
**Ensure:** $x_T, \{u_{T-1}^j\}_{j=1}^N, v_{T-1}$
7: **initialize** worker states $x_0^m, \{u_{-1}^{j,m}\}, v_{-1}^m$ identically for all workers $m$
8: **initialize** global state $\mathcal{Z}_0 \leftarrow Z_{\text{init}}$ for all tensors $Z \in \{x, u^j, v\}$
9: **for parallel** $m = 1 \ldots M$ **do**
10:     **for** $t = 0, \ldots, T-1$ **do**
11:       **for** each fragment $f \in \{1, \ldots, F\}$ **and tensor** $Z \in \{x, \{u^j\}, v\}$ **do**
12:         $K \leftarrow K_Z$         *select freq. for $x, u^j$, or $v$*
13:         **if** $(t - t_f) \bmod K == 0$ **then**     *check sync schedule*
14:           $\Delta_Z^{(f)} \leftarrow Z_t^{(f),m} - \mathcal{Z}_{t-K}^{(f)}$
15:           $\mathrm{h}_Z^{(f)} \leftarrow$ `async-send`$[\frac{1}{M}\sum_{r=1}^M \Delta_Z^{(f)}]$    *broadcast delta*
16:         **if** $(t - t_f - \tau) \bmod K == 0$ **then**    *check overlap schedule*
17:           $\Delta_{\text{agg}} \leftarrow$ `block-receive`$[\mathrm{h}_Z^{(f)}]$    *wait for data*
18:           $\tilde{Z} \leftarrow$ `OuterOpt`$_Z(\mathcal{Z}_{t-\tau-K}^{(f)}, \Delta_{\text{agg}})$
19:           $\mathcal{Z}_{t-\tau}^{(f)} \leftarrow \tilde{Z}$     *archive new global state at index $t - \tau$*
20:           $Z_t^{(f),m} \leftarrow \alpha Z_t^{(f),m} + (1-\alpha)\tilde{Z}$    *merge streams*
21:       $\hat{g}_t^m \leftarrow \text{clip}(\nabla F(x_t^m; \xi_t^m), \rho)$    *clipped stochastic gradient*
22:       **for** $j = 1$ **to** $N$ **do**    *update N first momenta*
23:         $u_t^{j,m} \leftarrow \beta_{1,j} u_{t-1}^{j,m} + (1-\beta_{1,j})\hat{g}_t^m$
24:       $v_t^m \leftarrow \beta_2 v_{t-1}^m + (1-\beta_2)(g_t^m)^2$
25:       $\Delta_t^m \leftarrow \frac{1}{\sqrt{v_t^m}+\epsilon}\left[(1 - \sum_{j=1}^N \omega_j)\hat{g}_t^m + \sum_{j=1}^N \omega_j u_t^{j,m}\right]$
26:       $x_{t+1}^m \leftarrow x_t^m - \eta_t \Delta_t^m$
27: **end parallel for**

---

**Streaming Schedule and Offsets:** Following Douillard et al. (2025), we assign a synchronization offset $t_f$ to each fragment. This enforces a round-robin schedule where, at any given step $t$, only a small subset of the model, specifically, fragments where $(t - t_f) \pmod K = 0$. This design flattens the communication spikes into a more uniform stream.

**Independent State Frequencies:** `Streaming MT-DAO` manages $N + 2$ distinct tensor types: the parameters $x$, $N$ first momenta $\{u^j\}$, and the second momentum $v$. This opens a new design space for distributed optimization: we define independent synchronization periods $K_x, \{K_j\}, K_v$ (L.6). This allows for schedules where rapidly changing parameters are synced frequently, while stable, long-term momentum states (high $\beta$) are synced on slower timescales.

**Computation Overlap:** To hide the latency of these synchronizations, we implement the overlap mechanism proposed by Douillard et al. (2025). The inner optimizer continues stepping for $\tau$ iterations while the outer gradients are transmitted and aggregated. The updates are then merged into the live stream with a blending factor $\alpha$ (Lines 11–16).

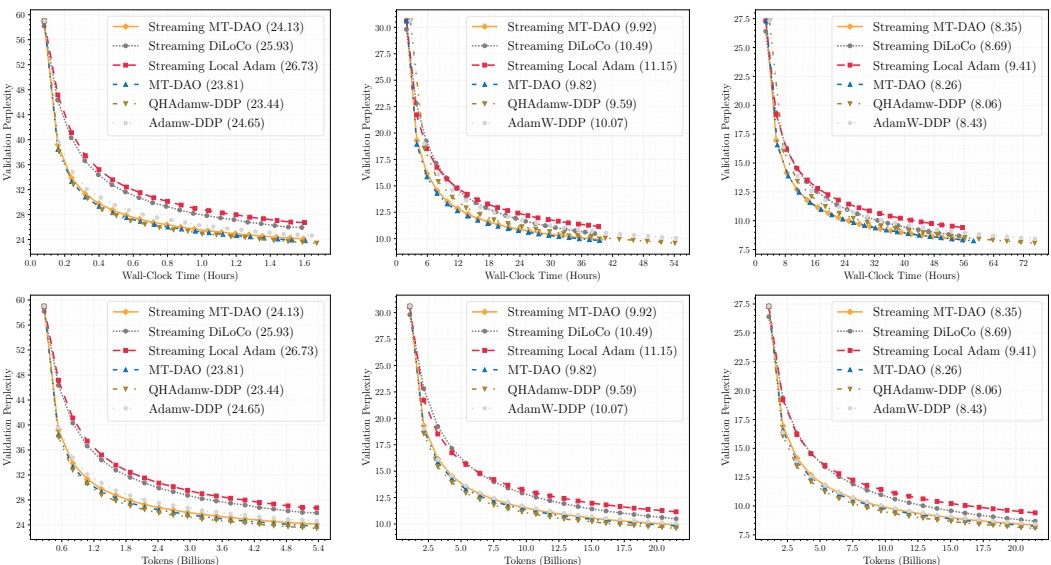

Figure 16: **Streaming MT-DAO vs. StreamingDiLoCo.** Validation perplexity vs. Time (top) and Tokens (bottom) for 16M, 125M, and 360M models. **Takeaway:** `Streaming MT-DAO` (orange) consistently outperforms `StreamingDiLoCo` and `Streaming Local Adam`. While `StreamingDiLoCo` relies on an outer optimizer to reconcile divergent workers, `MT-DAO` prevents excessive divergence via its slow inner momentum, leading to superior stability even with partial, strided communication.

**Experimental Setup.** We adopt the configuration from Douillard et al. (2025), using the strided fragment communication pattern. We set the synchronization periods to $K_x = K_1 = K_v = 128$ (closest power of two to the recommended 100) for `Streaming MT-DAO` ($N = 1$). We use 2 layers per fragment for the 16M model and 3 layers per fragment for the 125M/360M models. We utilize a blending coefficient $\alpha = 0.5$ and a communication overlap of $\tau = 1$ as recommended. For `StreamingDiLoCo`, we use the recommended server learning rate of 0.4 with server momentum 0.9. For `Streaming MT-DAO`, we use the $\omega_1 = 0.95, \beta_1 = 0.999, \beta_2 = 0.999, \eta = 0.002$ `MT-DAO` hyperparameters tuned in Appendix H.2 with the batch size and clipping threshold as for all our other experiments with the same model scale. When using the non-streaming variant of `MT-DAO` we use the standard $K = 32$ unified sync frequency. For `Streaming Local Adam` we use $\omega_1 = 1.0$ (no quasi hyperbolic term) and $\beta_1 = 0.9, \beta_2 = 0.999, \eta = 0.001$ as tuned for the standard `Adam` in Appendix H.2.

**Results.** Figure 16 shows that `Streaming MT-DAO` closely mirrors the performance of `MT-DAO`, with a small gap, across model scales, starting with a 1.3% gap for the small model which shrinks to $\approx 1\%$ for the 125M and 360M models. This is in-line with the findings of Douillard et al. (2025) which indicate very similar performance between the streaming and non-streaming variants of their method. `Streaming MT-DAO` and `MT-DAO` consistently outperform `StreamingDiLoCo` and `Streaming Local Adam`. For example, at the 360M scale, `Streaming MT-DAO` achieves 8.35 perplexity compared to 8.69 for `StreamingDiLoCo`, a $\approx 4\%$ improvement under matched token budgets.

Beyond the effects discussed in Section 5.4, the lower sync frequencies of the streaming regime ($K > 100$) also play a role in the performance gap. With such infrequent syncs, standard fast momentum decays immediately at the start of a round and then optimizes solely for the local trajectory for an extended duration, decoupling the local optimization from the global trajectory. `StreamingDiLoCo` attempts to correct this via the outer Nesterov optimizer, but this correction is infrequent for large $K$. In contrast, the slow momentum component of `MT-DAO` persists across the long streaming intervals, ensuring the local update direction remains anchored to the long-term global trajectory. While our method introduces an additional communication requirement for synchronizing

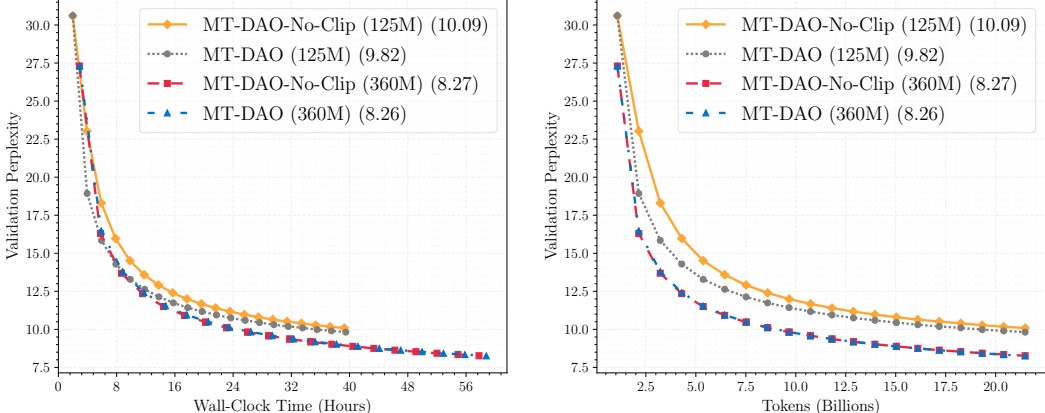

Figure 17: **Sensitivity to Gradient Clipping.** Validation perplexity vs. Time (left) and Tokens (right) for 125M and 360M models trained with standard clipping ($\rho = 0.5$ for the 125M and $\rho = 0.25$ for the 360M, dotted lines) versus no clipping ($\rho = \infty$, solid lines). While the 16M model diverged without clipping (not shown), the 125M and 360M models converged successfully with minimal performance degradation, demonstrating that `MT-DAO`'s heavy reliance on the slow momentum stabilizes updates at scale, we use $\omega_1 = 0.95$.

optimizer states, this overhead is effectively mitigated by the streaming architecture. Due to the communication-computation overlap ($\tau$) and update interpolation, we observe minimal impact on wall-clock training time with our hardware configuration. Furthermore, any potential overhead can be readily offset by increasing the overlap factor $\tau$, allowing computation to proceed uninterrupted. Although we utilize the default $\tau = 1$, which proves sufficient for these model scales, Douillard et al. (2025) demonstrates that values up to $\tau = 10$ remain effective in practice.

> **Synergy with Streaming:** `MT-DAO` is fully compatible with streaming communication. Its slow momentum compensates for the extended staleness of model fragments in the streaming regime, offering a substantial perplexity improvement over `StreamingDiLoCo` without additional overhead.

### H.5 ABLATION ON THE CLIPPING THRESHOLD

In standard LLM pre-training with `Adam`, gradient clipping is considered a critical heuristic to prevent training divergence caused by "spiky" gradients or numerical instabilities where coordinates of the second moment term $\sqrt{v_t}$ approach zero while gradients are large.

**Mechanism of Stability.** In Figure 17, we compare `MT-DAO` trained with standard clipping against a version with clipping disabled ($\rho = \infty$). We observe that for 125M and 360M models, removing clipping results in negligible performance degradation. This stability is intrinsic to the `MT-DAO` design:

1. We only transition from the base optimizer to `MT-DAO` after the warmup phase is complete. By this stage, the gradients have typically stabilized compared to the initial training steps.

2. `MT-DAO` assigns a large convex coefficient to the slow momentum. This naturally dampens outliers without explicit clipping.

**Scale Dependency.** We observe that the robustness to unclipped gradients is scale-dependent. In our experiments, the 16M parameter model diverged immediately without clipping. However, the 125M and 360M models trained stably.

It is important to clarify that the necessity of clipping is primarily a property of the inner optimizer (`Adam`) rather than `MT-DAO` itself. `Adam` is susceptible to divergence when the preconditioner is ill-conditioned or when dealing with rare tokens in heavy-tailed distributions.

> **Clipping Robustness:** `MT-DAO` naturally dampens gradient spikes via its heavy weight on the slow momentum, removing clipping entirely is safe at larger model scales (125M+) where gradient norms are naturally lower. For robust training across all scales, we recommend retaining standard clipping.

## H.6  ABLATION WITH $K = 1$

Our primary design objective for `MT-DAO` is to reduce the performance gap of **communication-efficient** methods relative to fully synchronous `DDP`. In our main experiments ($K = 32$), we demonstrate that `MT-DAO` can match fully synchronous baselines by using slow momentum to bridge the gap between synchronization steps. In this section, we investigate the limit case of $K = 1$ to isolate the behavior of the algorithm when this specific challenge is removed. As shown in Fig. 18, while `MT-DAO` performs similarly to `DDP`, it trails slightly for both $N = 1$ and $N = 2$ at the 16M and 125M scales.

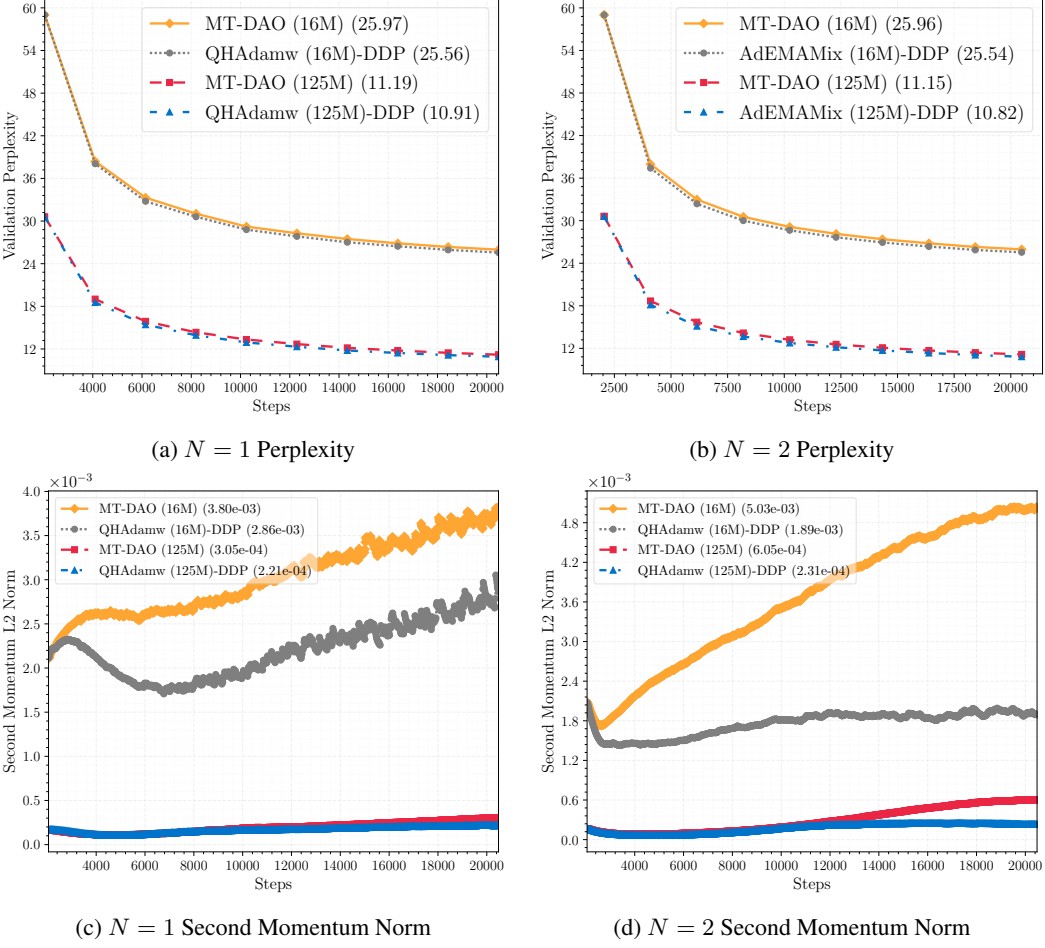

(a) $N = 1$ Perplexity

(b) $N = 2$ Perplexity

(c) $N = 1$ Second Momentum Norm

(d) $N = 2$ Second Momentum Norm

Figure 18: **Comparison at the Synchronous Limit** ($K = 1$). Top: Validation perplexity for `MT-DAO` ($K = 1$) versus `DDP` baselines for $N = 1$ (left) and $N = 2$ (right) variants across 16M and 125M scales. Bottom: The L2 norm of the second momentum state $v_t$ for the corresponding runs. While `MT-DAO` performs similarly to `DDP`, it trails slightly. The bottom plot shows that the local estimation of the second moment in `MT-DAO` leads to a larger norm than the global `DDP` estimate over time, resulting in a slightly lower effective learning rate.

We emphasize that we would not expect `MT-DAO` to **generally** outperform `DDP` at $K = 1$ for two key reasons. First, the noise inherent to Local SGD methods ($K > 1$) can act as an implicit regularizer (Lin et al., 2020), which may account for the shrinking or disappearance of the performance

gap to `DDP` in the main work, depending on the setting. This benefit is strictly lost at $K = 1$. Second, while for `Local SGD` methods with a simple first-order momentum, $K = 1$ is mathematically equivalent to standard `DDP`, for state-of-the-art **adaptive optimizers** this equivalence breaks down due to the non-linearity of the second moment update. We must distinguish between the local gradient estimate $g_t^m$ computed on a per-worker batch of size $B$, and the global gradient estimate $\bar{g}_t = \frac{1}{M} \sum_{m=1}^{M} g_t^m$ computed on the effective global batch of size $M \times B$. `DDP` computes the second moment using the lower-variance global estimate, i.e., $(\bar{g}_t)^2 = \left( \frac{1}{M} \sum_{m=1}^{M} g_t^m \right)^2$. In contrast, `MT-DAO` computes the second moment locally using the noisier $g_t^m$ (derived from batch size $B$) and subsequently averages these states, effectively computing $\frac{1}{M} \sum_{m=1}^{M} (g_t^m)^2$. By Jensen's inequality applied to the convex square function (element-wise), $\frac{1}{M} \sum_{m=1}^{M} (g_t^m)^2 \geq \left( \frac{1}{M} \sum_{m=1}^{M} g_t^m \right)^2$. Since the second moment estimate is a moving average of these squared terms, the `MT-DAO` estimate is guaranteed to be larger than that of `DDP`, resulting in a larger denominator in the adaptive update and implicitly reducing the average per-coordinate effective learning rate.

> **Regime of Applicability:** `MT-DAO` is specialized for infrequent communication ($K \gg 1$). In the synchronous limit ($K = 1$), it performs comparably to `DDP` but trails slightly. This stems from the nonlinearity of adaptive optimizers: by Jensen's inequality, `MT-DAO`'s local estimation yields a larger second momentum than `DDP`'s global estimation, implicitly reducing the effective learning rate as demonstrated empirically. Consequently, we recommend `DDP` as the default choice when fully synchronous training is feasible.

### H.7 Efficacy with Muon

To demonstrate the universality of our approach, we apply `MT-DAO` to `Muon` (Jordan et al., 2024), a recent optimizer that utilizes Newton-Schulz iterations for update orthogonalization. The Newton-Schulz preconditioning is typically applied to the momentum term. This means our multi-timescale considerations regarding momentum variance and drift apply directly to the underlying state before preconditioning occurs. While we leave the full theoretical analysis of preconditioned local updates for future work, our framework is structurally compatible with these methods. We now demonstrate that `MT-DAO` is empirically effective when using `Muon` as an inner optimizer.

**Experimental Setup.** We use the default weight decay of $0.1$ and set Nesterov to `true`, using the default PyTorch implementation of `Muon` with the `match_rms_norm` learning rate adjustment recommended by Liu et al. (2025). Following standard practice for this optimizer, we employ a split optimization strategy: `AdamW` trains the embeddings and layer norms, while `Muon` trains all 2D matrices (Jordan et al., 2024). We independently tune the base `AdamW` learning rate, finding the optimum to be 1e-3, and the Muon learning rate, finding the optimum to be 2e-3. We then jointly tune the multiplier we sweep over for the quasi-hyperbolic variants in Fig. 24. We use $\beta$=0.9 for base `Muon` and $\beta = 0.999$ for `MT-DAO` and `QHMuon`. Adam parameters are fixed at $\beta_1, \beta_2 = 0.9, 0.999$ as in the rest of our work for the base optimizer and $\beta_1, \beta_2 = 0.999, 0.999$ for `MT-DAO` and `QHMuon`. We use the clipping threshold appropriate for `Adam` (Table 2).

**Results.** Our results shown in Figure 19 indicate that `MT-DAO` significantly outperforms base `Muon` and matches or exceeds `QHMuon` at all model scales. Beyond proving that `MT-DAO` is effective with `Muon`, our results are also the first, to the best of our knowledge, to indicate that using an independent weight for the gradient is beneficial for `Muon`. While the base `Muon` implementation includes a `Nesterov` term, this is equivalent to a quasi-hyperbolic formulation where the weight of the gradient is directly tied to the momentum $\beta$. Our hyperparameter sweep shown in Fig. 24 indicates that this coupling is likely to be suboptimal when $\beta$ is very high, since a higher weight should be assigned to the gradient to compensate for the low reactivity of the momentum.

> **Efficacy under Newton-Schulz:** Our findings confirm that the same principles that underlie `MT-DAO` apply effectively within Newton-Schulz preconditioning, enabling `MT-DAO` to match the performance of the base `Muon` optimizer and match or exceed `QHMuon`. We also generally recommend `QHMuon` with a high $\beta$ over standard `Muon`.

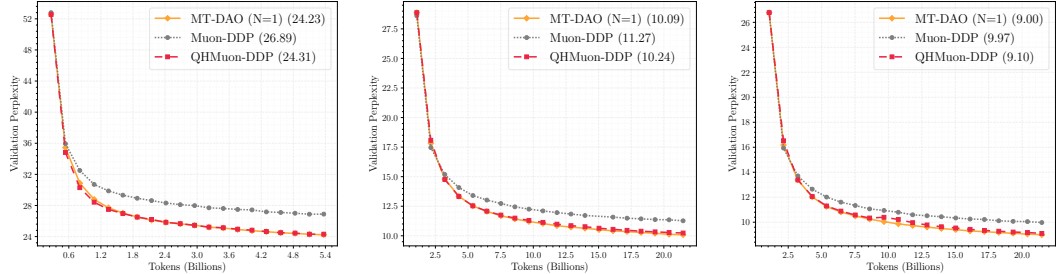

Figure 19: **MT-DAO with Muon.** Validation perplexity vs. Tokens for 16M (left), 125M (middle), and 360M (right) models. `MT-DAO` significantly outperforms the base `Muon-DDP` baseline and matches or exceeds `QHMuon-DDP` at all model scales. This demonstrates that the benefits of multi-timescale optimization and independent gradient weighting transfer effectively to matrix-based optimizers using Newton-Schulz preconditioning.

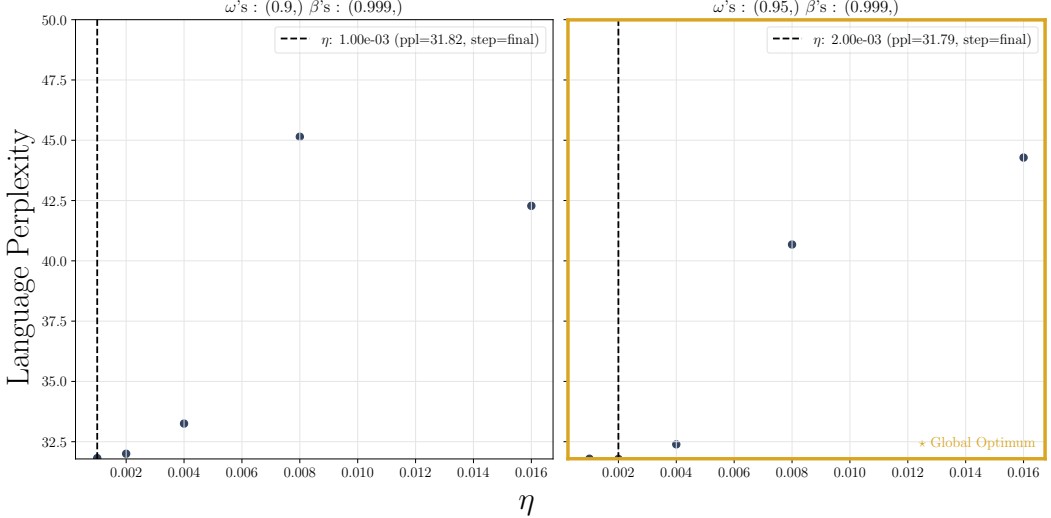

Figure 20: **Hyperparameter Tuning Surface: QH-AdamW** ($N = 1$). Validation perplexity on 16M models. Performance improves as the weight $\omega$ shifts heavily towards the slow momentum ($\beta = 0.999$), confirming that the fast component (gradient) acts primarily as a reactive correction term. We select the optimum (marked in yellow) for scaling experiments.

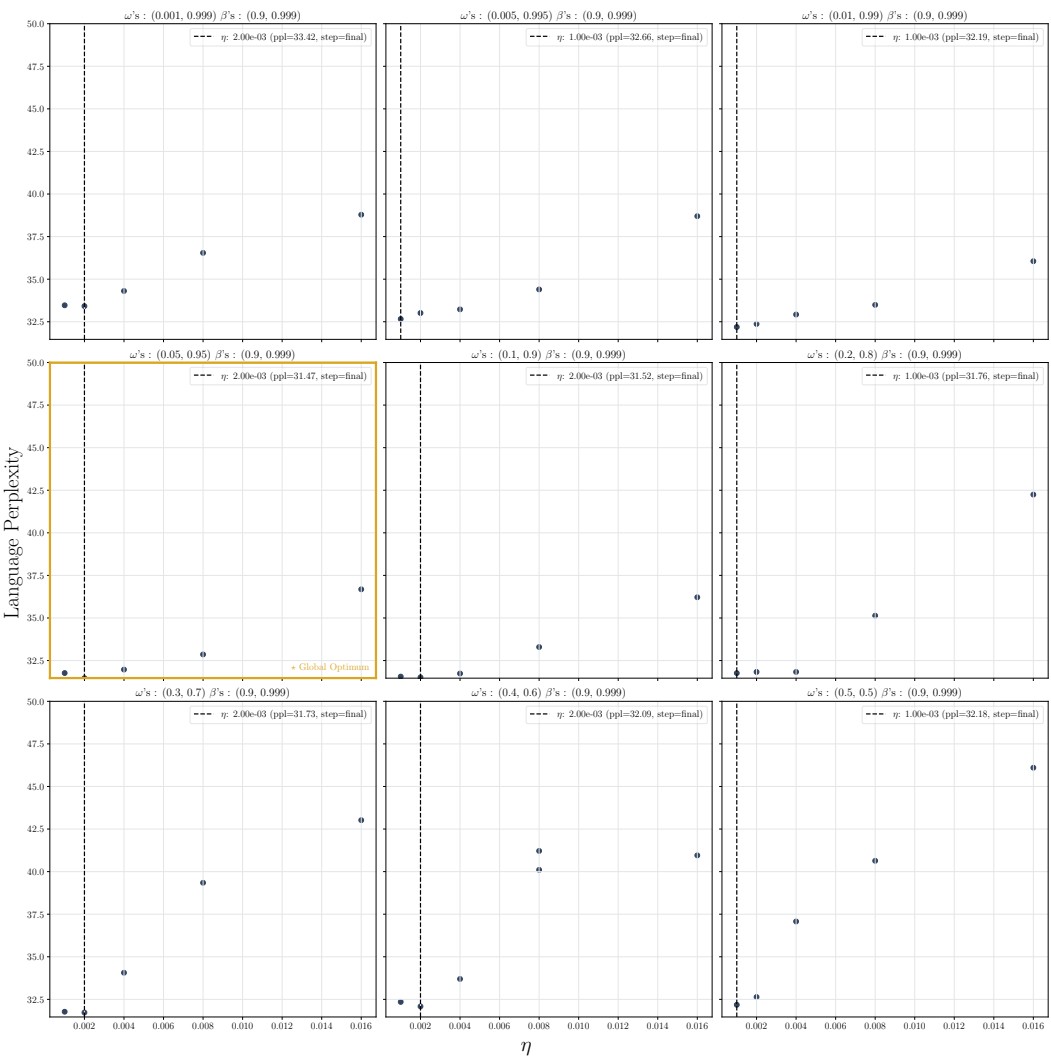

Figure 21: **Hyperparameter Tuning Surface: AdEMAMix** ($N = 2$). Validation perplexity on 16M models. Performance improves as the weight $\omega$ shifts heavily towards the slow momentum ($\beta_2 = 0.95$), confirming that the fast momentum ($\beta_1 = 0.05$) acts primarily as a reactive correction term. We select the optimum (marked in yellow) for scaling experiments.

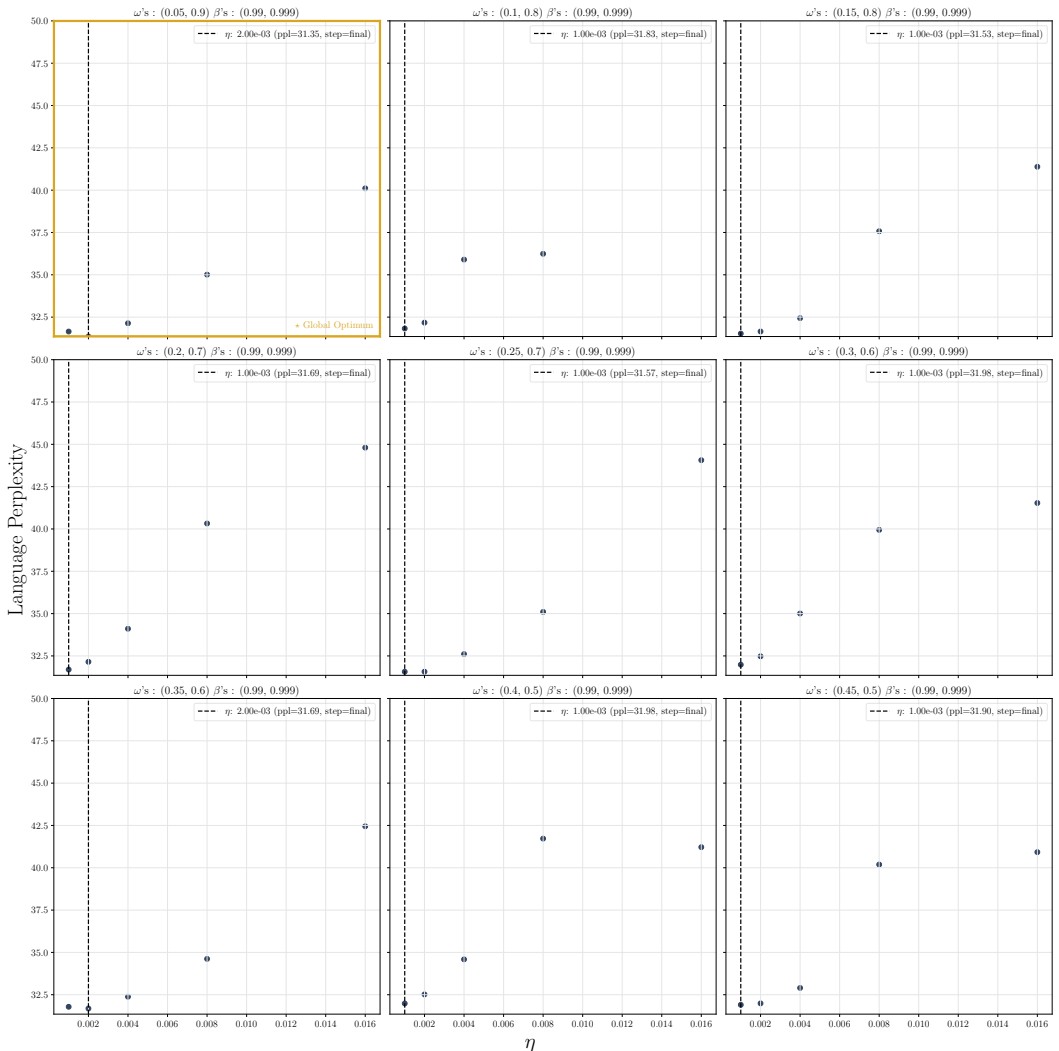

Figure 22: **Tuning** $N = 2$ `QHAdEMAMix`. Optimal performance is found with a heavy bias towards the slowest momentum ($\omega_2 = 0.9$), confirming that long-horizon signal is the primary driver of performance in distributed settings. We select the optimum (marked in yellow) for scaling experiments.

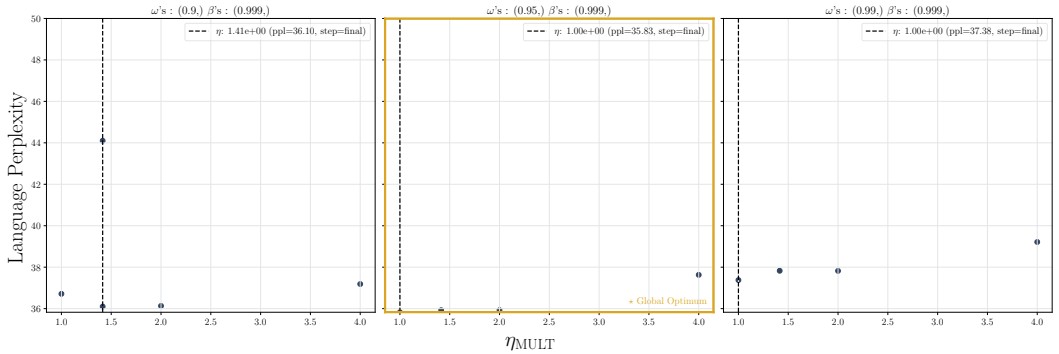

Figure 23: **Tuning** `QHMuon`. Using our pre-defined grid for quasi-hyperbolic $\omega$, we find that utilizing an $\omega = 0.95$ provides the best performance for the `QHMuon` optimizer.

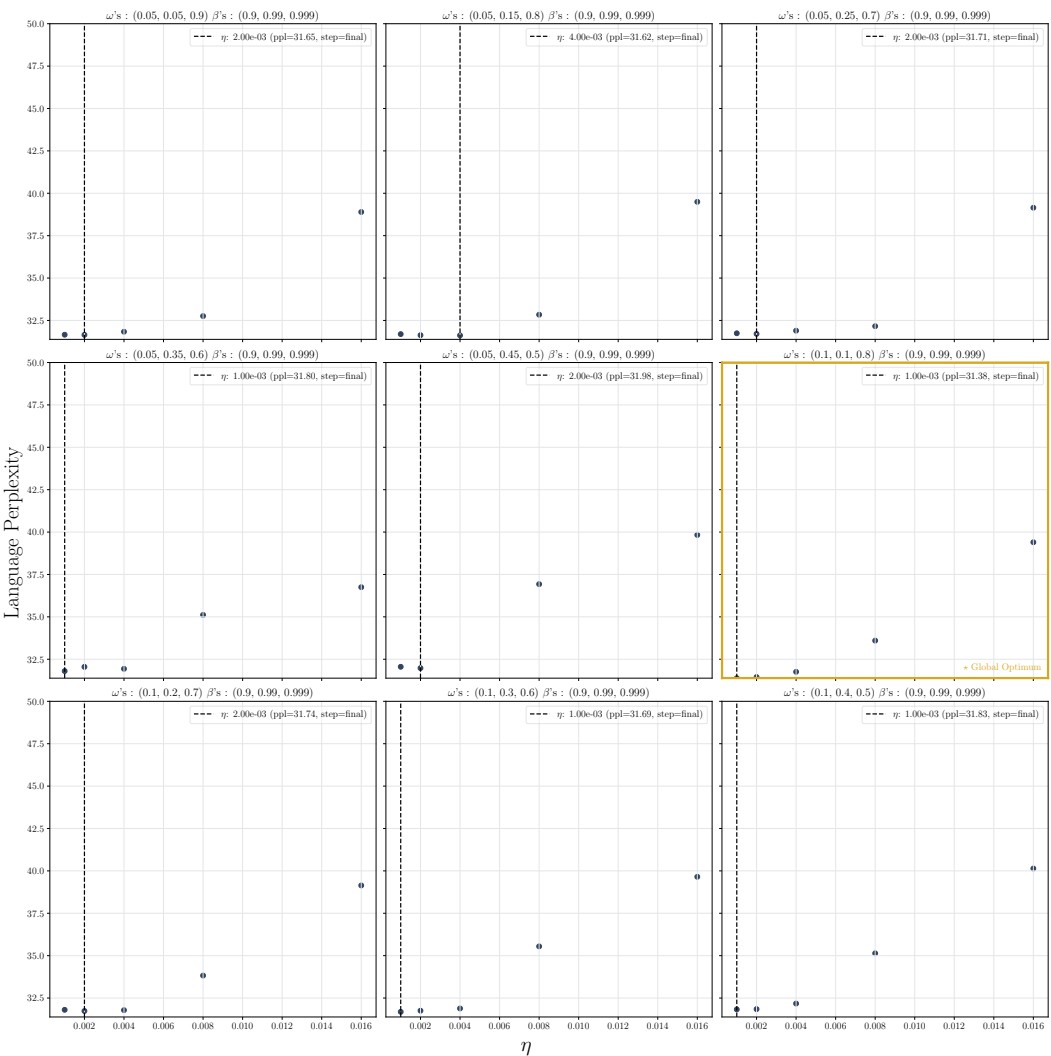

Figure 24: **Tuning** $N = 3$ `AggMo.` Optimal performance is found with a heavy bias towards the slowest momentum ($\omega_3 = 0.8$), confirming that long-horizon signal is the primary driver of performance in distributed settings. We select the optimum (marked in yellow) for scaling experiments.

# I LLM USAGE DECLARATION

As declared in the submission form, `LLMs` were used in this work in order to aid or polish writing and for retrieval and discovery of related work. We used `GPT-5` and `Gemini 2.5 PRO` primarily to abbreviate or rephrase text or to evaluate the clarity of our writing and provide guidance on areas of improvement. We also used the deep research feature present in both models in order to discover, but not describe or interpret, additional papers for our extended literature review in Appendix G. Finally, we used both models to generate plotting code and as general code assistants.

# J REPRODUCIBILITY

We provide the complete source code for our `mt_dao` framework, accompanied by detailed setup instructions, to ensure the reproducibility of our results.

**System & Dependencies:** The required environment, including specific versions of Ubuntu, CUDA, and Python, can be installed using the provided `system_setup.sh` and `install_env.sh` scripts. These scripts handle all necessary dependencies.

**Data:** We provide scripts to download and prepare the datasets. The distribution of data across clients for both IID and non-IID settings is managed through declarative YAML configuration files found in `mt_dao/conf/dataset/streams/`.

**Execution & Hyperparameters:** Example scripts are available for launching federated, centralized, and evaluation runs (e.g., `fed_125m_example.sh`, `cen_125m_example.sh`). To fully reproduce our paper's experiments, users can utilize the provided base launcher scripts and set the specific hyperparameters detailed in the paper. A concrete example of this process is available in `scripts/iclr_mt_dao`.

