# OpenReview forum: "MT-DAO: Multi-Timescale Distributed Adaptive Optimizers with Local Updates"
_ICLR.cc/2026/Conference — ICLR 2026 Poster_

### Official Review · Reviewer_KKBR · 2025-10-28

**Soundness:** 3
**Presentation:** 3
**Contribution:** 3
**Rating:** 8
**Confidence:** 3

**Summary:**

This paper introduces MT-DAO, a novel family of optimizers designed to bridge the performance gap between DDP and adaptive optimizers using low-frequency communication strategies (e.g., Local SGD). The authors identify "timescale mismatch" as the root cause of this gap and propose employing multiple slow and fast-moving first momenta to capture update dynamics across different time scales. In large-scale language model pretraining experiments, MT-DAO successfully eliminates the performance disparity with synchronous DDP and outperforms existing low-communication baselines in perplexity. Combining rigorous theoretical analysis, solid empirical validation, and practical significance, this work constitutes a major contribution to distributed deep learning optimization.

**Strengths:**

- Novelty: Identifies the performance gap between low-frequency communication strategies and DDP as stemming from temporal scale mismatch, and innovatively adapts multi-momentum strategies to the distributed setting.
- Empirical Rigor: Comprehensive experiments demonstrate MT-DAO’s reliability.
- Clarity and Readability: The paper is well-structured, clearly articulating motivation, methodology, and results.

**Weaknesses:**

Please refer to Questions.

**Questions:**

- The novelty of this paper lies in the introduction of a multi-momentum strategy. However, all experimental results were obtained under MT-DAO with N=1, meaning only a single momentum was used. I wonder whether increasing the value of N would lead to further improvements in model performance.
- If we set K=1, effectively turning MT-DAO into a DDP optimizer, will its performance surpass the DDP baseline? And does it hold any advantages over other DDP optimizers that adopt multi-momentum strategies, e.g. adEMAMix?

---

> ### Author Response · Authors · 2025-11-22
> **Reply**
>
> ## Dear Reviewer ${\color{brown}{\text{KKBR}}}$,
>
> We sincerely thank you for your thoughtful review and for recognizing the **novelty** of our multi-timescale approach, the **empirical rigor** of our validation, and the **clarity** of our presentation. We are encouraged by your assessment of MT-DAO as a "major contribution." Below we address your specific questions regarding the number of momenta ($N$) and the behavior at high communication frequency ($K=1$).
>
> ---
>
> ### **Q1 — “All results use (N=1). Would increasing (N) further improve performance?”**
>
> We have conducted extensive new ablations for $N \in \{2, 3\}$ across multiple model scales in **Appendix H.2**. As detailed in **Shared Concern 1 (SC1)**, the $N=1$ (Quasi-Hyperbolic) variant captures the vast majority of the performance gain with **zero memory overhead**.
>
> - **Mechanism:** As explained in **Appendix H.2**, the $N=1$ QH formulation effectively acts as a **two-timescale optimizer** by mixing the instantaneous gradient ($\beta=0$) with a slow EMA. This aligns with findings from [1,2] that a single slow momentum plus the current gradient is sufficient to approximate complex multi-momentum dynamics.
> - **Evidence:** Our new experiments in **Figure 13** and **Figure 14** show that increasing $N$ to 2 or 3 yields negligible perplexity differen compared to the $N=1$ baseline.
> - **Recommendation:** Given the marginal gains vs. the increased communication/memory cost of synchronizing extra buffers, we recommend $N=1$ as the robust, efficient default with $N>1$ reserved for situations where the bandwidth and memory constraints are sufficiently relaxed.
>
> ---
>
> ### **Q2 — “If (K=1) (DDP), will MT-DAO surpass DDP? Any advantage over DDP multi-momentum (e.g., AdEMAMix)?”**
>
> At **$K=1$**, MT-DAO effectively becomes similar to a DDP optimizer. While we expect it to perform comparably to strong DDP baselines (including AdEMAMix-DDP), potentially slightly underperforming, we do **not** expect it to surpass them in this regime, as the specific benefits of MT-DAO are designed to counteract the timescale mismatch inherent to **infrequent communication ($K \gg 1$)**.
>
> - **SGDM:** For SGD with Momentum, MT-DAO with $K=1$ is **mathematically equivalent** to standard DDP. The averaging of local momenta at every step is identical to computing momentum on the global averaged gradient while averaging parameters effectively averages their updates.
> - **Adaptive Optimizers:** For Adam/ADOPT, $K=1$ is **not** identical to standard DDP due to the non-linearity of the second moment update ($\mathbb{E}[g^2] \neq (\mathbb{E}[g])^2$). However, under IID data we would not expect this difference to be large.
>
> Prior literature suggests that the noise inherent in Local SGD methods ($K>1$) can sometimes act as an implicit regularizer [3], potentially improving generalization compared to fully synchronous large-batch training (DDP). Setting $K=1$ would remove this noise source. Consequently:
>
> 1. **Performance:** We expect MT-DAO ($K=1$) to perform similarly to DDP variants, but not to outperform them.
> 2. **Efficiency:** Running MT-DAO at $K=1$ incurs the communication overhead of synchronizing momenta without the communication savings of local updates.
>
> **Summary:** MT-DAO is explicitly designed to maintain DDP-level performance **despite** infrequent communication. If bandwidth allows for $K=1$, standard DDP is the natural choice due to having to synchronize only the gradient; MT-DAO's value proposition is enabling high performance when $K=1$ is infeasible.
>
> ---
>
> ### **Closing**
>
> Thank you once more for helping us clarify the scope of our contribution. We hope this response, alongside the **General Comment** and **Shared Concerns** above addresses your queries. We await any further questions and kindly ask you to consider raising your scores if these clarifications strengthen your assessment.
>
> — The Authors
>
> **References:**
>
> [1] Ma & Yarats, Quasi-hyperbolic momentum and Adam for deep learning
>
> [2] Pagliardini et al, The AdEMAMix Optimizer: Better, Faster, Older.
>
> [3] Lin et al., Don't Use Large Mini-Batches, Use Local SGD

---

> > ### Comment · Reviewer_KKBR · 2025-11-26
> > **Question**
> >
> > Thank you for the authors’ response. However, regarding Question 2, I still have some lingering concerns. A brief comparative experiment would help substantiate the current claim.

---

> ### Author Response · Authors · 2025-11-27
> **Empirical Results**
>
> ## **Re: Question 2 — “If (K=1) (DDP), will MT-DAO surpass DDP?”**
>
> We thank the reviewer for their continued engagement. As requested, we have conducted a complete set of ablations at the synchronous limit ($K=1$) for both $N=1$ and $N=2$ variants across two model scales. These new results are detailed in **Appendix H.6** and **Figure 18**.  Our experiments **confirm** our expectations, specifically, MT-DAO ($K=1$) closely trails, but slightly underperforms the equivalent fully synchronous DDP baselines. For example at the 125M scale MT-DAO ($K=1$) underperforms the baselines by approximately 0.28 to 0.33 perplexity points .
>
> We reiterate that our primary goal for MT-DAO is to reduce the performance gap of **communication-efficient** methods relative to DDP.  Furthermore, we did not expect MT-DAO to surpass DDP in the fully synchronous regime ($K=1$).
>
> Our method is specifically engineered to counteract the worker drift and noise compounding that arise in communication-efficient settings ($K \gg 1$). In the case of $K=1$, the problem MT-DAO solves (drift) does not exist. Furthermore, setting $K=1$ removes the implicit regularization effects of Local SGD[3], which may sometimes improve performance. Therefore, our theoretical prior is that MT-DAO should at best match, or slightly underperform, a well-tuned DDP baseline.
>
> ---
>
> ## **Empirical Results (Appendix H.6):**
>
> As shown in the top row of Figure 18, MT-DAO tracks the strong DDP baselines closely but consistently trails them by a small margin.
>
> - **QH-Variant ($N=1$):** At the 125M scale, MT-DAO ($K=1$) trails the fully synchronous `QHAdamW-DDP`baseline by approximately  $0.28$ perplexity points.
> - **AdEMAMix-Variant ($N=2$):** Similarly, the multi-momentum MT-DAO variant trails `AdEMAMix-DDP` by $0.33$ perplexity points.
> - **Insights:** The bottom row of Figure 18 empirically demonstrates the consequence of the non-linearity of the second moment update. As expected from Jensen's inequality ($\mathbb{E}[g^2] \geq (\mathbb{E}[g])^2$), we observe that the L2 norm of the second momentum for MT-DAO grows larger than that of DDP, for example they get approximately 38% larger for the 125M model in the $N=1$ case by the end of training. This inflated denominator results in a systematically smaller effective learning rate, explaining the slight lag in convergence.
>
> ---
>
> ## **Closing:**
>
> If bandwidth permits $K=1$, we recommend DDP as the default choice. MT-DAO’s value proposition is maintaining DDP-competitive performance in low-bandwidth settings where $K=1$ is not feasible. Thank you once more for your response. We await any additional feedback or questions.

---

### Official Review · Reviewer_CDpB · 2025-10-29

**Soundness:** 3
**Presentation:** 3
**Contribution:** 3
**Rating:** 8
**Confidence:** 4

**Summary:**

This paper proposes MT-DAO, a family of optimizers to stabilize and accelerate training in a communication-limited environment. The authors observed that when the synchronization interval K is large, a relatively large $\beta$ leads to training instability. They introduced the Quasi-hyperbolic method into training, provided convergence proofs, and experimentally validated its effectiveness.

**Strengths:**

- The paper has a clear structure with well-defined motivation
- Convergence proofs are provided, and experiments address the questions of interest

**Weaknesses:**

Most of my concerns are already mentioned in the Limitations of the paper itself.

- The scalability of MT-DAO. Other than the largest model is at the size of 720M, the experiments are conducted under four workers with 1 GPU per each.
- A possibly weak baseline with ADOPT instead of Adam. Similarly seen in Figure 2, Local SGD converges faster than MT-DAO-SGDM in this toy example.
- There is a suspicion of overclaiming because the paper only conducted experiments with N=1. When N=1, it degenerates into each worker storing momentum similar to AdEMAMix and combining updates like QHM.

**Questions:**

1. Could the choice of batch size impact the results? The paper chooses a fixed global batch size settings from [1], which is not in a Local SGD setting. According to [2], DiLoCo seems to tolerate a larger batch size than DDP for a better wall-clock time performance, which I think is pretty reasonable because it does local updates.
2. The paper seems not to discuss the choice of the weight parameter w, does it have a clear pattern of range or recommended value?

[1] Benchmarking optimizers for large language model pretraining
[2] Communication-Efficient Language Model Training Scales Reliably and Robustly: Scaling Laws for DiLoCo

---

> ### Author Response · Authors · 2025-11-22
> **Reply**
>
> ## Dear Reviewer ${\color{purple}{\text{CDpB}}}$
>
> Thank you for the review and for recognizing our paper’s **clear motivation**, **convergence analysis**, and **targeted experiments**. We address all your points below.
>
> ---
>
> ### **Weakness 1 — Scalability to larger models / hardware setting**
>
> **Yes; MT‑DAO scales:** As detailed in **Shared Concern(SC3)**, we report **1.3B-parameter results** trained for **80B** tokens with **AdamW** to demonstrate optimizer-agnostic scalability (**§5.5**, **App. H.1**).
>
> - **Performance:** MT-DAO  matches or exceeds **AdamW-DDP** on downstream few-shot tasks and tracks the performance of **QHAdamW-DDP.** Notably, our 1.3B model outperforms checkpoints from open-source baselines like **OLMo 2** [1] trained on similar budgets and approaches the performance of **SmolLM2** [2] despite training on **$3\times$ fewer tokens**.
> - **Efficiency:** It achieves this performance approximately **$50$% faster** in terms of wall-clock time than DDP on our 100 Gbit/s Ethernet setup.
> - **Hardware**: Our new large-scale experiments use 4 workers with 8 GPUs each, with DDP intra-node and MT-DAO cross-node. Our Batch $\times$ Worker sweeps test up to 8 workers showing benefits over the baseline.
>
> ---
>
> ### **Weakness 2 — More Optimizers and LocalSGD**
>
> - **Fig. 2 (Convex Toy):** We now explain in **§2.2** that this toy convex problem illustrates a specific trade-off: In the **low-$\beta$ regime** (top row), where the gradient consistently points towards the global minimum, standard momentum accelerates convergence; here, MT-DAO's mixing of a slow momentum effectively acts as a very slight damper, merely slowing down convergence. However, the critical insight lies in the **high-$\beta$ regime**, where standard Local SGD becomes unstable and fails to converge due to momentum drift over $K$ steps. MT-DAO remains stable in this regime. In **non-toy setup** (non-convex, high noise), MT-DAO improves performance as confirmed by Fig. 1 and our scaling experiments.
> - **Baselines:** We have **strengthened our baselines** to include standard AdamW. As detailed in **SC3** and **Appendix H.1**, we expanded our evaluation to include **AdamW** at the 1.3B scale. The new results (**Fig. 7** show that MT-DAO tracks **QHAdamW-DDP** and outperforms **Local AdamW** and **Nesterov** (outer) in downstream tasks, confirming that our gains hold for standard optimizers beyond ADOPT. (We originally used ADOPT to rigorously isolate the impact of $\beta_1$).
>
> ---
>
> ### **Weakness 3 — “Overclaiming with N=1”**
>
> We experimented $N \in \{2, 3\}$ across multiple scales (**App. H.2**), As detailed in **Shared Concern 1 (SC1)** above. The $N=1$ (Quasi-Hyperbolic) variant captures the performance gains with **zero memory overhead**.
>
> - **Mechanism:** The $N=1$ QH formulation acts as a **two-timescale optimizer** (instantaneous gradient + slow EMA), sufficient to approximate complex multi-momentum dynamics per findings in [3,4] (see **App. H.2**).
> - **Evidence:** New experiments (**Fig. 13, 14**) show negligible perplexity differences  for $N=2,3$ vs $N=1$. E.g., at 360M, QH-N=1 reaches 8.26 PPL vs 8.26 for N=2.
> - **Recommendation:** Given the marginal differences vs. increased comms/memory costs, we recommend $N=1$ as the robust, efficient default.
>
> ---
>
> ### **Q1 — “Does batch size change the picture?”**
>
> **Yes:** As detailed in **SC2** above, and **Appendix H.3**, our Batch $\times$ Worker sweeps reveal two key distinctions versus Local Adam.
>
> - **Robust Scaling:** MT-DAO maintains performance as worker count increases (decreasing per-worker batch), whereas Local Adam degrades significantly (**Fig. 15**), confirming slow momentum counteracts local noise. The optimal global batch size for MT-DAO also **increases** with worker count.
> - **Mechanism:** As derived in **App. F.1**, increasing $B$ reduces variance linearly, but increasing $\beta$ suppresses the **exponential drift** from local updates. MT-DAO exploits this lever, avoiding the diminishing returns of simply increasing $B$.
>
> ---
>
> ### **Q2 — “How to set the weight parameter $\omega$?”**
>
> Based on our extensive hyperparameter sweeps, we recommend **$\omega_1 \in [0.90, 0.99]$** with $\beta_1=0.999$. These values align with recommendations in [3,4]. Our methodology consistently tunes $\omega$ and the learning rate on a small proxy (16M) and transfers them directly to larger scales (up to 1.3B) without re-tuning, demonstrating transferability.
>
> ---
>
> ### **Closing**
>
> We are grateful for your constructive suggestions and look forward to any further feedback or requests. We also kindly ask you to consider increasing your scores if our improvements are satisfactory.
>
> — The Authors
>
> ### References:
>
> [1] OLMo Team, 2 OLMo 2 Furious
>
> [2] Allal et al., SmolLM2: When Smol Goes Big -- Data-Centric Training of a Small Language Model
>
> [3] Ma & Yarats, Quasi-hyperbolic momentum and Adam for deep learning
>
> [4] Pagliardini et al, The AdEMAMix Optimizer: Better, Faster, Older

---

### Official Review · Reviewer_BGje · 2025-10-30

**Soundness:** 2
**Presentation:** 2
**Contribution:** 2
**Rating:** 4
**Confidence:** 3

**Summary:**

The authors hypothesize that the gap between the fully synchronous DDP and sparse comm strategies is due to a time scale mismatch; that is, the optimizer's fast moving momentum decays too quickly between synchronization steps, causing replicas to diverge. To remedy this, they propose a bandwidth-efficient distributed data parallel training method for LLMs, which incorporates multiple fast and slow moving momentums. They show that using this approach they can close the gap between DDP and sparse comm methods, and can preserve mutual information better between model replicas.

**Strengths:**

The paper addresses an important question. The convergence gap between infrequent communication strategies and synchronous DDP settings is a glaring problem. I like the authors approach to the problem which is formulated around the fact that this is due to a time scale mismatch between synchronization intervals.

**Weaknesses:**

The narrative of the paper is a bit unclear to me. The method is proposed as a generic plugin across many optimizers and the authors even provide algorithms for SGDM and Adam. However, the experiments are only demonstrated on ADOPT. This also harms the claim that this method is useful for optimizers that use multiple first order momentums, however, as far as I am aware, ADOPT only use a single first order moment? I invite the authors to correct me if I am missing something.

**Questions:**

1. Why are the experiments only conducted on Adopt? In my opinion, this significantly harms your claims.
2. Infrequent communications mean that each replica is effectively seeing a smaller batch size (compared to the global batch size across replicas). In such cases, it has been previously shown that increasing the betas work robustly [1]. How does your work differ from the insights given in such works.
3. Can you compare your method with more recent methods such as streaming DiLoCo? I think such comparisons are crucial to validate the efficacy of the proposed work.
4. When tuning the (beta, w), did you conduct a grid search? How well do these parameters scale across model size?
5. What are the hyper parameters used in Fig. 5?

[1] - Small Batch Size Training for Language Models: When Vanilla SGD Works, and Why Gradient Accumulation Is Wasteful

---

> ### Author Response · Authors · 2025-11-22
> **Reply [1/2]**
>
> ## Dear Reviewer **${\color{red}{\text{BGje}}}$,**
>
> Thank you for the careful review. We revised the paper to address your points directly:
>
> ---
>
> ## **Weaknesses  1 — “Experiments are only demonstrated on ADOPT”**
>
> We now **go beyond ADOPT** and show MT‑DAO works as intended for **AdamW** (up to **1.3B parameters**), the standard optimizer for LLM pre-training. The conclusions mirror ADOPT: $N=1$ quasi‑hyperbolic MT‑DAO closes most of the DDP gap while substantially **reducing synchronizations**.
>
> - **AdamW (new §5.5; App. H.1/H.2):** At fixed tokens, **MT‑DAO‑AdamW** outperforms AdamW‑DDP and Prior Comms-efficient baselines and **approaches QHAdamW‑DDP** with fewer comms. We include a downstream task results (**Fig. 7/§5.5**) showing MT-DAO matching or exceeding baselines on downstream tasks.
>
> ---
>
> ## **Weaknesses  2 — Using an optimizer with a single first momentum**
>
> We clarify that MT-DAO claims that utilizing **multiple timescales**—not necessarily multiple momentum *buffers*—is useful for communication-efficient optimizers.
>
> - **Mechanism:** As detailed in **SC1**, even in the $N=1$ variant, the gradient ($g_t$) acts as a **fast leg** ($\beta=0$) while ($u_t$) provides a **slow EMA** ($\beta \approx 0.999$). This formulation is mathematically equivalent to an $N=2$ system where the fast momentum has $\beta=0$. Thus, $N=1$ already mixes two time‑scales (fast+slow) with only **one EMA buffer**—providing a strictly superior trade-off between performance and memory cost.
> - **Diminishing Returns:** Consistent with prior work [7], our new ablations (**App. H.2, Fig. 13, 14**) show that increasing $N$ to 2 or 3 yields diminishing returns. For example, at 360M, QH-$N=1$ reaches **8.26** PPL vs **8.26** for standard $N=2$.
>
> Please read the **Shared Concern (SC1)** in our reply to the general comment above for further details.
>
> ---
>
> ## **Weaknesses  3 — MT-DAO for optimizers with multiple first momentums**
>
> We emphasize that we do not claim our method is exclusively "useful for optimizers that use multiple first order momentums." Rather, we claim that **injecting multiple timescales** is useful for **communication-efficient Local SGD/Adam-style** optimizers. Our baselines (Local Adam, Local ADOPT) are standard single-momentum methods, and we show that modifying them with multi-timescale dynamics (via MT-DAO) strictly improves performance. As stated above, we have now added results for more than one momentum to address your concerns.
>
> ---
>
> ## **Q1 —  “Why only ADOPT?”**
>
> We initially used ADOPT because:
>
> - ADOPT let us isolate **first‑moment time‑scales** (keep ($\beta_1\le\beta_2$) with the convergence rate unaffected by $\beta_2$ before expanding to other optimizers.
> - **We now show AdamW up to 1.3B** results; patterns match ADOPT (new §5.5; App. H).
>
> ---
>
> ### **Q2 — Prior work shows larger $\beta$ is robust at small batch. How is this different?**
>
> Our contribution accounts for the **time-induced effect** of local updates ($K \gg 1$), which is non-existent in DDP, and which requires setting a much larger $\beta$ than would be necessary to compensate for using small batch sizes in DDP.
>
> - **Variance Analysis**: Increasing batch size ($B$) reduces per‑step noise linearly ($\mathrm{Var}[\hat g_t] \propto 1/B$) [1], as does increasing $\beta$, which justifies using high $\beta$ for small batches in DDP. However, in local-update regimes, the variance of the momentum after $K$ steps is dominated by the exponential decay term $(1-\beta^{2K})$ (SC2; App. F.1):
>
>     $$\operatorname{Var}(u_{t+K}) \approx \frac{1-\beta}{1+\beta} \underbrace{(1-\beta^{2K})}_{\text{Exponential Decay}} \cdot \frac{\sigma_m^2}{B}$$
>
> - **Novel Insight:** The drift of workers is not solely an effect of small batch size; it depends critically on the interplay between $\beta$ and $K$. MT-DAO provides the insight that $\beta$ must be increased sufficiently to suppress this exponential term.
> - **Evidence:** Our new **Batch $\times$ Worker sweeps** (**Fig. 15**) confirm that simply increasing batch size is inefficient for removing the $K$-step penalty, MT-DAO outperforms the Local Adam baseline even with exceedingly large batch sizes.
>
> See SC2 for a more detailed response on the point of the Batch $\times$ Worker sweeps, in our reply to the general comment above. **Our reply continues below.**

---

> ### Author Response · Authors · 2025-11-22
> **Reply [2/2]**
>
> ---
>
> ### **Q3 — Compare with Streaming‑DiLoCo; performance & memory**
>
> We did not initially include Streaming‑DiLoCo because it aims for uniform bandwidth utilization rather than total reduction, and its streaming mechanism is an orthogonal addition to the contributions we make in our work. However, we now include a **Streaming MT-DAO** variant for completeness and compare against Streaming‑DiLoCo.
>
> - **Performance:** In **App. H.4** (**Fig. 16**), we show that under matched token budgets (and a 360M model size), **Streaming MT-DAO** outperforms standard Streaming DiLoCo (**8.35 vs 8.69 PPL**). This confirms that synchronizing the inner momentum state (MT-DAO) provides superior stability to the outer-optimizer approach (DiLoCo).
> - **Memory:** MT-DAO ($N=1$ with FedAvg outer) stores `Params + Inner Momentum`, saving one model-sized buffer compared to DiLoCo (Nesterov), which stores `Params + Inner Momentum + Outer Momentum`.
>
> ---
>
> ### **Q4 — Grid-Search for $\beta,\omega$**
>
> - **Grid Search:** Our initial $(\beta, \omega)$ sweeps were presented in **Figure 4** and **Figure 8**. We have now added further fine-grained sweeps for $\omega$ in **Appendix H.2** for $N=1$ and multiple $N$ variants. We find a **broad plateau** of effective settings where $\omega \in [0.90, 0.99]$ consistently works well for $\beta=0.999$.
> - **Scaling:** There is no theoretical motivation to re-tune these coefficients with model scale; they govern the temporal dynamics (steps, horizon) rather than parameter count. This methodology—fixing regularization parameters and tuning only LR with model size—is standard practice in **AdamW [7]**, **AdEMAMix [8], amongst others [**6]**;** none of which recommend retuning momentum or linear combination coefficients per scale. We **do not retune** $(\beta,\omega)$ with model size. We tune on a 16M proxy and transfer settings directly to 125M $\to$ 720M $\to$ 1.3B (**new §5.5**). Crucially, despite using transferred hyperparameters, MT-DAO **outperforms** the rigorously tuned DDP baselines for these parameters from [6], which tuned hyperparameters specifically for comparing optimizers for LLM pre-training.
>
> ---
>
> ### **Q5 — What are the hyper‑parameters in Fig. 5?**
>
> - $\beta_1=0.999$, $\omega_1=0.98$; LR = $2.8 \times$ base LR; same $\beta_2$, scheduler, and clipping as baselines; $K$ as indicated. We have added the $\beta_1=0.999$, $\omega_1=0.98$ to the caption.
>
> ---
>
> ## **Closing**
>
> We thank you for your questions and helping us improve our work. We would appreciate any further feedback and we are happy to provide any additional information. If our response has sufficiently addressed your concerns, we kindly ask you to consider raising your score.
>
> ---
>
> **References**
>
> [1] McCandlish et al., An Empirical Model of Large‑Batch Training
>
> [2] Stich, Local SGD Converges Fast and Communicates Little
>
> [3] Lin et al., Don’t Use Large Mini‑Batches, Use Local SGD
>
> [4] Charles et al., Communication‑Efficient Language Model Training Scales Reliably and Robustly: Scaling Laws for DiLoCo
>
> [5] Ma & Yarats, Quasi-hyperbolic momentum and Adam for deep learning
>
> [6] Semenov et al., Benchmarking Optimizers for Large Language Model Pretraining
>
> [7] Loshchilov and Hutter, Decoupled Weight Decay Regularization
>
> [8] Pagliardini et al, The AdEMAMix Optimizer: Better, Faster, Older

---

### Official Review · Reviewer_uewt · 2025-11-04

**Soundness:** 3
**Presentation:** 4
**Contribution:** 3
**Rating:** 6
**Confidence:** 3

**Summary:**

This paper presents a unified perspective on momentum-based distributed optimizers. A novel multiple time-scale distributed optimizer (MT-DAO) with convergence guarantees is proposed that maintains momentum at different time scales. Results on language modelling task with 3 different sizes (16M, 125M and 720M) demonstrate that MT-DAO outperforms prior approaches and also match DDP.

**Strengths:**

- Clean framework to develop momentum-based distributed optimizers, nice analytical studies.
- Strong set of language modelling experiments with 3 sizes upto 720M using modern optimization tools including completeP for LR transfer, WSD scheduler, standard data mixtures, etc - great job!

**Weaknesses:**

See questions below.

**Questions:**

- The MT-DAO approach is proposed and motivated with multiple fast/slow-moving momenta but experiments show results with N=1, why?
- Following up on above, the proposed framework looks like a generalization of AdEMAMix optimizer -- how does it directly compare for LLM training in DDP? Is it feasible to match the performance of AdEMAMix optimizer (and other recent matrix-based optimizers such as Muon) with much less communication steps?
- How does the proposed approach compare with DiLoCo in terms of performance and memory requirement? Moreover, DiLoCo supports complex adaptive rules in the inner optimization and not just limited to momentum, so was wondering if MT-DAO supports that and how it affects theoretical results.
- How critical is gradient clipping for this approach to work well or can it work without clipped gradients as well? Does DDP baseline also use the same clipping values?

---

> ### Author Response · Authors · 2025-11-22
> **Reply**
>
> ## Dear Reviewer **${\color{magenta}{\text{uewt}}}$,**
>
> Thank you for highlighting our paper’s **clean framework**, **analytical studies**, and **modern LLM experiments.** Please find our answers below.
>
> ---
>
> ### **Q1 — “Experiments use $N{=}1$ only — why?”**
>
> We addressed the "N=1" concern by running variants with $N=2$ and $N=3$ across multiple model scales in **Appendix H.2**.
>
> - **Why $N{=}1$:** As detailed in **Shared Concern 1 (SC1)** and **Appendix H.2**, the **$N=1$ Quasi-Hyperbolic (QH)** formulation effectively acts as a **two-timescale optimizer** by mixing the instantaneous gradient ($\beta=0$, fast) with a slow EMA ($\beta \approx 0.999$). This is mathematically equivalent to an $N=2$ formulation where the fast momentum has $\beta=0$.
> - **Evidence:** Our new ablations (**Fig. 13, 14**) show that increasing $N$ to 2 or 3 yields **diminishing returns**. For example, at the 360M scale, QH-$N=1$ reaches **8.26** PPL vs. **8.26** for standard $N=2$.
> - **Conclusion:** The $N=1$ QH variant can capture multi-timescale benefits without the memory or communication cost of extra buffers. We recommend it as the efficient default.
>
> Please also read **SC1** in the general comment above for full details.
>
> ---
>
> ### **Q2 — “Relation to AdEMAMix and Muon under DDP?”**
>
> - **AdEMAMix:** We have added direct comparisons to **AdEMAMix** (which maintains two momentum states) in the low-communication setting (**Appendix H.2**).
>     - **Result:** At the tested model scales (16M–360M), MT-DAO ($N=1$) **slightly trails** the performance of AdEMAMix ($N=2$). However, consistent with the trends observed in our other scaling experiments (where performance gaps consistently narrow as model size increases, e.g., **Fig. 5** and **Fig. 7**), we anticipate this marginal gap to shrink at larger scales (720M and 1.3B).
>     - **Efficiency:** Crucially, MT-DAO ($N=1$) achieves this competitive performance with **significantly lower memory and communication costs** than AdEMAMix ($N=2$), which requires storing an additional momentum buffer (+1$\times$ Model State) and synchronizing every step.
> - **Muon:** For matrix-based optimizers like **Muon**, the Newton-Schulz preconditioning is typically applied to the momentum term. This means our multi-timescale considerations regarding momentum variance and drift apply directly to the underlying state. While we leave the full theoretical analysis of preconditioned local updates for future work, our framework is compatible with these methods.
>
> ---
>
> ### **Q3 — DiLoCo Performance and Supporting Inner Optimizers**
>
> - **Performance & Mechanism:** DiLoCo is effectively **Nesterov Momentum** combined with a **purely** local update scheme (optimizer states are never synced). Our main paper compares against a **Nesterov** baseline that *does* synchronize states (**Fig. 5, New Fig.7**) since synchronizing optimizer states is known to improve performance and we want our comparison to be fair between the quasi-hyperbolic formulation and Nesterov. MT-DAO outperforms it in terms of perplexity.
> - **New Evidence:** Our new **1.3B parameter** results (**§5.5**) confirm that MT-DAO outperforms Nesterov-based baselines in performance. Furthermore, we have added a **Streaming MT-DAO** comparison in **Appendix H.4** (**Fig. 16**), showing that when we apply our method to the streaming setting, it outperforms standard Streaming DiLoCo (**8.35 vs 8.69 PPL**) under matched budgets.
> - **Inner Rules:** Yes, MT-DAO supports complex inner rules. The framework applies to the momentum handling; any adaptive rule can be composed with the multi-timescale update.
>
> ---
>
> ### **Q4 — Gradient Clipping and Importance**
>
> MT-DAO does not inherently require gradient clipping to function and does not increase the necessity of gradient clipping compared to previous works. The necessity of clipping is a property of the inner optimizer. We include clipping as a common practice for the pre-training of large models.
>
> - **Policy parity.** We use **identical clipping thresholds** across **MT‑DAO** and all baselines at each model size:
>     - **ADOPT‑based** runs: **clip‑norm = 1.0** (rarely active due to inner clipping lambda).
>     - **AdamW‑based** runs: we use the values recommended by [2] (e.g., 1.0 at 16M, 0.5 at 125, 0.25 at 360M, 0.1 at ≥ 720M).
> - **Ablation.** We performed a clipping ablation ($\rho = \infty $) across scales (**Appendix H.5, Fig. 17**). We do find that stability is **scale-dependent**: 16M models diverge without clipping, but 125M/360M models train stably without it.
>
> ---
>
> ### **Closing**
>
> We appreciate your instructive comments and questions. We await further feedback and inquiry. If these additions address your concerns, we kindly ask you to consider raising your scores.
>
> ---
>
> ### **References**
>
> [1] Semenov et al., Benchmarking Optimizers for Large Language Model Pretraining

---

> ### Author Response · Authors · 2025-12-02
>
> As detailed in Shard Concern 3 Part 2 above, as a further extension to our set of inner optimizers, we have successfully applied MT-DAO to **Muon**, which utilizes Newton-Schulz iterations for update orthogonalization. As the preconditioning is typically applied to the momentum term, our multi-timescale analysis regarding variance and drift applies directly to the underlying state before preconditioning.
>
> - **Experiments:** We conducted experiments across **16M, 125M, and 360M** model scales (Fig. 19). Results shown in **App.H.7**  demonstrate that **MT-DAO significantly outperforms the base Muon-DDP baseline,** by upwards of 9% at the 360M scale, and matches or exceeds **QHMuon-DDP** (they are generally within $1$% of each other). This confirms that the benefits of multi-timescale optimization transfer effectively to Muon-like methods.
> - **Insight:** Our results are the first to indicate that using an **independent weight for the gradient** is beneficial for Muon. Standard Muon couples the gradient weight to the momentum $\beta$. Our sweeps show this is suboptimal when $\beta$ is high; MT-DAO allows for a high $\beta$ (to handle noise) while maintaining a high weight on the current gradient (for reactivity), yielding superior performance to standard Muon.

---

### Author Response · Authors · 2025-11-22
**General Rebuttal**

## **Dear Reviewers, ACs, and PCs,**

We thank the reviewers for their constructive feedback and recognition that we tackle an “important question”~(${\color{red}{\text{BGje}}}$**)** with a "well-defined motivation" (${\color{purple}{\text{CDpB}}}$) using a "clean framework” and "nice analytical studies" (${\color{magenta}{\text{uewt}}}$) bringing a "major contribution" (${\color{brown}{\text{KKBR}}}$) to the field.

We significantly expanded our evaluation with: **1.3B AdamW runs**, **N-momentum ablations**, **Streaming‑DiLoCo** comparisons, and an exploration of the **impact of batch size.**

---

## **Summary of Revisions and new results**

**[${\color{magenta}{\text{uewt}}}$, ${\color{brown}{\text{KKBR}}}$] Ablating $N$ (Quasi-Hyperbolic vs. Standard).** We addressed the "N=1" concern by running variants with $N=2$ and $N=3$ across multiple model scales.

- **Finding:** Increasing $N$ yields diminishing returns, consistent with prior findings [6, 7].
- **Insight:** As detailed in **Shared Concern 1~(SC1)**, the **$N=1$ Quasi-Hyperbolic (QH)** variant functions as a **two-timescale** optimizer. It secures the benefits of multi-momentum tracking **without the memory cost** of extra buffers required by $N>1$ methods.

**[${\color{purple}{\text{CDpB}}}$, ${\color{red}{\text{BGje}}}$] Scaling to 1.3B Parameters (AdamW).** We trained a **1.3B parameter model** on **80B tokens** using **AdamW** to demonstrate optimizer-agnostic scalability (see **SC3**).

- **Results:** MT-DAO matches or exceeds **AdamW-DDP** on challenging downstream few-shot tasks and approaches the performance of **QHAdamW-DDP** [3]. Our 1.3B model outperforms checkpoints from open-source baselines like **OLMo 2**[1] trained on similar budgets and approaches the performance of **SmolLM2** [2] despite training on **$3\times$ fewer tokens**.
- **Efficiency:** It achieves this DDP-competitive performance approximately **$50$% faster** in terms of wall-clock time than DDP on our 100 Gbit/s Ethernet setup.

**[${\color{red}{\text{BGje}}}$, ${\color{magenta}{\text{uewt}}}$] Expanded Baselines and Optimizers.**

- **Optimizers:** We added **AdEMAMix** ($N=2$) [6] and **AggMo** ($N=3$) [7] comparisons. MT-DAO closely approaches the perplexity of these optimizers while reducing communication overhead.
- **Streaming:**  We added a **Streaming MT-DAO** variant including: strided fragments and overlapping communication-computation with blended updates. We emphasize that streaming is **orthogonal** to our proposed inner optimizer changes. Under matched token budgets at the 360M scale, it outperforms standard **Streaming DiLoCo** [5] in perplexity by 3% following the hyperparameter recommendations of [5]. Furthermore, **Streaming MT-DAO** comes within $1$% of the perplexity of standard MT-DAO for models larger than $16$M while providing a far more uniform usage of bandwidth.

**[${\color{red}{\text{BGje}}}$, ${\color{purple}{\text{CDpB}}}$] Batch Size vs. Momentum Theory.** We addressed questions regarding the impact of batch size on inter-worker update variance.

- **Theoretical Insight:** As explained in **SC2**, precise control of $\beta$ is far more important in communication-efficient settings due to the *exponential time-decay* of the momentum, absent from DDP settings.
- **Empirics:** Our new **Batch $\times$ Worker sweeps** (Fig. 15) demonstrate that MT-DAO effectively counteracts using smaller local batch sizes. Furthermore, we observe that the **optimal global batch size increases**, as observed by [9].

---

### **Roadmap of Revisions**

- **§2.3:** Clarified the multi-timescale nature of $N=1$ (Quasi-Hyperbolic).
- **§5.5 & Fig. 7:** Added 1.3B AdamW perplexity and Downstream Task results.
- **Appendix H:** Added N-ablations ($N=2,3$) and Batch $\times$ Worker sweeps.
- **Appendix F:** Formalized the variance reduction derivation ($\beta$ vs. $B$).
- **Appendix H:** Added AdEMAMix, AggMo, and Streaming DiLoCo comparisons.

---

### **Closing**

In total, we have conducted **235 new experiments**, comprising **5** very large-scale training runs, **85** baseline comparisons/ablations, and **145** tuning trials. All revisions are highlighted in ${\color{blue}{\text{blue}}}$. We believe our changes directly address the reviewers' concerns and would welcome further **feedback** or **inquiry**.

— The Authors

### **References**

[1] OLMo Team, 2 OLMo 2 Furious

[2] Allal et al., SmolLM2: When Smol Goes Big…

[3] Ma & Yarats, Quasi-hyperbolic momentum and Adam for deep learning

[4] Jordan, Muon: An optimizer for hidden layers in neural networks

[5] Douillard et al., Streaming DiLoCo with overlapping communication…

[6] Pagliardini et al, The AdEMAMix Optimizer…

[7] Lucas et al., Aggregated Momentum: Stability Through Passive Damping

[8] Marek et al., Small Batch Size Training for Language Models…

[9] Charles et al., Communication-Efficient Language Model Training Scales Reliably and Robustly…

---

> ### Author Response · Authors · 2025-11-22
> **Replies to Shared Concerns (SCs)**
>
> ## **SC1 — Multi-Momentum vs. $N=1$~(${\color{magenta}{\text{uewt}}}$, ${\color{purple}{\text{CDpB}}}$,${\color{brown}{\text{KKBR}}}$, ${\color{red}{\text{BGje}}}$)**
>
> A recurring question was why we only experiment with $N=1$. We clarify that the **Quasi-Hyperbolic (QH)** formulation ($N=1$) effectively operates as a **two-timescale optimizer**, mixing a slow EMA (large $\beta$) with a fast gradient leg. It behaves analytically as a two-timescale optimizer without the memory cost incurred by $N>1$ optimizers. As formalized in **Appendix D**, the instantaneous gradient $g_t$ acts as the "fastest" possible momentum (being equivalent to an EMA with $\beta=0$).
>
> - **Experiments with $N>1$:** We have conducted extensive experiments (**Appendix H.2**) varying $N$:
>     - **Finding:** Generally, we find that the quasi-hyperbolic variant of any optimizer with $N$ momenta performs very close to an $N+1$ optimizer. For example, at the 360M scale, Quasi-Hyperbolic MT-DAO ($N=1$) reaches a validation perplexity of **8.26**, effectively tied with the **8.26** achieved by Non-Quasi-Hyperbolic MT-DAO ($N=2$) (**Fig. 13**), while reducing communication volume and memory.
>     - **Insight:** This is consistent with AdEMAMix [6], which finds that "the final loss for $\beta_1=0$ and $\beta_1 = 0.9$ are very similar—$\beta_1=0.9$ yielding slightly better results for smaller batch sizes.".
> - **Memory Efficiency:** Unlike $N>1$ Nesterov-based methods, which require storing an outer momentum buffer ($+1\times$Model State), **MT-DAO ($N=1$) incurs zero memory overhead** over standard DDP.
> - **Recommendation:** We generally recommend using MT-DAO (N=1) with $\omega_1 \in [0.9,0.99]$  and $\beta_1 = 0.999$ as a default. We advise reaching for the diminishing returns provided by $N>1$ only when bandwidth and memory requirements are sufficiently relaxed.
>
> ## **SC2 — Interactions between Batch Size ($B$) vs. Momentum ($\beta$)~**(${\color{red}{\text{BGje}}}$, ${\color{purple}{\text{CDpB}}}$)
>
> We share our novel insights on the interactions between batch size, $\beta$ and the variance of the momentum.  We start from the variance of the momentum after $K$ local steps (derived in **Appendix F.1 with the batch-size adjusted version in Eq.7**):
>
>  $$\operatorname{Var}(u_{t+K}) \approx \frac{1-\beta}{1+\beta} \underbrace{(1-\beta^{2K})}_{\text{Exponential Decay}} \cdot \frac{\sigma_m^2}{B}$$
>
> - **Mechanism:** While $B$ scales the variance linearly ($1/B$), the term involving $\beta$ and $K$ acts exponentially. As $K$ increases (sparse communication), standard momenta decay rapidly ($\beta^{2K} \to 0$), causing the variance to increase.
> - **Novel Insight:** Previous work [8] which has investigated increasing $\beta$ to compensate for an insufficient batch size for DDP does not have to account for this time component. We provide the **unique insight** that $\beta$ should be increased sufficiently to reduce variance under *exponential time-decay* ($K \gg 1$), which leads to our much higher recommendations for $\beta_1$ compared to DDP settings.
> - **Example:** By utilizing a high-$\beta$ slow component (e.g., $\beta=0.999$) with $K=32$, we suppress momentum variance by roughly **32,000$\times$**. A similar increase in batch size would likely move far beyond compute-optimal batch sizes.
> - **Robust Scaling:** **Fig. 15 (App. H.3)** confirms two distinct scaling behaviors:
>     1. **Robustness to Workers:** MT-DAO scales better with the number of workers ($M$) than baselines. For a global batch size $B_G$ as $M$ increases (and per-worker batch $B_{G}/M$ decreases), MT-DAO maintains performance while Local Adam degrades, proving that the slow momentum effectively mitigates the increased local noise.
>     2. **Optimal Batch Size Shifts:** We observe that the optimal global batch size for MT-DAO *increases* with the number of workers. This corroborates findings from [9], which indicate that communication-efficient optimizers can tolerate (and benefit from) larger critical batch sizes than standard DDP.  Furthermore, MT-DAO improves performance over Local Adam even in **exceedingly large-batch** settings where update noise is minimal.
>
> ## **SC3 Part 1 — Robustness Across Optimizers & Scales**~(${\color{red}{\text{BGje}}}$,${\color{purple}{\text{CDpB}}}$)
>
> - **AdamW at Scale:**  We have added 16M,125M, 360M, and 1.3B parameter experiments using AdamW (Fig. 7, App. H.1). MT-DAO matches or exceeds AdamW-DDP and closely tracks QHAdamW-DDP on downstream tasks for the 1.3B model, improving average accuracy on average across tasks relative to baselines. Our 1.3B checkpoints outperform OLMo 2 [1] checkpoints trained on comparable budgets and approach SmolLM2 [2] performance, validating the quality of the resulting model.
> - **More Optimizers:** We added comparisons against **AdEMAMix** and **AggMo** ($N=3$) in DDP and low-comms settings (**App. H.2**). MT-DAO consistently approaches the token-efficiency of these optimizers while reducing communication overhead.

---

> ### Author Response · Authors · 2025-12-02
> **Replies to Shared Concerns Part 2**
>
> ## SC3 Part 2 — Efficacy with Newton-Schulz Optimizers (Muon) (${\color{magenta}{\text{uewt}}}$,${\color{red}{\text{BGje}}}$,${\color{purple}{\text{CDpB}}}$)
>
> As a further extension to our set of inner optimizers, we have successfully applied MT-DAO to **Muon**, which utilizes Newton-Schulz iterations for update orthogonalization. As the preconditioning is typically applied to the momentum term, our multi-timescale analysis regarding variance and drift applies directly to the underlying state before preconditioning.
>
> - **Experiments:** We conducted experiments across **16M, 125M, and 360M** model scales (Fig. 19). Results shown in **App.H.7** demonstrate that **MT-DAO significantly outperforms the base Muon-DDP baseline,** by upwards of 9% at the 360M scale, and matches or exceeds **QHMuon-DDP** (they are generally within $1$% of each other). This confirms that the benefits of multi-timescale optimization transfer effectively to Muon-like methods.
> - **Insight:** Our results are the first to indicate that using an **independent weight for the gradient** is beneficial for Muon. Standard Muon couples the gradient weight to the momentum $\beta$. Our sweeps show this is suboptimal when $\beta$ is high; MT-DAO allows for a high $\beta$ (to handle noise) while maintaining a high weight on the current gradient (for reactivity), yielding superior performance to standard Muon.

---

### Author Response · Authors · 2025-12-03
**Final Comment Part 1/3**

Dear AC, PC, and Reviewers,

We are sincerely grateful for the reviewers' feedback and have revised our work to address all of their concerns. During the rebuttal phase, we conducted **235 new experiments,** including new baseline comparisons and ablations. For ease of parsing the rebuttal, we now present a summary of the reviewers' points and how we addressed them.

---

## **Part 1: Resolution of General Concerns (GC)**

### GC1

Raised by: ${\color{magenta}{\text{uewt}}}$, ${\color{purple}{\text{CDpB}}}$, ${\color{brown}{\text{KKBR}}}$, ${\color{red}{\text{BGje}}}$

- **Concern:** Is $N=1$ sufficient? Does it truly capture multi-timescale dynamics?
- **Answer:** We clarified that the **Quasi-Hyperbolic ($N=1$)** variant is analytically a **two-timescale optimizer,** as originally formulated in **Appendix D**, mixing a fast instantaneous gradient ($\beta=0$) with a slow EMA ($\beta \approx 0.999$). This captures gradient information across multiple timescales without the overhead of multiple momentum buffers.
- **Evidence:** New ablations for $N \in \{1, 2, 3\}$, presented in App. H.2, across scales show negligible perplexity gains for $N>1$. For example, at 360M, $N=1$ achieves **8.26 PPL**, tying the **8.26 PPL** of $N=2$, but uses significantly less memory. We therefore recommend $N=1$ as the efficient default. Our findings are in agreement with AdEMAMix [2], whose authors find that "the final loss for $\beta_1=0$ and $\beta_1 = 0.9$ are very similar—$\beta_1=0.9$ yielding slightly better results for smaller batch sizes.", where $\beta_1$ refers to the beta of their fast momentum and their slow first momentum uses a $\beta$ of $0.999$ or $0.9999$.

### GC2

Raised by: ${\color{purple}{\text{CDpB}}}$, ${\color{red}{\text{BGje}}}$, ${\color{magenta}{\text{uewt}}}$

- **Concern:** Is the method robust beyond ADOPT and small scales?
- **Answer:** We trained a **1.3B parameter model** on **80B tokens** using **AdamW** to demonstrate optimizer-agnostic scalability. See Fig. 7 and App. H.1.
- **Evidence for AdamW:** MT-DAO matches **AdamW-DDP** on downstream tasks and approaches the equivalent DDP optimizer (**QHAdamW-DDP**) while reducing wall-clock time by $\approx 55$%. It also outperforms open-source baselines (OLMo 2) trained on similar budgets. We also added **AdEMAMix** ($N=2$) and **AggMo** ($N=3$) baselines; MT-DAO approaches their perplexity with strictly lower memory/communication overhead.
- **Evidence for Muon:** We have also conduced experiments using Muon as the inner optimizer, our results in App. H.7 show that MT-DAO significantly outperforms Muon-DDP in terms of perplexity (e.g., by $9$% at a 360M scale) and matches or exceeds QHMuon (within $1$%). To the best of our knowledge, our work is also the first to demonstrate that a quasi-hyperbolic formulation of Muon where the weight of the gradient is a free parameter can outperform standard Muon with Nesterov.

### GC3

Raised by: ${\color{red}{\text{BGje}}}$, ${\color{purple}{\text{CDpB}}}$

- **Concerns: H**ow does batch size change the picture? How do our findings differ from previous work showing that increasing $\beta$ can compensate for the increased variance of small batch sizes in a DDP regime?
- **Answer:** We derived the variance of momentum after $K$ local steps (**App. F.1**). In sparse-comms regimes ($K \gg 1$), the exponential time-decay of the momentum $(1-\beta^{2K})$ requires strictly higher $\beta$ values than would be necessary to constrain update variance following a decrease in batch size in a DDP regime.
- **Evidence:** Our **Batch $\times$ Worker sweeps** (**Fig. 15**) show that:
    - MT-DAO outperform Local ADAM even in very large-batch regimes where gradient noise is limited, each worker is running at the critical batch size, demonstrating that its benefits extend beyond compensating for small batch sizes.
    - MT-DAO remains performant as worker count increases (and per-worker batch decreases), whereas Local Adam degrades significantly.
    - MT-DAO with higher worker counts shows increases in the optimal batch size relative to DDP, corroborating previous findings [1].

Please also see SC1, SC2, and SC3 in the extensive shared concerns comment for full details.

### References

[1] Charles et al., Communication‑Efficient Language Model Training Scales Reliably and Robustly: Scaling Laws for DiLoCo

[2] Pagliardini et al, The AdEMAMix Optimizer…

---

> ### Author Response · Authors · 2025-12-03
> **Final Comment Part 2/3**
>
> ## **Part 2: Reviewer-Specific Resolutions**
>
> ### **Reviewer ${\color{magenta}{\text{uewt}}}$**
>
> Reviewer ${\color{magenta}{\text{uewt}}}$ asked for ablation experiments for $N>1$, comparisons against the AdEMAMix/Muon optimizers, comparisons against DiLoCo, and the importance of gradient clipping.
>
> **Response:**
>
> - **Baselines (AdEMAMix/Muon) and $N>1$ Ablations:** Addressed via GC1 and GC2 above. We explicitly added **AdEMAMix** comparisons as requested. MT-DAO ($N=1$) approaches the perplexity of AdEMAMix ($N=2$) with 25% lower memory usage. We further show that MT-DAO with Muon outperforms standard Muon and matches or exceeds QHMuon.
> - **DiLoCo:** Addressed via our Nesterov baseline (Fig.5 and  Fig.7), and our new Streaming DiLoCo comparison in the reply to Reviewer ${\color{red}{\text{BGje}}}$. We also re-emphasize that our method provides lower memory requirements by not having to maintain an outer optimizer state.
>     - If a single global outer optimizer state is maintained the savings are equal to a full model copy.
>     - Alternatively, if each worker maintains a copy of the outer optimizer, MT-DAO requires **25% less memory** than DiLoCo.
> - **Clipping:** We performed clipping ablations (Fig. 17). Larger models (125M+) are stable without clipping in MT-DAO. We also clarified that strict clipping parity was maintained with baselines.
>
> ### **Reviewer ${\color{red}{\text{BGje}}}$**
>
> Reviewer ${\color{red}{\text{BGje}}}$ asked how we perform against **streaming DiLoCo**, how the hyperparameters were tuned, whether $\beta$ is just compensating for a smaller batch size per worker, and whether our framework extends beyond the ADOPT optimizer.
>
> **Response:**
>
> - **Optimizer Coverage:** Addressed via **GC2** above (AdamW and Muon generalization).
> - **Batch-size Theory:** Addressed via **GC3** above (Variance theory + Batch $\times$ Worker sweeps).
> - **Streaming DiLoCo:** We implemented a **Streaming MT-DAO** variant to show that streaming-like mechanisms can be applied to MT-DAO as an orthogonal improvement. As shown in **App H.4** (**Fig. 16)**, for a 360M model Streaming MT-DAO **outperforms** Streaming DiLoCo (**8.35 vs. 8.69 PPL**).
> - **Hyperparameters:** We clarify that we had provided a $(\beta,\omega)$ search grids in Fig.8 and established a robust default: $\beta_1=0.999$, $\omega_1 \in [0.90, 0.99]$, which we have now extended for $N>1$ optimizers, for **AdamW**, and for **Muon**. We have also updated the caption of Fig.5.
>     - Crucially, we demonstrated that by controlling the learning rate via CompleteP these parameters **transfer from 16M directly to 1.3B** without re-tuning and outperform the well-tuned DDP baselines, demonstrating scalability across model sizes.
>
> ### **Reviewer ${\color{purple}{\text{CDpB}}}$**
>
> Reviewer ${\color{purple}{\text{CDpB}}}$ asked about the scalability of MT-DAO to larger models and more workers, the choice of inner-optimizer, the toy-comparison in Fig.2, the number of momenta $N$, the interaction between batch size and performance, and the recommended default values for $\omega$.
>
> - **Scalability and Inner-Optimizer Choice:** Addressed via **GC2**. We provided results for **1.3B AdamW** runs on 4 nodes with 8 GPUs showing improvements in both downstream task performance and wall-clock time over standard AdamW with DDP. We also now show results for upwards of $8$ workers in **GC3** and with different inner optimizers.
> - **Impact of $N$** Addressed via **GC1**. We validated that the two-timescale dynamics of $N=1$ are sufficient and that $N>1$ offers diminishing returns.
> - **The Impact of Batch Size on Performance:** As detailed in GC2 above, and in **Appendix H.3**, we show that MT-DAO, w.r.t Local Adam, maintains performance as worker count increases (decreasing per-worker batch). We also find that the optimal global batch size for MT-DAO **increases** with worker count.
> - **$\omega$ Choice:** We recommend the robust default: $\beta_1=0.999$, $\omega_1 \in [0.90, 0.99]$ motivated by our extensive hyperparameter sweeps as described in our answer to **${\color{red}{\text{BGje}}}$.**
> - **The Toy-Problem in Fig.2**: We clarified that LocalSGD is expected to perform well on the toy convex problem, unlike the non-convex and large-scale experiments that we conduct in our paper; we have updated the caption in Fig.2 to reflect this.
>
> ### **Reviewer ${\color{brown}{\text{KKBR}}}$**
>
> Reviewer ${\color{brown}{\text{KKBR}}}$ asked about increasing the momenta beyond $N=1$, and what happens when $K=1$.
>
> - **Benefits of $N>1$:** Addressed via **GC1** above (showing diminishing returns for higher $N$).
> - **The $K=1$ (DDP) Limit:** We explicitly tested MT-DAO at **$K=1$** vs. DDP (**App. H.6**). MT-DAO trails well-tuned DDP slightly ($\approx 0.3$ PPL for a 360M model) due to the second-moment ($\mathbb{E}[g^2] \geq (\mathbb{E}[g])^2$). This matches the method's intended scope: MT-DAO is designed to close the gap when $K \gg 1$, **not to** replace DDP.

---

> > ### Author Response · Authors · 2025-12-03
> > **Final Comment Part 3/3**
> >
> > ## **Part 3: Map of Paper Additions**
> >
> > Given the extensive number of new experiments, the table below provides a centralized overview mapping each reviewer's concern to its specific resolution in the revised paper.
> >
> > | **Reviewer** | **Summary** | **Status** |
> > | --- | --- | --- |
> > | ${\color{magenta}{\text{uewt}}}$ | 1. Experiments with $N > 1$. | ✓ (Ablated $N \in \{2,3\}$; **App. H.2**) |
> > |  | 2. Relation to AdEMAMix & Muon. | ✓ (AdEMAMix: **App. H.2**; Muon: **App. H.7**) |
> > |  | 3. DiLoCo & Streaming performance. | ✓ (Added Streaming variant; **App. H.4**) |
> > |  | 4. Gradient Clipping necessity. | ✓ (Ablated no clipping, **App. H.5**) |
> > | ${\color{purple}{\text{CDpB}}}$ | 1. Scalability (Large Models/Hardware). | ✓ (1.3B AdamW run; **§5.5, App. H.1**) |
> > |  | 2. Additional Baselines (AdamW/LocalSGD). | ✓ (AdamW: **§5.5 and App. H.1**; AdEMAMix/AggMo: **App. H.2**) |
> > |  | 3. Batch Size vs. Momentum analysis. | ✓ (Sweeps: **App. H.3**; Theory: **App. F.1**) |
> > |  | 4. Tuning weight parameter $\omega$. | ✓ (Sweeps provided; **Fig.8** and **App. H.2**) |
> > | ${\color{brown}{\text{KKBR}}}$ | 1. Performance benefits of $N > 1$. | ✓ (Showed diminishing returns; **App. H.2**) |
> > |  | 2. Behavior at synchronous limit ($K=1$). | ✓ (Analyzed + empirical results in **App. H.6**) |
> > | ${\color{red}{\text{BGje}}}$ | 1. Validation beyond ADOPT (AdamW). | ✓ (1.3B AdamW experiments; **§5.5, App. H.1**) |
> > |  | 2. Utility for Multi-Momentum optimizers. | ✓ (Validated with AdEMAMix/AggMo; **App. H.2**) |
> > |  | 3. Robustness at small batch sizes. | ✓ (Theory: **App. F.1**; Sweeps: **App. H.3**) |
> > |  | 4. Comparison to Streaming DiLoCo. | ✓ (Outperformed by 3% PPL; **App. H.4**) |
> > |  | 5. Hyperparameter tuning & Grid Search. | ✓ (Transfer 16M$\to$1.3B verified; **§5.5, App. H.2**) |
> >
> > ## Closing
> >
> > We thank the reviewers once again for their constructive contributions, and the Area and Program Chairs for their efforts in organizing the conference.
> >
> > — The Authors

---

### Meta-Review · Area_Chair_Gexc · 2026-01-07

**Summary:**

The paper studies communication-limited distributed training where DDP is bandwidth-bound and infrequent communication / local updates are attractive but tend to underperform DDP for adaptive optimizers. This paper proposes MT-DAO, a family of optimizers that inject multi-timescale first-moment tracking (multiple momenta and/or quasi-hyperbolic mixing with the instantaneous gradient) to stabilize trajectories between synchronizations while retaining reactivity. In the simplest quasi-hyperbolic form, the method uses the current gradient as the “fast leg,” adding no extra memory/communication beyond standard stateful local training.

On the theory side, they provide convergence guarantees (presented explicitly for MT-DAO-SGDM) under standard smooth nonconvex assumptions with bounded heterogeneity. Empirically (language-model pretraining), they report closing the DDP gap and improving wall-clock time in low-bandwidth settings, including the headline that at 720M scale MT-DAO reaches a target perplexity in 24% fewer steps and 35% less time than a single-momentum DDP baseline.

**Reviewer Concerns:**

Reviewer BGje

(1) Narrative unclear; claimed general plugin but experiments only on ADOPT
 — Addressed: authors add AdamW experiments up to 1.3B and explain results mirror ADOPT

(2) How is this different from prior “increase betas for small batch” observations?
 — Addressed: authors give a time-decay / local-step variance argument and provide batch×worker sweeps

(3) Streaming DiLoCo  comparison
 — Addressed: authors implement Streaming MT-DAO and report it outperforming Streaming DiLoCo


Reviewer uewt

(1) Why only N=1 in experiments?
 — Addressed: authors add N∈{2,3} ablations; argue quasi-hyperbolic variant already provides two-timescale behavior and higher N has diminishing returns

(2) Additional experiments -- Compare to AdEMAMix / Muon; feasibility under fewer comm steps
 — Addressed: authors add AdEMAMix comparisons and Muon experiments

(3) memory requirements; can MT-DAO support complex inner rules?
 — Addressed: authors give performance/memory discussion, add streaming comparison, and state framework composes with adaptive inner

(4) How critical is gradient clipping; parity with baselines?

 — Addressed: explicit parity policy and no-clipping ablation


.

Reviewer CDpB

(1) Scalability limited;

 — Addressed: authors add 1.3B AdamW results and describe hardware used

(2) Baseline choice (ADOPT vs Adam); toy example odd
 — Addressed: authors add standard AdamW baselines and clarify toy’s intent

(3) Also didnt like N=1; batch-size effect and choice of w

 — Addressed: N ablations, batch sweeps, and recommended w given

Reviewer KKBR

(1) Also concern about N -- Addressed with other reviewers with ablation

(2) Additional experiments and comparison with  additional baselines. None

**Reviewer Scores:**

The scores are high, I think concerns of all the reviewers were adequately addressed. the rebuttal is very strong.

---

### Decision · Program_Chairs · 2026-01-26

Accept (Poster)